# Hidden delays of climate mitigation benefits in the race for electric vehicle deployment

Yue Ren [1], Xin Sun[2,3,4], Paul Wolfram [5], Shaoqiong Zhao[1], Xu Tang[1], Yifei Kang[6], Dongchang Zhao[2,3,4] & Xinzhu Zheng [1]✉

Although battery electric vehicles (BEVs) are climate-friendly alternatives to internal combustion engine vehicles (ICEVs), an important but often ignored fact is that the climate mitigation benefits of BEVs are usually delayed. The manufacture of BEVs is more carbon-intensive than that of ICEVs, leaving a greenhouse gas (GHG) debt to be paid back in the future use phase. Here we analyze millions of vehicle data from the Chinese market and show that the GHG break-even time (GBET) of China's BEVs ranges from zero (i.e., the production year) to over 11 years, with an average of 4.5 years. 8% of China's BEVs produced and sold between 2016 and 2018 cannot pay back their GHG debt within the eight-year battery warranty. We suggest enhancing the share of BEVs reaching the GBET by promoting the effective substitution of BEVs for ICEVs instead of the single-minded pursuit of speeding up the BEV deployment race.

Vehicle electrification is widely perceived as an indispensable solution to climate change. According to the International Energy Agency (IEA), electric vehicles (EVs), including both light-duty and heavy-duty vehicles, enabled a net reduction of 40 million tonnes of carbon dioxide-equivalent ($CO_2$e) on a well-to-wheel basis in 2021[1]. Although assessments vary across studies due to different system boundaries and underlying assumptions, EVs' overall long-term climate benefits relative to internal combustion engine vehicles (ICEVs) in the context of electricity generation decarbonization dominate the mainstream view[2–5] (see more literature in Supplementary Table 1). For this reason, the world has experienced a rapid expansion of the EV market. In 2021, the EV stock reached 16.5 million worldwide, triple the amount in 2018[1]. A growing number of countries and regions have announced ambitious vehicle electrification targets for the coming decades[6–9]. On August 5, 2021, the White House announced a target of 50% electric for all new vehicles sold in 2030[8]. On July 14, 2021, the European Commission announced a ban on the sale of new gasoline and diesel vehicles, including hybrid vehicles, beginning in 2035[7].

China, which leads the global EV market, has made considerable efforts to support EV deployment and has taken it as one of the most effective pathways towards the carbon neutrality goal in transportation. Favoured by a series of public policy instruments and subsidies[10,11], the EV industry in China continues to evolve, and the market share of EVs has almost doubled over the past decade. In 2021, EV production and sales in China amounted to 3.5 million, a 1.6-fold increase from 2020[12]. The industry's future is promising under the ambitious vision espoused by the government. According to the New Energy Automobile Industry Development Plan (2021–2035)[6] announced by the Chinese government, the targeted penetration rate of new energy vehicles, including BEVs, hybrid electric vehicles (HEVs), and fuel cell electric vehicles (FCEVs), will reach 20% by 2025. According to the Action Plan for Carbon Dioxide Peaking before 2030[13], the market share of new energy vehicles will reach ~40% by 2030. The majority of these vehicles are expected to be BEVs, as they account for 80% of new energy vehicles[14].

Although the climate mitigation benefits of BEVs relative to ICEVs are favourable[15,16], a fact often ignored is that the benefits do not come

---

[1]School of Economics and Management, China University of Petroleum-Beijing, Beijing 102249, China. [2]China Automotive Technology and Research Center Co., Ltd, No. 68, East Xianfeng Road, Dongli District, Tianjin 300300, China. [3]Automotive Data of China (Tianjin) Co., Ltd., No. 3 Wanhui Road, Zhongbei Town, Xiqing District, Tianjin 300393, China. [4]Automotive Data of China Co., Ltd., Boxing 6th Road, Beijing Economic Development Zone, Beijing 100176, China. [5]Joint Global Change Research Institute, Pacific Northwest National Laboratory and University of Maryland, College Park, MD, USA. [6]Beijing Yiwei New Energy Vehicles Big Data Application &Technology Research Center, Beijing 100081, China. ✉e-mail: xinzhuzheng@cup.edu.cn

for "free". The production of BEVs, particularly the manufacturing of batteries, usually emits more greenhouse gas (GHG) than ICEVs[17,18]. This GHG debt can only be offset until BEVs are driven to the break-even point[19–23]. This means that the deployment of BEVs cannot yield mitigation benefits when purchased or driven on the road immediately; there is a time delay. However, such temporal distributions have often been ignored in most existing estimations when comparing BEVs and ICEVs. The life-cycle emissions of BEVs and ICEVs are usually evenly distributed per kilometre based on assumed driving mileage in the vehicles' life cycle[24] and the per-kilometre climate effect is compared. Only a minority of studies[19–23] addressed the lagging effect of climate benefits. They focused on certain vehicle models while lacking a broad view at the national scale. Filling this gap is essential for formulating deep decarbonization policies on BEV deployment and designing mitigation road maps for the transportation sector.

In this study, we present the time delay in the climate mitigation benefits of BEVs in China using vehicle-level data. The data contain almost all the BEVs (nearly 1.5 million) and 82% of the ICEVs (145.9 million) in the light-duty passenger vehicle category produced and sold in China during 2012–2018. To the best of our knowledge, this is the largest dataset that has been used to assess the climate mitigation benefits of BEVs in the Chinese market (see Methods for more details on the data), allowing us to simultaneously investigate from a full picture perspective down to a detailed one in this sector. We quantify the greenhouse gas break-even time (GBET), which describes the time that BEVs take to repay the initial GHG debt incurred by the production of carbon-intensive battery packs, by compiling life cycle assessment (LCA) with cross-vehicle comparison (see more details in "Methods").

We also perform sensitivity and uncertainty analyses to explore how the results are affected by different assumptions and comparison benchmarks. Our findings can assist with more accurate estimations of emission trends and better pathways to carbon neutrality by presenting policy-makers with the temporal characteristics of emissions in addition to the total effect of emissions. The perspectives and methods of addressing the GBET in our study can also be extended to assessing the longevity threshold of other green infrastructure investments.

## Results

### Greenhouse gas break-even time of China's battery electric vehicles

By comparing the average GHG emission level of each BEV produced and sold from 2012 to 2018 car by car with their fuel-powered counterparts (see more details in Methods), we confirm the existence of BEV GHG debt. The average emissions of producing a BEV are ~1.4 times that of an ICEV. BEVs in China would take an average of 4.5 years to offset the manufacturing "debt", with the timelines varying from zero (i.e., the production year) to over eleven years (Fig. 1). As previous GBET studies within the Chinese context are rare, we compare our estimates with those for other countries[19–23], as Supplementary Table 2 shows. The comparisons show that our GBET estimates for BEVs in China are generally longer than those in Europe[21], which is about 2–3 years. There are two potential reasons for the differences. First, the GHG emission intensity of power grids in China is higher than those in Europe, given the dominant role of coal-fired power generation in China. Higher GHG emission factors weaken the mitigation effect of

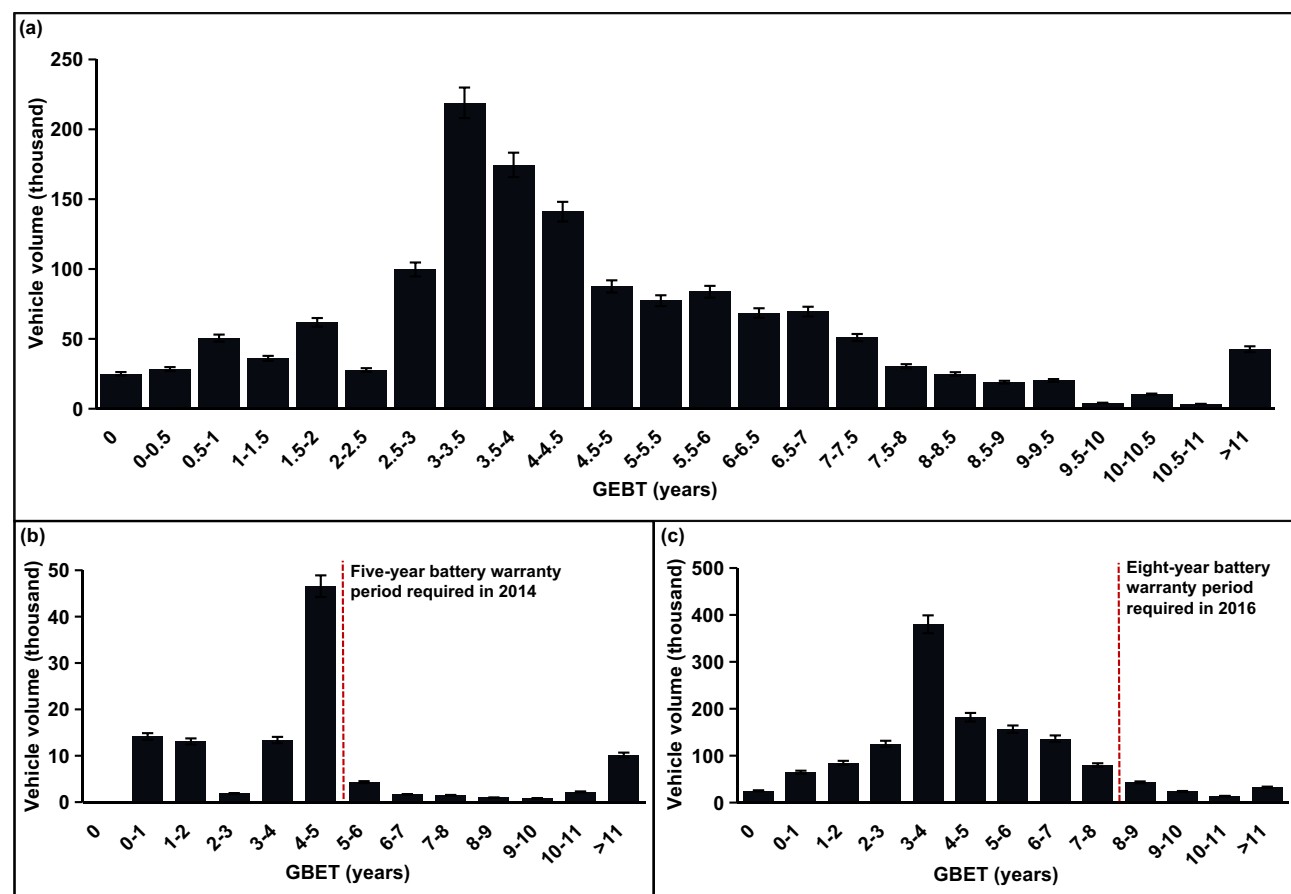

**Fig. 1 | Distribution of the greenhouse gas break-even time (GBET) of battery electric vehicles (BEVs) in the Chinese market. a** Distribution of the GBET from 2012 to 2018. **b** Distribution of the GBET from 2012 to 2015. **c** Distribution of the GBET from 2016 to 2018. According to the official battery warranty periods, five-

year warranty was required in 2014[25], and eight-year warranty was required in 2016[26]. A five-year threshold was used for 2012 and 2013 based on the 2014 requirement. The error bars indicate a 95% confidence interval. Source data are provided as a Source Data file.

BEVs in the use phase and result in higher GBET. Second, the annual VKT of most BEVs in the Chinese market (Supplementary Figs. 1, 2) is lower than 15,000 km, the assumption that has been widely used in previous studies. Lower annual VKT implies shorter effective substitution mileage for comparable ICEVs and results in higher GBET of BEVs.

The large scale of the dataset we use enables a broad view of the GBET distribution across country-wide BEVs. The shape of the GBET distribution curve is skewed, with a standard deviation of 2.4 years and a skew factor of 0.8. Approximately 70.4% of the vehicles would pay off the GHG debt within a standard deviation range (i.e., 2.1–6.9 years). Moreover, approximately one-fifth of the BEVs produced and sold before 2016 failed to pay back the GHG debt within five years, which was the EV battery warranty period required by the Chinese government in 2014[25]. In 2016, the required battery warranty was extended to eight years[26], and 8% of BEVs produced and sold between 2016 and 2018 cannot achieve the GBET within the battery warranty threshold (Supplementary Table 3). The BEVs with zero GBET emit less GHG emissions in the vehicle cycle than their oil-powered counterparts and only account for 1.7% of the total sample. These "zero-GBET" BEVs are predominantly A00-class cars, which have significantly lower battery capacity and lighter curb weight than other vehicle classes. The BEVs whose GBET is over 11 years account for ~2.9% of the total sample. A total of 97.8% of these BEVs are in the MPV-A0 class. This occurrence is probably because the weight difference in MPV-A0 vehicles between electricity-powered and fuel-powered vehicles is the largest (see more details in Supplementary Table 4), resulting in an enormous GHG debt and the longest GBET.

Two contradictory trends influence the year-over-year changes in BEV GBETs. On the one hand, with battery technology advancement, the battery capacity and the driving range of BEVs increase, offering more effective substitution mileage for ICEVs and resulting in a decreasing trend in GBET. On the other hand, higher-capacity batteries usually have a larger size, heavier weight, and heavier curb weight for support, which probably leads to more carbon debt in the production phase and therefore needs a longer time to repay. Under the combined effect of these two trends, the GBET of BEVs produced and sold between 2012 and 2018 shows a fluctuating trend, which varies by vehicle class (Table 1). For example, the GBET of A-class Cars and A0-class SUVs shows an overall increasing trend with year-to-year fluctuations, while that of B-class MPVs generally decreases.

The GBET of BEVs also shows significant heterogeneity among various transport modes (Car, SUV, and MPV) and size classes (A00, A0, A, B, and C) (see more details of the vehicle classification in Supplementary Table 5). The impact of influencing factors is bidirectional as well. On the one hand, heavier transport modes and larger vehicle sizes usually have larger battery capacity and heavier battery weight, resulting in higher GHG emissions in the production phase and, thus, more GHG debt (see Supplementary Table 4). This trend potentially increases GBET. On the other hand, the fuel-powered counterparts of heavier transport modes and larger vehicle sizes are energy intensive (see Supplementary Table 4) and emit more GHGs during the fuel cycle, resulting in more notable emission reduction benefits of BEVs relative to ICEVs and shorter GHG debt pay-off periods. This trend potentially decreases GBET. Under the combined effect of these two trends, the GBET of BEVs shows an overall increasing trend with larger size classes (A00 < A0 < A < B) and larger transport modes (Car < SUV < MPV).

More specifically, the effects of transport mode and size class interact. The impact of transport mode varies across size classes. For A0-class vehicles, the GBET increases in the order of Car, SUV, and MPV. This order implies that the increasing impacts of more GHG debt caused by the heavier weight of larger transport modes exceed the decreasing impacts caused by improving the debt repayment efficiency during the fuel cycle (the terminology fuel is used conventionally, referring to electricity production, transmission, and use for BEVs). For A-class and B-class vehicles, the GBET of the car is the largest (6.3–7.3 years), the MPV is in the middle (5.8–6.1 years), and the SUV is the smallest (3.1–4.8 years). This indicates that under these two size classes, the positive effect of the increase in the rate of debt repayment in the fuel cycle of SUVs and MPVs completely offsets the negative effect due to the increased curb weight. Similarly, the effect of size class on GBET is related to the transport mode. For Cars, GBET shows an increasing trend with a larger size. The GBET prolongation effect caused by the increase in GHG debt with the increase in size class exceeds the reduction effect caused by the increase in fuel-cycle emission reduction. SUVs and MPVs show the opposite trend: GBET decreases with increasing size class. In this case, the relative advantages of BEVs in fuel-cycle emission reduction brought by the increase in size class are more dominant. Thus, overall, SUVs and MPVs with larger size classes and cars with smaller size classes have shorter GBETs.

## Regional heterogeneity of battery electric vehicles' greenhouse gas break-even time

The GBET of BEVs in China varies significantly across provinces (Fig. 2a). The average GBET of BEVs produced and sold in 2018 is 6.9–7.9 years in the northeastern provinces, 2–6 years longer than the average in the southwestern provinces. The cross-province variances of four factors (Fig. 2b), including the composition of size classes, the composition of transport modes, annual vehicle kilometres travelled (VKT), and the local power grid's GHG emission intensity, might explain the regional heterogeneity of GBET. In the same year, the share of A00-class vehicles in the total BEV sales by province ranged from 5.7% in Qinghai to 90.2% in Guangxi, with a mean of 42.4%. The share of Cars in the total BEV sales by province ranged from 26.4% in Jilin to 96.3% in Guangxi, with a mean of 70.3%. The 2018 provincial average of annual VKT ranged from 678 km in Tibet to 15,927 km in Guangdong, and the provincial average of GHG emission intensity of electricity production ranged from 38 $gCO_2e/kW$ in Tibet to 801 $gCO_2e/kW$ in Tianjin, a range of twenty times greater at its high end than at its low end.

For the BEVs produced and sold in 2018, the top five provinces with the longest GBET were Jiangxi, Jilin, Heilongjiang, Liaoning, and Tibet. For the three northeastern provinces (i.e., Jilin, Heilongjiang, Liaoning), the reasons for the long GBET are similar, as these provinces have larger vehicle sizes/modes, higher GHG emission intensities for local power grids and relatively lower VKT. The higher GHG emission intensity is consistent with the coal-dominant power generation structure in these regions, and the lower VKT implies that BEVs are not sufficiently used, leading to more time to pay back the GHG debt. Jiangxi province's vehicle sizes and transport modes are relatively small, but the high GHG emission intensities and the low VKT contribute to the long GBET. The long GBET in Tibet is mostly caused by its extremely low VKT, which is nearly 90% less than the national average on a yearly basis. This can probably be explained by the lagging progress in the construction of charging infrastructure for BEVs and the special topography of Tibet (e.g., the limited driving range of electric vehicles may not meet the local long-distance travel needs). Although Tibet has the lowest GHG emission intensity for its power grid, as coal power generation accounts for only 1.5% of the total provincial amount, the lowest utilization rate of electric vehicles in Tibet completely offset the climate benefits brought by its low-carbon electricity generation structure and explains why it has the long GBET. The five provinces with the shortest GBET are Guizhou, Guangxi, Hunan, Tianjin, and Sichuan. Smaller vehicle sizes/modes, cleaner electricity mix, or higher VKT explains the short GBET of these provinces.

**Table 1 | The descriptive statistics of greenhouse gas break-even time (GBET) of battery electric vehicles (BEVs) in the Chinese market by transport mode and size class**

| Transport mode | Size class | 2012 | 2013 | 2014 | 2015 | 2016 | 2017 | 2018 |
|---|---|---|---|---|---|---|---|---|
| Car | A00 | 2.2** | 3.1** | 2.3*** | 2.5*** | 3.5*** | 3.3*** | 3.2*** |
| | | (0.8) | (1.1) | (1.7) | (2.0) | (2.3) | (1.9) | (1.7) |
| | A0 | 4.2** | 4.1** | 4.2** | 4.2*** | 3.8*** | 3.3*** | 3.8** |
| | | (0.2) | (0.2) | (0.3) | (0.5) | (0.5) | (0.6) | (1.3) |
| | A | 5.0* | 6.2* | 4.9* | 7.1** | 7.1*** | 6.4*** | 6.3*** |
| | | (0.5) | (0.3) | (0.8) | (2.6) | (1.3) | (1.6) | (1.6) |
| | B | – | – | 8.1* | 9.8** | 7.8** | 7.5** | 7.3** |
| | | | | (0.6) | (1.2) | (0.6) | (1.7) | (1.7) |
| SUV | A0 | 3.7* | 3.7* | 3.6* | – | 5.1** | 5.7** | 5.2*** |
| | | (0.1) | (0.1) | (0.1) | | (1.1) | (1.2) | (1.2) |
| | A | – | – | 5.0* | 3.2* | 5.6** | 5.0*** | 4.3*** |
| | | | | (0.1) | (0.9) | (1.2) | (0.7) | (1.0) |
| | B | – | – | – | – | 3.8* | 4.1* | 3.1** |
| | | | | | | (0.1) | (0.2) | (0.3) |
| | C | – | – | – | – | – | – | 4.8*** |
| | | | | | | | | (0.7) |
| MPV | A0 | 8.6** | 9.4** | 10.5** | 10.9** | 10.9*** | 7.8*** | 8.4*** |
| | | (3.3) | (2.2) | (1.7) | (0.7) | (0.6) | (3.6) | (3.1) |
| | A | – | – | – | – | | 7.2* | 6.1** |
| | | | | | | | (1.1) | (0.6) |
| | B | – | – | – | – | 9.0* | 5.2** | 5.8* |
| | | | | | | (1.4) | (1.6) | (2.3) |

A short string indicates insufficient data for the transport mode in the given year. The values in parentheses represent the standard deviations. Low, medium, and high confidence levels correspond to the sample size <1000, [1000,10,000], and >10,000 represented, denoted by *, **, and ***.
*SUV* sports utility vehicle, *MPV* multi-purpose vehicle.

## Sensitivity and uncertainty analysis

The estimations of GBET are mainly affected by two sources of uncertainty. One is the parameter uncertainty, and the other is the uncertainty due to using different comparison benchmarks. In terms of parameter uncertainty, we cluster all the influencing factors into ten groups (Supplementary Table 6) and perform a one-variable-at-a-time perturbation sensitivity analysis for each grouped variable (see more details in "Methods"). The top six sensitive factors, in descending order, are curb weight, GHG emission factors of vehicle material production, battery capacity, GHG emission factors of battery material production, annual VKT, and GHG emission factors of power grids (Supplementary Table 7). We can tell that GBETs are more sensitive to vehicle-cycle factors (the former four factors) than to fuel-cycle factors (the latter two). This is different from previous LCA studies, which revealed stronger effects of fuel-cycle factors on life-cycle emissions than vehicle-cycle ones[5,27–29]. The reason for the different analysis results is that the estimation of GBET only considers the GHG emissions before the break-even point, while the LCA considers life-cycle emissions—the smaller scale of the former results in a weakened influencing power of fuel-cycle factors. The six sensitive factors are then included in the uncertainty analysis using the range approach and orthogonal experimental design (OED) method (see more details in Methods). The results (Supplementary Fig. 3) show that the national average GBET (4.5 years in the basic estimation) drops to -1.9 years in the scenario of the lower extremity with radical annual VKT increase, lighter weight vehicles, cleaner power grid, and low-carbon material emission factors while increases to 6.5 years in the upper extremity.

Moreover, the GBET of BEVs also highly depends on the comparison benchmark of ICEVs, which varies over a large range. For the robustness check, we pair BEVs with ICEVs across different size classes and include both the most and least efficient ICEVs as benchmarks to present pessimistic and optimistic GBET estimations. Compared

between adjacent size classes, the GBET estimations fluctuate from −74 to 156% (Supplementary Table 8). Compared with the ICEVs in the same size class but whose GHG emissions are in the lower quartile (i.e., top 25% low-emission ICEVs), the GBET increases by 1.9–6.7 years, with an average increase of 3.9 years (Supplementary Fig. 4). In this case, nearly half of the BEVs produced and sold in 2018 cannot repay GHG debt within 11 years. When we change the benchmark to the ICEVs whose GHG emissions are in the higher quartile (i.e., top 25% high-emission ICEVs), the GBET decreases by 1.6–5.1 years, with an average decrease of 2.9 years (Supplementary Fig. 4). In this case, all BEVs sold in 2018 achieve the GBET within 7 years and 95% of them actually within 3 years.

## Discussion

The GBET estimation in this study alerts policy-makers that BEVs' climate benefits do not come for free but are conditional upon the time needed to pay back their GHG debt incurred in the vehicle production stage. This circumstance also brings the understanding of the delayed climate benefits of China's BEVs from an abstract level to a concrete threshold. Such findings have enormous implications for the real world.

First, new GBET-based indicators can be developed to guide BEV deployment. For example, the percentage of BEVs that have reached GBET (P-GBET) is an indicator to supplement the widely used indicator BEV penetration rate (PR). In other words, it is not only how many BEVs are produced and sold but also how many of them have positive emission reductions that contribute to the climate benefits of the transportation sector. So far, China and many other countries have leveraged multiple procurement incentives, such as tax credits, discounts or rebates, exemption of BEVs from congestion controls, and separate licence plate quotas for BEVs[30–33], to pursue higher PR. However, once the sale is complete, these policies are no longer in

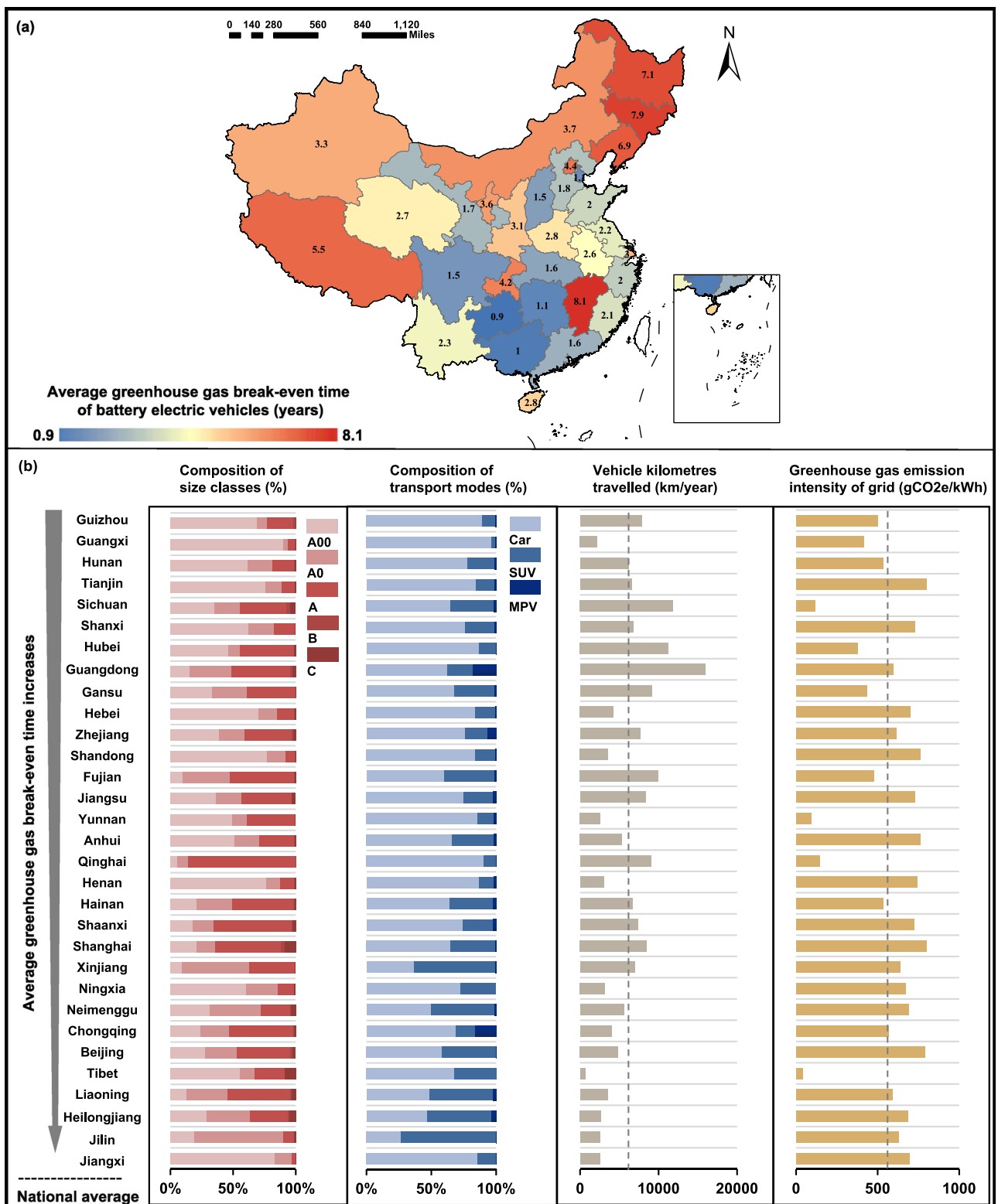

**Fig. 2 | The greenhouse gas break-even time (GBET) of battery electric vehicles (BEVs) and influencing factors by province in 2018. a** The average GBET of BEVs by province in 2018. Data for the Hong Kong Special Administrative Region (SAR), Macao SAR, and Taiwan province are unavailable. **b** Four influencing factors of GBET by province in 2018, including the composition of vehicle size classes, the composition of transport modes, the average annual vehicle kilometres travelled, and the greenhouse gas emission intensity of grids at the provincial level. SUV sports utility vehicle, MPV multi-purpose vehicle. Source data are provided as a Source Data file. The base map use in **a** was applied without endorsement using data from the standard map service published by the Ministry of Natural Resources (http://bzdt.ch.mnr.gov.cn/).

effect, leaving the true climate effect unmanaged[34]. A direct solution to this problem is to set up follow-up policies for BEV deployment, such as adopting stage subsidies (i.e., extending the subsidy timeline from the purchase time point to the time when it achieves its GBET), investing in charging infrastructures, and motivating ICEV replacement, to promote higher P-GBET.

GBET can also assist in setting technical standards for the life expectancy or battery replacement time of BEVs to ensure net climate benefits. For example, China currently requires an eight-year or 120,000 kilometres warranty (whichever occurs first) on EV batteries[26]. However, as our estimations in the previous sections show, not all vehicles can achieve their GBET within the battery warranty period, especially for some large-size or large-mode vehicles. This situation calls for a longer required warranty period for heavy transport modes. In fact, GBET provides guidelines for differentiated warranty periods and other longevity-relevant standards. This process, on the one hand, facilitates climate benefits by avoiding the early replacement of batteries and, on the other hand, motivates BEV suppliers to improve the climate performance of their products.

It is worth mentioning that a smaller GBET does not necessarily yield higher emission reduction over the life cycle. For example, BEVs with a longer expected driving range tend to have a longer lifetime, thus resulting in more emission reduction benefits over the full life cycle. However, increased driving range often relies on larger battery capacity and heavier weight, which increases GHG emissions in battery production and leads to higher GHG debt. Thus, it will take longer to pay these GHG debts, leading to longer GBET. The GBET is a supplementary indicator to the existing metrics, as it provides information on how quickly the climate benefits are generated, while previous LCA assessment suggests the scale of the benefits throughout the vehicle's lifetime. Both information should be considered in assessing the climate mitigation effects of vehicle electrification.

In addition, although trade-offs might exist between life-cycle emissions reductions and faster payback times, there is still room for synergy. Policy-makers can encourage more explorations in more and faster BEV emission reductions, for example, reducing the carbon debt by vehicle lightweighting[20,35–39], material recycling[40,41], battery recycling and reuse[42–44]. Another strategy is to accelerate the GHG debt repayment rate by intensifying the usage of existing BEVs via vehicle sharing or prioritizing BEVs as taxis[45,46]. Intensifying BEV usage by vehicle sharing rather than expanding vehicle ownership would simultaneously shorten GBET, achieve more GHG emission reductions and solve other problems, such as traffic congestion, mineral resources depletion, infrastructure construction pressure and environmental pollution[47,48]. This strategy is feasible, as essentially what people truly need is high-quality transportation service rather than the vehicle itself[49]. Moreover, aligning BEV production sites with the planned renewable power grid can facilitate faster and more BEV GHG emissions reductions[50]. Currently, China's BEV and battery productions are mainly located in the southeast coastal region and the northeast, where grid GHG emission intensity is relatively high (see more details in Supplementary Fig. 5). The geographical spread of battery and car manufacturers in China is determined by historical production advantages, such as the availability of mature production lines. For example, Contemporary Amperex Technology Co., Limited (CATL), the largest EV battery manufacturer in China, initially produced phone batteries. Its historical production advantages facilitate agglomeration externalities, technology spillover, and productivity gains, allowing it to quickly shift to BEV battery production. As a step forward to match the low-carbon development of power grids, CATL constructed more factories in southwestern provinces with abundant renewable energy, such as the first-zero-carbon factory built in Yibin, Sichuan province[51]. Incorporating cleaner electricity production into the layout of BEV production is favourable for both shortening GBET and reducing life-cycle emissions.

Although our findings have great implications, we also notice that there are several limitations. First, we assume that a BEV's yearly effective substitution mileage for ICEVs is the annual average VKT in the province it is sold, without considering the rebound effect or spillover effect[52] of BEV uptake on GHG emissions. This assumption might bias the GBET estimation. In the scenario where the first-time car owner purchased a BEV to replace public transportation service rather than ICEVs, the effective substitution mileage is lower than the BEV's annual VKT and results in an underestimation of the GBET. In the other scenario where a BEV is bought to actually replace the already owned ICEV, due to the limited driving range of the BEV, the user might reduce the car usage compared to when owning an ICEV. Here a positive spillover effect occurs, and the effective substitution mileage is higher than the BEV's annual VKT. In fact, to what extent BEVs effectively substitute ICEVs is complicated, as it is relevant to consumers' behaviours[34]; this relationship has not been fully discussed and leaves ample opportunities for future research.

Second, while we distinguish the GBET of the same vehicle model (1894 models in total from 2012 to 2018) by production and sale location, other real-world data at the vehicle-level level, including the on-road fuel efficiency and the real-time GHG emission factor relating to the charging time, are not acquired. The lack of these data would bias GBET estimations. The difference between on-road fuel efficiency and official fuel efficiency reported by manufacturers varies, influenced by real-world environmental factors and driving behaviours[53,54]. The use of annual average power grid emission factors without considering the seasonal and daily effects on the electricity mix might underestimate our estimation of GBET. For example, most of the BEVs in Shanghai are charged at night[55], when the grid emission intensity is higher than average since residential electricity demand peaks and less renewable energy is available for power generation at this time[56]. Using the marginal electricity emission factors[57] allows for more accurate estimations, though doing so is challenging due to the lack of data.

Third, we have not considered the effect of battery recycling, the degradation process, or the vintage effect on EV energy consumption and GHG emissions. EVs that use secondary-use and recycled batteries have a much lower GHG debt than initially produced EVs[58]. Considering battery degradation[59], the GBET of BEVs could be longer than the estimations. It is also worth mentioning that additional delays stem from older, less efficient vehicle stocks remaining in the car fleet for a long time[60].

Despite the limitations, our study expands the understanding of BEVs' climate benefit delays from a conceptual level to a concrete threshold measure using Chinese market data. The scale of the data allows us to simultaneously investigate from a full-picture perspective (i.e., national perspective and regional heterogeneity) down to a detailed one (vehicle model perspective). This study is a timely reminder for policy-makers to pay more attention to the temporal distribution of climate effects and provide guidelines for BEV deployment policies and longevity standard design. GBET-based indicators, such as the share of BEVs that have achieved the GBET, could be a vital supplemental factor for existing indicators of BEV penetration rate. They provide additional dimensions for policy-makers to consider, especially in promoting the effective substitution of BEVs for ICEVs, rather than just speeding up the BEV deployment race.

## Methods

The GBET of BEVs is defined as the time needed to repay the GHG debt incurred during the production of carbon-intensive battery packs and associated vehicle materials. The GBET estimations are based on the life cycle assessment (LCA) of vehicle GHG emissions and cross-vehicle comparisons between BEVs and their fuel-powered counterparts year by year. We begin this section by establishing the LCA setup. Then, we present how BEVs are paired with ICEV benchmarks for GBET estimation, as well as the data sources and key assumptions. Sensitivity and

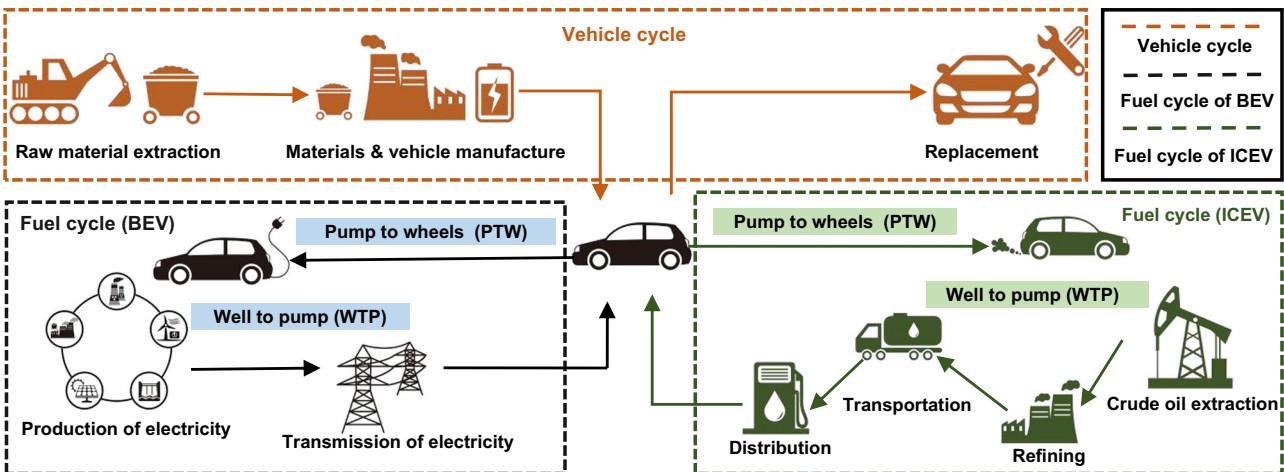

**Fig. 3 | System boundary of life cycle assessment for both battery electric vehicles (BEVs) and internal combustion engine vehicles (ICEVs) in this study.** The life cycle system boundary evaluated in this study includes the vehicle cycle and fuel cycle of passenger vehicles. The vehicle cycle starts with raw material acquisition, then moves to material processing and manufacturing, complete vehicle production, and maintenance (tire, lead battery, and fluid replacement). The fuel cycle refers to "Well to Wheels (WTW)", including the production of fuel (Well to Pump/WTP) and the use of energy (Pump to Wheels/PTW). For BEVs, the fuel terminology is used in a conventional sense, referring to electricity production, transmission, and use.

uncertainty analysis is finally performed to demonstrate how the results change with different parameter assumptions and various paired ICEV benchmarks.

## Life cycle assessment of greenhouse gas emissions

The GHG emissions of BEVs and ICEVs are estimated using the China Automotive Life Cycle Assessment Model (CALCM). The functional unit of this LCA is 1 km travelled by a passenger vehicle in China during 11 years. The model is the compilation and evaluation of a vehicel system's inputs, outputs, and potential environmental impacts over its life cycle[61]. Here, we followed the instructions of national standards GB/T24044–2008[62], GB/T 24040–2008[63], and international standard ISO 14067-2018 to perform the assessment[64]. For both BEVs and ICEVs, the life cycle system boundary evaluated in this study includes the vehicle cycle and fuel cycle of passenger vehicles. The vehicle cycle starts with raw material acquisition, then moves to material processing and manufacturing, complete vehicle production, and maintenance (tire, lead battery, and fluid replacement). The fuel cycle refers to "Well to Wheels (WTW)", including the production of fuel (Well to Pump/WTP) and the use of energy (Pump to Wheels/PTW). For ICEVs, WTP includes crude oil extraction, refining, and processing; PTW refers to fuel combustion. For BEVs, the fuel terminology is used in a conventional sense, referring to electricity production, transmission, and use. GHG emissions of BEVs in WTP occur with electricity production and transmission, while GHG emissions of BEVs in PTW are zero, as there is no GHG emission during the use phase of electricity. The transportation of materials, the manufacturing of equipment and infrastructure, and the production and treatment of manufacturing wastes are excluded (Fig. 3).

## Calculation of greenhouse gas break-even time

The estimations of the GBET include two phases: the matching of the BEVs with their fuel-powered counterparts and the calculation of the GBET by comparing the matched vehicles. The methods of these two phases are described below.

Since the GBET of BEVs is calculated at the vehicle level, we find fuel-powered counterparts for each of the BEVs produced and sold from 2012 to 2018. One BEV can have multiple fuel-powered counterparts because, in the real world, a certain BEV can be perceived as a possible substitute for many fuel-powered vehicles. Considering that in most cases, the replacements happen in the same vehicle class, we

compare each of the BEVs with the ICEVs in the same vintage, transport mode (Car, SUV, and MPV), and size class (A00, A0, A, B, and C) (see more details of the vehicle clarification in Supplementary Table 5) to generate basic estimations, referring to Fig. 4. As the comparison is "one (BEV) to many (ICEVs)," for systematic comparison, we generate a representation of the selected ICEVs, whose parameters are the average of the matched counterpart ICEVs. Then, the comparison turns to "one (BEV) to one (representative ICEV). Then we calculate the average GBET of BEVs within the same stratum to generate an overall estimate. In the uncertainty analysis, we consider more possibilities of the substitutes across different classes and more possibilities of the representative ICEVs (see sensitivity and uncertainty analysis for more details).

Using the representative ICEVs as the benchmark for comparison, we calculate the GBET of BEVs at the individual vehicle level. The differences in vehicle-cycle GHG emissions between BEVs and ICEV benchmarks are first estimated by Eq. (1), revealing the magnitude of GHG debt. Then, annual paid-back GHG emissions are calculated by comparing the yearly emissions of BEVs and ICEV benchmarks when they are driven for the effective substitution mileage, as shown in Eq. (2). We assume that the effective substitution mileage is the BEVs' annual VKT. When the cumulative paid-back emissions equal the GHG debt, the break-even time is achieved, as Eq. (3) shows.

$$E_{debt}(t_0) = E_{BEV}(t_0) - E_{ICEV}(t_0) \qquad (1)$$

$$E_{payback}(t) = \sum_{t_0}^{t}(E_{BEV}(t) - E_{ICEV}(t))\,(t \geq t_0) \qquad (2)$$

$$GBET = \begin{cases} 0, \text{if } E_{debt}(t_0) \leq 0 \\ t', \text{s.t } E_{debt}(t_0) + E_{payback}(t') = 0 \end{cases} \qquad (3)$$

where $E_{debt}(t_0)$ is the GHG debt of BEVs relative to ICEV counterparts in the production year $t_0$. $E_{BEV}(t_0)$ and $E_{ICEV}(t_0)$ are vehicle-cycle GHG emissions of BEVs and ICEVs, respectively. $E_{payback}(t)$ denotes the cumulative GHG payback by year $t$. $E_{BEV}(t)$ and $E_{ICEV}(t)$ are yearly fuel-cycle emissions of BEVs and ICEVs, respectively, when they are driven for the effective substitution mileage in year $t$. If $E_{debt}(t_0) \leq 0$, the GBET is zero, meaning that the BEV emits fewer GHGs than its comparable

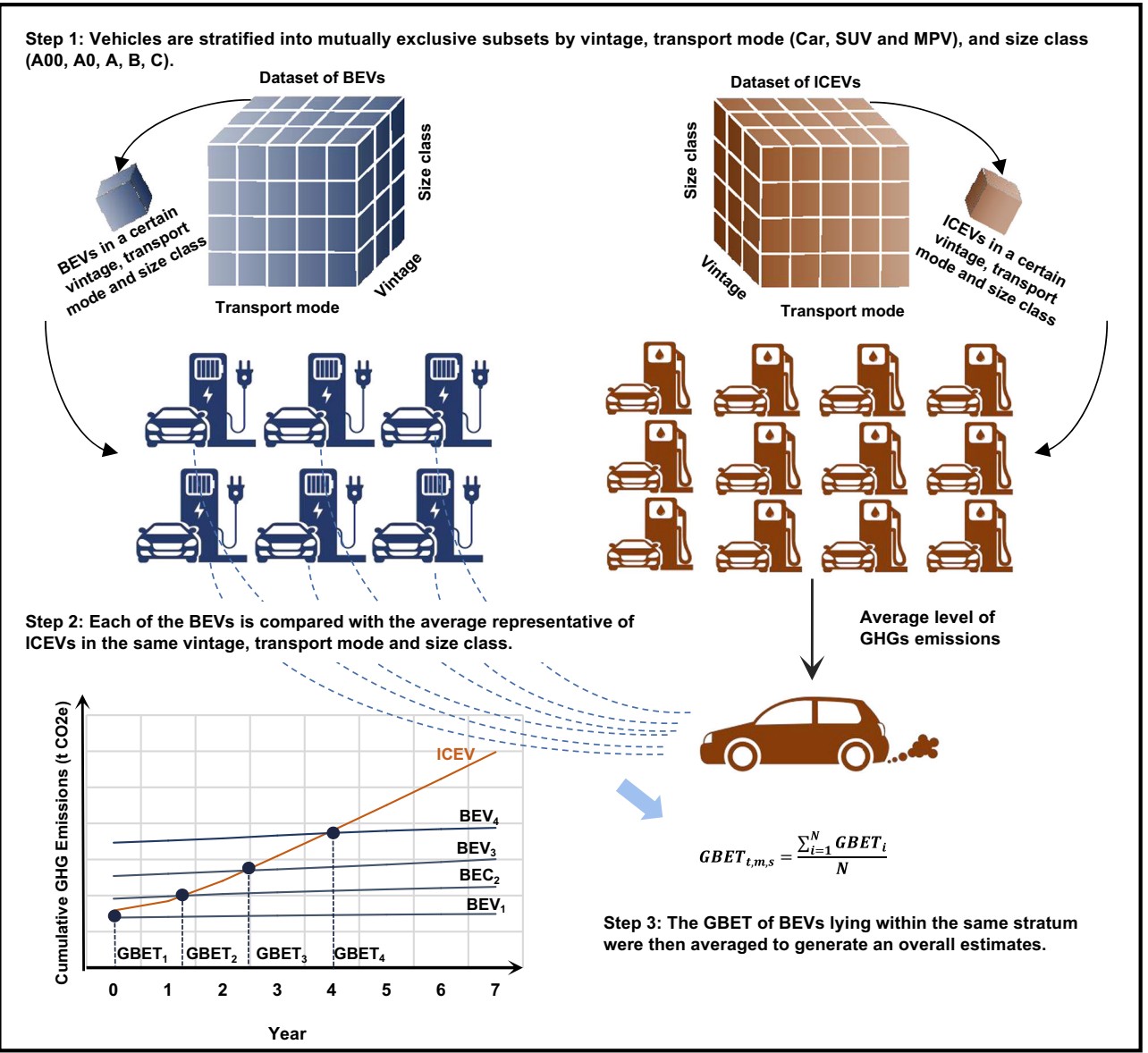

**Fig. 4 | Scheme of the greenhouse gas break-even time (GBET) estimation process and matching methods between battery electric vehicles (BEVs) and internal combustion engine vehicles (ICEVs).** SUV sports utility vehicle, MPV multi-purpose vehicle.

ICEV benchmark. Otherwise, the GBET is year $t'$, when the BEV paid back its GHG debt for the first time. The year-to-year changes in fuel-cycle emissions are considered for both BEVs and ICEVs. Since our calculations are on a yearly basis, within the same year, we assume that cumulative emissions increase linearly. In other words, when we find an integer interval $[t, t + 1]$ where $E_{debt}(t_0) + E_{payback}(t) > 0$ and $E_{debt}(t_0) + E_{payback}(t+1) < 0$, we use the linear interpolation method to find the exact time $t'$.

### Data sources and assumptions

The data used in the estimations of GBET can be classified into four categories according to their resolution levels (see Supplementary Table 9). The first category is the real-world vehicle-level dataset, which contains the model type, year, and location of production and sales of almost all BEVs (nearly 1.5 million units) and 82% of ICEVs (145.9 million units) in China from 2012 to 2018 (see Supplementary Figs. 5, 6). The dataset came from China's Compulsory Traffic Accident Liability Insurance (CTALI), which is provided by the China Automotive Technology & Research Center (CATARC)[65,66].

Since CTALI is compulsory for every vehicle registered in China, the data have wide coverage and high credibility. The vehicle-level data allow us to distinguish the GBET of BEVs across vehicle models, years, and locations.

The second category is vehicle model information. The CTALI database recorded 227 types of BEV models and 1667 types of ICEV models from 2012 to 2018. For each vehicle model type, more technical specifications, including the model type, curb weight, battery weight, battery capacity, and fuel consumption, were collected from the Announcement of Vehicle Manufacturing Enterprises and Vehicle Products[67] which is governed by the Ministry of Industry and Information Technology (MIIT) of China. The fuel consumption for each model type was based on the New European Driving Cycle (NEDC) testing conditions[68]. More statistical descriptions of the data are provided in Supplementary Figs. 7–9. These technical details were used in the LCA analysis, enabling the estimation of the GHG emissions at the vehicle model level. Moreover, by combining this information with each vehicle's production year and sale location, we can identify the regional heterogeneity of GBET for the same vehicle model using the

VKT data and GHG emission factors of power grids that vary across provinces.

The third category of data is reported at the province level, including annual Vehicle Kilometres of Travel (VKT) for both BEVs and ICEVs and emission intensity of power grids. The VKT data for 2018 were extracted from the National Big Data Alliance of New Energy Vehicles (NDANEV)[69], which records the real-world driving, charging, and fault status of vehicles car by car. According to the requirements of national standard GB/T 32960[70], the data are uploaded to the platform every 30 seconds when the vehicle is driving, and the fault state is uploaded every second. Between 2018 and July 17, 2022, the NDANEV accessed 9.27 million new energy vehicles with a total VKT of 295.5 billion kilometres. Although the data in NDANEV are car-by-car, vehicle-level VKT data were not used in the GBET analysis since we had no access to the vehicle identification information to match the NDANEV database with the CTALI database. Thus, we aggregated the data at the provincial level, assuming that the VKT of vehicles within the same provinces is homogeneous. More statistical information on the real-world VKT data is provided in the SI (see Supplementary Figs. 1, 2).

In the estimation of GBET, the vehicle's VKT data by province change annually. Based upon the real-world data from NDANEV in 2018, we projected the VKT before and after 2018 using two methods: one holds conservative attitudes towards the VKT increase, while the other is more radical. Under the conservative estimation, the targeted VKT in 2030 by province contains five levels, i.e., 18,000, 15,000, 13,000 km, 12,000, and 8000 km, to reflect the regional heterogeneity in new energy vehicle development speed. The VKT in each province before and after 2018 was then projected linearly, assuming that the VKT increases at distinct speeds across provinces (see Supplementary Table 10). When using radical estimation, we set a more ambitious VKT goal for 2030, reflecting a possible scenario where passenger BEVs and the charging infrastructures in China would develop dramatically (see Supplementary Table 11).

The emission intensity of electricity generation by province was calculated based on the power generation structure and GHG emission factors across power generation types, assuming that the electricity consumption structure is the same as that of electricity generation. Such an assumption might underestimate the emission intensity because marginal electricity consumption for BEVs usually relies on coal and natural gas power plants whose operations are relatively stable with higher GHG emission intensities than the grid structure. The provincial power generation structure data from 2012 to 2019 were acquired from the China Electricity Council[71], and those from 2020 to 2028 were from forecasting data referenced from Li et al.[72]. The GHG emission factors of different power generation technologies (i.e., coal, wind, solar, nuclear, etc.) were referenced from the IPCC Fifth Assessment Report (AR5)[73]. In the basic estimations, we used the medium value reported by IPCC AR5; this value falls within the range of most existing research on GHG emissions from power generation technologies in China (see more literature in Supplementary Table 12). We employed the maximum and minimum values from existing research in the uncertainty analysis. The results of the emission intensity of the power grid by province are presented in Supplementary Tables 13–15.

The last category of data is the life cycle inventory (LCI) data from the latest China Automotive Life Cycle Database (CALCD)-2021 (see Supplementary Tables 16, 17), developed by the CATARC[66]. These data are homogeneous across provinces. We compared the LCI data from CALCD-2021 with two internationally well-known LCI databases, the Greenhouse gases, Regulated Emissions, and Energy use in Technologies Mode (GREET) and ecoinvent 3.6[74,75]. We found fairly high consistency among these databases (see Supplementary Table 17).

## Sensitivity and uncertainty analysis

We considered two sources of uncertainties that might modulate the estimations of GBET. One is the parameter uncertainties in LCA analysis, and the other is the matching methods between BEVs and ICEVs.

We performed a one-variable-at-a-time perturbation sensitivity analysis of the input parameters (Supplementary Table 6) that influence the GBET. For each variable, sensitivity coefficients ($\sigma_i$) were calculated, indicating the percentage change in GBET when the variable changed by 1% (Eq. 4).

$$\sigma_i = \frac{\frac{GBET_i - GBET'_i}{GBET'_i}}{\frac{Inf_i - Inf'_i}{Inf'_i}} \quad (4)$$

where $GBET'_i$ represents the value in the case of the first (baseline) solution; $GBET_i$ represents the value of GBET under the assumed change of variable $i$; $Inf'_i$ denotes the initial value of variable $i$; and $Inf_i$ represents the changed variable $i$. Higher sensitivity coefficients denote higher sensitivity of GBET estimation to the changes of variables. Overall, we include ten influencing factors in our analysis. Since the LCI data are large in volume, we grouped them into four factors for ease of execution: GHG emission factors of vehicle material production, GHG emission factors of battery material production, Electricity consumption during the vehicle production stage, Electricity consumption during the battery production stage (see more details in Supplementary Table 6). The rest six factors are obtained directly from the databases we use. Based on the sensitivity analysis results, we identified the top six sensitive factors in descending order: curb weight, GHG emission factors of vehicle material production, battery capacity, GHG emission factors of battery material production, annual VKT, and GHG emission factors of power grids (shown in Supplementary Table 7).

We further considered these sensitive factors in the uncertainty analysis. The most common method of uncertainty analysis is the Monte Carlo simulation. However, it is challenging for the GBET estimation as the distribution curve of multiple input parameters, especially those of the LCI data, are elusive. Here, we conduct uncertainty analysis by combining the range approach with orthogonal experimental design (OED). The range approach tests the effects of sampling the parameters at the extreme of their ranges of variability on the output uncertainty[76,77] so as to avoid making a judgement about the probability of different occurrences[78]. We assumed a uniform coverage of the uncertainty input space, i.e., ±5%, for these factors: curb weight, battery capacity, GHG emission factors of vehicle material production, and GHG emission factors of battery material production. For the GHG emission intensity of power grids, we used the GHG emission factors of power generation from the IPCC report[73] in the basic estimation and the low or high values referenced from studies in the Chinese context as the two extreme ends (Supplementary Tables 13–15). For the annual VKT, we considered a conservative scenario and a radical development scenario, respectively, to reflect the variations (Supplementary Tables 10, 11). The OED is an effective method for arranging and analyzing multi-factor interactions. As an alternative to presenting all combination forms of multiple factors, the OED method efficiently schedules multifactorial experiments with optimal combination levels[79,80]. For the above six sensitive factors, we used an orthogonal table (Supplementary Fig. 3) containing 18 representative scenarios to investigate their combined impacts, following the guidelines of ref. 81.

The GBET of BEVs also highly depends on the comparison benchmark of ICEVs. In the basic estimations, we used the average level of ICEVs in the same vehicle classification (i.e., production year, transport mode, and size class) as the benchmark for each BEV. Considering the possibility that BEV buyers might not be potential buyers for an ICEV in the same size class, we compared each BEV with ICEVs in adjacent size classes (see Supplementary Table 8). Moreover, to

present the impact of varying ICEV benchmarks on GBET estimations, we used the average level of ICEVs as references and considered the pessimistic and optimistic situations by comparing the BEVs to the most and least efficient ICEVs in the uncertainty analysis. More specifically, if the studied BEV is an A0-class SUV, we used the average emission level and the top and bottom 25% emissions level of fuel-powered A0-class SUVs as benchmarks in the comparison (see Supplementary Fig. 4). Compiling these scenarios facilitates a more comprehensive understanding of the GBET estimations.

## Data availability
The source data for Figs. 1 and 2 are provided with this paper as a Source Data file (https://doi.org/10.6084/m9.figshare.22437775). Additional data used in the analyses are provided in the Supplementary Information. The technical specifications of vehicle models are publicly available from the Automobile Announcement Inquiry website (http://chinacar.com.cn/search.html). The China Automotive Life Cycle Database (CALCD) and China Automotive Life Cycle Assessment Model (CALCM) are available from the China Automotive Technology & Research Center (CATARC) upon request (http://www.catarc.info/). The vehicle-level sales data from China's Compulsory Traffic Accident Liability Insurance (CTALI) are confidential. Owing to the restriction in the licensing agreement, the authors have no right to disclose the original dataset publicly. Source data are provided with this paper.

## Code availability
The code used for estimating greenhouse gas break-even time is freely available at Figshare (https://doi.org/10.6084/m9.figshare.22491034).

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

## Acknowledgements

This research was financially supported by the National Natural Science Foundation of China (No. 71904201 to X.Z., 72174206 to X.T.), Young Elite Scientist Sponsorship Program by Beijing Association for Science and Technology (No. BYESS2023461 to X.Z.), the Chinese Academy of Engineering (No. 2023-XBZD-05 and No. 2022-XY-83 to X.T. and X.Z., 2022-XZ-35 to X.T.), and the Science Foundation of China University of Petroleum, Beijing (No. 2462022YXZZ005 to X.Z.).

## Author contributions

X.Z. and X.S. conceived the original idea. X.Z., Y.R., and X.S. designed the research. X.S., Y.R., Y.K., and D.Z. prepared the data. Y.R. ran the simulation with the help of X.S. Y.R. drew the figures. X.Z. and Y.R. wrote the manuscript with the contributions from P.W., S.Z., X.S., and X.T. X.Z., Y.R., X.S., P.W., S.Z., X.T., Y.K., and D.Z. discussed the findings and commented on the manuscript. X.Z. supervised the research.

## Competing interests

The authors declare no competing interests.
