## [Peer Review File · Nature Communications]

REVIEWER COMMENTS

Reviewer #1 (Remarks to the Author):

The authors have performed a spatially explicit LCA study of BEVs in China with a focus on the temporal distribution of emissions, namely that production emissions occur in year 0. The use of greenhouse gas break even time (GBET) is used to indicate when a BEV begins climate mitigation effects over its ICEV alternative. The novelty in this study lies in the assessment of the temporal aspect of emissions, namely when production emissions occur rather than spreading these over the lifetime of the vehicle. The depth of empirical data used is also of novelty.

General comments:

While I think the authors raise an important point regarding the temporal basis of the emissions (the GBET is an important factor to consider in the assessment of BEVs), I think there is some potential in the evaluation and discussion of the results that remains unexplored in the current form of the manuscript. Furthermore, I have concerns regarding the lack of transparency regarding the methods and data used in this study, and therefore its reproducibility.

- Description of dataset: it is unclear whether all data comes from the dataset, or if assumptions are made. At a minimum, additional description and perhaps labelling of Figure S1 with data sources is necessary.
- Methods: some details regarding the analysis are missing, which make it difficult to understand exactly how the comparison between BEVs and ICEVs was made – this may be clarified by some careful rewording and generally adding more to the existing methods section. More detailed comments are below.
- References: the use of references in the introduction, methods and discussion are somewhat superficial and would be strengthened by the use of multiple references for some statements. Some specific suggestions are made below.
- Statistical analysis: this is also noted for specific lines in the manuscript below.
- Sensitivity analysis: I find myself wondering if a multivariate regression would be more useful? It seems this would be more robust analysis taking advantage of the large dataset the authors have access to, but I admit such analyses are somewhat outside my experience.
- Since insufficient data is provided for reproducing the study, benchmarking to existing literature becomes even more important. While I understand that the data come from a confidential database and therefore cannot be published, I think this makes it even more important to benchmark the results against literature or existing database values. How, then, do the numbers in this study compare to previous findings (e.g., are your production emissions on par with other literature? Are the breakeven points/payback periods similar)?
- Given the authors' argument that that climate mitigation effects of BEVs are delayed, there should be a quantification of the difference between this study and other studies that disregard the temporal emissions distributions. In other words: how "off" are calculations that use a lifetime emissions intensity against the temporal distribution here?

- There seems to be large variation in a lot of the vehicle and fuel chain parameters investigated across the country; I think it would strengthen the results if the authors can perform some simple statistical analyses showing the distribution of parameter values across the data. For example, is the VKT across the country normally distributed between 678 and 15927 km? What about the links between vehicle size and battery capacity, or fuel efficiency (both BEV and ICEV)? Some visualizations or tables with this information will help readers understand the overarching results better, I think. There is some mention of parameter value ranges in the SI, but some additional details and perhaps presenting it in a more accessible manner than a block of text would be helpful.

While the writing is generally good, the article and particularly the SI would benefit from a spellcheck and readthrough by 3rd party for grammar and English phrasing.

Specific comments by line number:

35: "EVs": clarify which vehicle classes (light duty, heavy duty) are included in this number. The authors should also explicitly state in the text which vehicle classes they are including in the study.

39: "From the life cycle perspective, the mitigation benefits of EVs relative to internal combustion engine vehicles (ICEVs) are reported in the range from 18% to 43% of GHGs ": There have been many, many LCA studies for this – one study is not sufficient for this statement.

60: Please define what is included in the term "new energy vehicles" and perhaps what share of these are BEVs

76-77: This point can be made clearer by rewording. I understand this as: newly purchased, single-occupancy BEVs replace public transport activities rather than single- or low-occupancy ICEVs, then the carbon intensity of those passenger-kms will increase?

82-86: What is the source of dataset? How are the values collected (government mandate?) I assume that the dataset is not publicly available and/or the authors are unable to provide it (but am happy to be wrong!) – but is there somewhere the authors can reference to such as a website where readers can learn more?

89: The phrase "one by one" is unclear. Is this at the individual vehicle level? Or is this performed by vintage and vehicle size? I have made further comments on this in the methods section.

103: For the vehicles with GBET= 0. Please explain how this occurs, what types of vehicles this occurs in (what are their characteristics) and how robust these findings are?

106: battery lifetime values: are these empirically determined? 3-4 years seems quite short; are there any explanations for these values?

120-122: Is engine power vs weight a possible explanation for the SUV values? Have you investigated this? Perhaps SUVs generally have more powerful engines, making a lower GBET possible? I wonder if some statistics would help identify potential explanations here.

125-126: It would be helpful to qualify this statement some more: do SUVs and MPVs have e.g., larger batteries or more powerful engines as cars in the same size class?

Table 1: statistical info, such as standard deviation or confidence intervals would be interesting here (or in the SI)

149: How is the CO₂e of the electricity mix calculated? (annual average?) how would charging timing assumptions affect conclusions? Any insight as to when most Chinese BEV owners charge their vehicles?

155: VKT: this is annual vkm travelled, and not lifetime vkm travelled, correct? Please state explicitly.

Figure 3b: Is this the average vehicle weight of both powertrain types? Please clarify.

204-216: The authors point to the importance of further shortening BEV GBET and provide a sensitivity analysis of certain key parameters on the GBET. However, is a 1% improvement in each of these parameters equally likely or “easy” to achieve? The authors mention in the caption for Figure 6 that GHG emissions intensity of the grid has a larger “improvement space,” but has the lowest effect on GBET per Figure 5. How do these counter-acting patterns balance out? Some comment on this by e.g., drawing from historical/projected trends in these parameters may help the relevance.

204-216: Can the authors provide further insight in the text with regards to why the relative ranking of the six factors is what it is? Is it because the GBET “front loads” the production emissions and therefore these hold a heavier weight than with average lifecycle emissions intensities that spread production emissions over the entire lifetime?

207-209: What about vehicle lifetime? While I understand that the total lifetime vkm travelled per vehicle does not affect GBET, it is generally the argument that batteries with larger capacities will have longer lifetimes in terms of vkm travelled, which arguably compensates for their higher production emissions. This could be something worth acknowledging – that while GBET is indeed important, there are other factors usually accounted for in other LCA studies that are not captured here (and vice versa) that are important to consider in assessing the climate mitigation effects of vehicle electrification

Figure 6: I think rather a sequential rather than a diverging colour scheme would be more appropriate here. This colour scheme also seems counterintuitive, as green generally implies “good” while red implies “bad” and yellow “neutral”.

Figure 6: Furthermore, I think this Figure should be expanded to include all parameter pairings; for example, an improvement in VKT does not preclude and decrease in the power grid intensity.

243-252: While older BEV vintages indeed generally did not provide climate mitigation effects, these vehicles are likely retired or approaching retirement (another reason to make these data available if possible!); it seems that most BEVs today are expected to provide mitigation effects over ICEVs. What, then, is the policy implication of this? Perhaps tie this back to the policies mentioned in the Introduction. The push for EVs both in China and globally is seemingly inevitable: these production emissions will happen regardless. How can the findings of this paper be better put to use “in the real world”? How does this way of accounting for emissions affect, e.g., China’s climate goals? The benchmarking activity discussed in other comments will help with this.

272: Please supplement this with other references, such as:

- Luk, Jason M., et al. "Review of the fuel saving, life cycle GHG emission, and ownership cost impacts of lightweighting vehicles with different powertrains." *Environmental science & technology* 51.15 (2017): 8215-8228.
- Raugei, Marco, et al. "A coherent life cycle assessment of a range of lightweighting strategies for compact vehicles." *Journal of Cleaner Production* 108 (2015): 1168-1176.
- Burd, Joshua Thomas Jameson, et al. "Improvements in electric vehicle battery technology influence vehicle lightweighting and material substitution decisions." *Applied Energy* 283 (2021): 116269.
- Das, Sujit, et al. "Vehicle lightweighting energy use impacts in US light-duty vehicle fleet." *Sustainable materials and technologies* 8 (2016): 5-13.

292-305: What about intensifying use, e.g., prioritizing BEVs for car sharing programs or taxis? This would lead to a rapid shifting of stock and take better advantage of (presumably) rapidly decarbonizing electricity mixes in production (which would also shorten the GBET).

308-309 There are certainly many more studies that have come to the same conclusion! I think this statement warrants multiple references.

312-314 Please add a sentence qualitatively assessing the grid carbon intensity of the regions where existing production lines are – are they relatively high carbon or low?

Methods:

The methods section is clear with regards to how the calculations are performed. However, it is difficult for the reader to discern the data sources and assumptions made for many aspects of the work. This obscurity, combined with the lack of benchmarking against other studies as previously mentioned, unfortunately makes the entire study a bit of a 'black box'.

The matching of BEV and ICEV models seems to be key to the outcomes of the study. The authors write that they compare "each of the BEVs produced and sold during 2012-2018" to "ICEVs in the same vehicle model and size class."

- Is the dataset used literally a database of each individual vehicle in the Chinese fleet? Are the comparisons made on an individual car basis, or just by vintage and size class?
- What properties are used to determine the comparative ICEV (do all BEV models in China have a corresponding ICEV model as well)?
- Perhaps the authors can make a comment as to how good these matches of BEVs and ICEVs are and how these assumptions affect results. Using a parameter such as curb weight without any adjustment would be biased as BEVs would likely weigh more than an ICEV with a similar physical footprint. Do the BEVs and ICEVs that are compared have different VKT (the description in lines 365-366 is ambiguous as to this).

In general, please be very clear and explicit as to how this is done (the term “one by one” is also somewhat open to interpretation).

Given the regionalized grid GHG emission factors presented in table S5: are vehicle production emissions also regionalized? If so, please describe where is this information from (or what assumptions are made). On a similar note, please also document the geographical assumptions for the LCA factors presented in Table S3 in the methods or the SI.

What assumptions/data are used for the use phase energy intensity (fuel intensity)? Are these values reported (i.e., empirical), or standard driving cycles, or something else? Are they specific to each vehicle/model? Can these values be published?

403: Please add citations for this statement.

404-405: given the range of VKT given in these lines, it is difficult to conclude that the annual mileage in China is, in fact, much less than 15 000 km (for example, if Tibet were an outlier). Can the authors provide more information regarding the VKT dataset?

427: is there a reference (website or otherwise) for this data source? Is trade between provinces and neighbouring countries included and if so, how is this considered in the calculations?

432-439: Can the authors comment as to how applicable the global median electricity generation factors are to China? Are there any China-specific factors that can be used instead or as part of a sensitivity analysis?

432-439: Can the authors comment as to the effect of seasonal and diurnal effects on the electricity mix?

Supplementary Information:

Figure S2 warrant a full page, I think. It may also be worth considering a different way of presenting the information as in its current state, it is somewhat difficult to interpret. Changing the caption title and titling the colour legend might help – do the colours represent the change in GBET (as implied by the caption), or the absolute GBET? For propriety, it would be interesting to see the figures for the corner comparisons as well.

Can the authors comment on the current state of where vehicles are produced (i.e., in the main analysis)? How was it determined where regions get their BEVs from in Figure S3? The additional analysis in the SI regarding the effect of regional seems valuable, but is rather confusing. What is meant by “vehicle percentage”? This is very unclear from the text, caption and the corresponding results – it seems that each column should sum to 100%, but Guizhou, for example, seems to have 100% imports from Beijing, but also 50% imports (yellow) from most of the other regions.

This additional analysis also warrants some further discussion, particularly in where the data come from and how the results are produced: were the authors able to substitute the energy use in vehicle and battery production with the specific energy mix from each province, or is the lifecycle inventory

database regionalized, and this is a representation of that? Perhaps the authors can also highlight the provinces with the largest production volumes.

Minor comments:

81 recommendation: use alternative term for "one-by-one" (is this literally car by car?)

137 "hatchures" : -

170 Fix: kwh  kWh

172 Change ~ to -

174-176 Please expand on what "other factors" are/could be

Fig 3a use of the terms "top" and "bottom" regions is confusing

189: 38 kg CO₂e/kWh should be g CO₂e/kWh. This is also a problem in Table S5 – please check the manuscript and SI thoroughly for other occurrences.

Figure 4: kg CO₂e: are these values for the entire vehicle lifetime? Per year? Please specify. What is the assumed lifetime?

204: "GHG emission factors" please specify – are these production emission factors?

222: I would argue that the VKT are part of the vehicle cycle, and not the fuel cycle.

297: "charging infrastructure availability", not "charging infrastructure's availability"

408: NDANEV is not defined anywhere – what does it stand for and is there a reference?

437: sentence fragment

There does not seem to be a Figure 1.

The monochromatic colourmaps in Figures 2, 3 and 4 can be challenging to distinguish – perhaps consider at least a two colour theme or using hatching patterns.

Figure 6: I would argue that the VKT is a parameter describing the vehicle cycle (i.e., is a property of the vehicle, not the fuel chain)

Do you have any data regarding the distribution of vehicle lifetimes in China, especially split by powertrain type? It would be interesting to see how many BEVs exit the fleet before their GBET if this data is available.

Reviewer #2 (Remarks to the Author):

1. "The White House stated that half of all new cars sold in the United States in 2030 should be EVs"

The White House announcement is just a target, not a mandatory one.

2. “ From 2012 to 2013, only 15% of the BEVs could pay back the carbon debt within the battery lifetime (the real lifetime of China’s BEV battery in this period is approximately 3-4 years). In 2014-2015 and 2016-2018, this number increased to 80% and 92%, respectively, as the batteries’ real lifetime extended to around five years and ...”

I see only one reference for the battery lifetime. I am curious whether they are any other references that show similar results. The number, 3-4 years seem low. Also, what is considered the end of a lifetime? Is the battery capacity below a certain threshold?

3. Besides the contributing factors that the author indicated in the paper, the fuel economy of the comparable ICEV is also very important factor that changes the results.

Have you also considered a sensitivity study of comparing with a different size of ICEV? The EV buyers are probably not potential buyers for an average ICEV in the same size class. As the authors also indicated that the GBET of BEVs is highly dependent on the reference of ICEVs for comparison.

4. I see “GHG emission factors” and “GHG emissions intensity” are used interchangeably in the paper. Are they the same or different?

5. Figure 4, please add the right unit, kg CO₂e per kwh. Also, Why do you use different scales of the unit in the figure and bar chart (kg vs g)?

6. Page 3: “...early stage of development19...” please delete “19”

7. Table 1: what are the numbers in the parenthesis? I assume they are the number of samples. Better to state it clearly in the table title.

8. Figure 2: Add analysis year to the figure title. “....in 2018”

9. “that more than 80% of BEVs produced and sold since 2014 can bring positive climate benefits within their battery lifetime.”

This seems an important conclusion. I don’t see any figure or table indicating the results. Please advise.

Also, was a different battery lifetime considered for different vehicle model years? 3-4 years for MY2012-2013 vs. 8 years for MY2014 and beyond? Please be clear.

10. For “... as lightweight materials substitution may increase the emissions during BEV’s life cycle..” Please also consider the following reference.

11. The language seems off in some places. Please do another editor review.

Reviewer #3 (Remarks to the Author):

First of all, I would like to thank the Editorial Board for considering me as a reviewer for the manuscript. The manuscript is well written and provides a valuable discussion on battery electric vehicle (BEV) deployment. I believe it provides interesting insights for BEV deployment. However, I believe some aspects are preventing its publication, at its current state, in Nature Communications.

First, the methodology lacks robustness in its definition and assumptions. In addition, several aspects identified by the authors have already been identified in previous studies, e.g., the effective substitution mileage. Therefore, I believe the manuscript does not present innovative aspects in its assessment compared to existing studies. This does not mean the article is irrelevant, but it creates the need to contextualize the relevance of the results compared to other available studies. In light of this, further discussion is necessary to incorporate the implication of the findings in comparison to current results in the scientific literature.

I strongly encourage the authors to consider these changes and resubmit the manuscript. Further comments are presented below:

- Line 18 (abstract): The so-called consensus still does not exist for the entire life cycle of BEV, just to their use phase. If a consensus existed, research on the emissions and impacts of BEV or even the trade-off of using biofuels during the transition to an electrified transportation system would not exist today. Therefore, this sentence is not accurate. However, the same does not apply to the sentence on line 64, as it provides a more accurate definition.

- Check the spacing in Table 1 caption.

- Line 184: From this line, the authors use the terminology 'fuel cycle' to describe the electricity use in BEV. This is not accurate. I understood that the authors are referring to energy to power BEVs, not fuel-related emissions. Please clarify.

- Line 230: Due to the missing units on the axis, it is difficult to understand what the authors want to show with the figure. Is it %?

- Line 235: check Figure 6 caption.

- The sentence in lines 255 to 257 is not valid. Several authors have been assessing other pathways to reduce the climate change impacts of BEV, based on its weight, including the weight of the batteries, among several other strategies. Therefore, this sentence should be revised.
- Line 281 to 284: The authors need to clarify these sentences as it is confusing. I believe they mean that BEVs are exempt from the lottery system. Is that correct? A deeper discussion on the role of policy in adopting the BEV should be included. See, e.g., Zhuge et al., 2020.
- Lines 324-325: Although the authors acknowledged that battery recycling was not included, a better justification for its exclusion should be presented. To date, several articles provide secondary data to provide this assessment. The effects of battery recycling should be better discussed. Besides, the discussion of other strategies for the battery's end of life should be included, e.g., its use in a secondary application. See, e.g., Koroma et al. 2022.
- The sentence on lines 346-348 is inferential and does not add new information to the manuscript.
- Section "Methods", line 358. The authors do not describe the functional unit effectively used to provide the assessment.
- Lines 380-381: The manuscript describes multiple system boundaries. However, it does not disclose the one adopted in the manuscript. When analyzing Figure S1, the system boundaries are disclosed, but they refer to fuel production and usage. Did the authors mean electricity production? The definition of the outputs also does not reflect the reality as other emissions occur during the manufacturing and use of BEV.
- The use of acronyms should be revised, for example, in lines 368 and 369.
- Lines 393-400: As the authors can not disclose the used primary data, more information is necessary to understand the relevance and coverage of the source what the database. This is a major shortcoming in the assessment of data quality. If possible, an assessment of its robustness or a comparison with other available data sources should be conducted to demonstrate its relevance.
- Lines 408-409: Revise the sentence.
- Section 441: Are the influencing factors described in the section beginning in line 203? If yes, please specify. The sensitivity analysis, which is a perturbation analysis (even though the method is not identified by the authors and is referred to as an elasticity index), does not present the reasoning for assessing the selected parameters. Table S2 is missing the units of the current and new parameters.

Additional comments:

- Please revise the use of acronyms in the Supplementary Material (SM). This is an independent document and must present be elaborated accordingly.
- I identified some typos in the SM, such as "emisons".
- What is the legend of Figure S3 implying? Vehicle imports percentage?

- The additional results are never mentioned in the main manuscript. Reference to its content should be provided.

- It is unclear if the authors accessed the end-of-life of the vehicles. Please clarify this aspect.

- At any moment, the manuscript discusses other barriers to BEV deployment. Even though climate change is a great driver of the decisions, other impact categories, such as mineral resources depletion, and the use of critical raw materials, are highly relevant and might delay their production and availability in the market.

REVIEWER COMMENTS

Reviewer #1

The authors have performed a spatially explicit LCA study of BEVs in China with a focus on the temporal distribution of emissions, namely that production emissions occur in year 0. The use of greenhouse gas break even time (GBET) is used to indicate when a BEV begins climate mitigation effects over its ICEV alternative. The novelty in this study lies in the assessment of the temporal aspect of emissions, namely when production emissions occur rather than spreading these over the lifetime of the vehicle. The depth of empirical data used is also of novelty.

We thank the reviewer for the positive evaluation of our work. The reviewer accurately captures the novelty of this article: we used millions of vehicle data in China to evaluate the temporal threshold of the BEVs' mitigation effect, which replaced the traditional practice of amortizing the emissions of electric vehicles over the entire life cycle. This work suggests new GBET-based indicators to guide decision-makers to focus on the adequate substitution of BEVs for ICEVs rather than solely speeding up the electric vehicle deployment.

General comments:

While I think the authors raise an important point regarding the temporal basis of the emissions (the GBET is an important factor to consider in the assessment of BEVs), I think there is some potential in the evaluation and discussion of the results that remains unexplored in the current form of the manuscript. Furthermore, I have concerns regarding the lack of transparency regarding the methods and data used in this study, and therefore its reproducibility.

We truly appreciate that the reviewer agrees with the research significance and spent their precious time helping us improve the manuscript. We have now made substantial revisions to the manuscript following your enlightening suggestions and we feel that the manuscript has been significantly improved. In particular, we 1) added more policy implications and thoroughly discussed the analysis results, 2) rewrote the methodology section, and added more detailed descriptions of the data. Please kindly find our point-to-point responses (in black) below each general and specific comment (in *brown italics*). The line numbers in this response letter correspond to those in the clean version of manuscript.

- Description of dataset: it is unclear whether all data comes from the dataset, or if assumptions are made. At a minimum, additional description and perhaps labelling of Figure S1 with data sources is necessary.

Thanks for the constructive recommendations. We rewrote the data description section and added a table to better describe the data in the revised manuscript. The relevant text now reads, in Lines 450-518:

“Data sources and assumptions

The data used in the estimations of GBET can be classified into four categories according

to their resolution levels (see Table S9). The first category is the real-world vehicle-level dataset, which contains the model type, year, and location of production and sales of almost all BEVs (nearly 1.6 million units) and 82% of ICEVs (145.9 million units) in China from 2012 to 2018 (see Figure S5-S6). The dataset was referenced from China's Compulsory Traffic Accident Liability Insurance (CTALI), which is provided by the China Automotive Technology & Research Center (CATARC)^{61,62}. Since CTALI is compulsory for every vehicle registered in China, the data have wide coverage and high credibility. The vehicle-level data allow us to distinguish the GBET of BEVs across vehicle models, years, and locations.

The second category is vehicle model information. The CTALI database recorded 227 types of BEV models and 1667 types of ICEV models from 2012 to 2018. For each vehicle model, more technical specifications, including the model type, curb weight, battery weight, battery capacity, and fuel consumption were collected from the Announcement of Vehicle Manufacturing Enterprises and Vehicle Products⁶³ which is governed by the Ministry of Industry and Information Technology (MIIT) of China. The fuel consumption for each model type was based on the New European Driving Cycle (NEDC) testing conditions⁶⁴. More statistical descriptions of the data are provided in Figure S7-S9. These technical details were used in the LCA analysis, enabling the estimation of the GHG emissions at the vehicle model level. Moreover, by combining this information with each vehicle's production year and sale location, we can identify the regional heterogeneity of GBET for the same vehicle model using the VKT data and GHG emission factors of power grids that vary across provinces.

The third category of data was reported at the province level, including annual Vehicle Kilometres of Travel (VKT) for both BEVs and ICEVs and emission intensity of power grids. The VKT data for 2018 were extracted from the National Big Data Alliance of New Energy Vehicles (NDANEV)⁶⁵, which records the real-world driving, charging, and fault status of vehicles car by car. According to the requirements of national standard GB/T 32960, the data are uploaded to the platform every 30 seconds when the vehicle is driving, and the fault state is uploaded every second. Between 2018 and July 17, 2022, the NDANEV accessed 9.27 million new energy vehicles with a total VKT of 295.5 billion kilometres. Although the data are real-world and car-by-car, car-level VKT data were not used in the GBET analysis since we had no access to the vehicle identification information to match the NDANEV database with the CTALI database. Thus, we aggregated the data at the provincial level, assuming that the VKT of vehicles within the same provinces is homogeneous. More statistical information on the real-world VKT data is provided in the SI (see Figure SI-S2).

In the estimation of GBET, the vehicle's VKT data by province change annually. Based upon the real-world data from NDANEV in 2018, we projected the VKT before and after 2018 using two methods: one holds conservative attitudes towards the VKT increase, while the other is more radical. Under the conservative estimation, the targeted VKT in 2030 by province contains five levels, i.e., 18,000 km, 15,000 km, 13,000 km, 12,000 km, and 8,000

km, to reflect the regional heterogeneity in new energy vehicle development speed. The VKT in each province before and after 2018 was then projected linearly, assuming that the VKT increases at distinct speeds across provinces (see Table S10). The radical estimation set a more ambitious VKT goal for 2030, reflecting a possible scenario where passenger BEVs and the associated charging infrastructure in China develop at a dramatic speed (see Table S11).

The emission intensity of electricity generation by province was calculated based on the power generation structure and GHG emission factors across power generation types, assuming that the electricity consumption structure is the same as that of electricity generation. Such an assumption might underestimate the emission intensity because marginal electricity consumption for BEVs usually relies on coal and natural gas power plants whose operations are relatively stable with higher GHG emission intensities than the grid structure. The provincial power generation structures from 2012 to 2019 were acquired from the China Electricity Council⁶⁶, and those from 2020 to 2030 were from forecasting data referenced from Li et al. (2021)⁶⁷. The GHG emission factors of different power generation technologies (i.e., coal, wind, solar, nuclear, etc.) were referenced from the IPCC Fifth Assessment Report (AR5)⁶⁸. In the basic estimations, we used the medium value reported by IPCC AR5; this value falls within the range of most existing research on GHG emissions from power generation technologies in China (see more literature in Table S12). In the uncertainty analysis, we employed the maximum and minimum values from existing research. The results of the emission intensity of the power grid by province are presented in Table S13-S15.

The last category of data is the life cycle inventory (LCI) data from the latest China Automotive Life Cycle Database (CALCD)-2021 (see Table S16-S17), developed by the CATARC⁶². These data are homogeneous across provinces. We compared the LCI data from CALCD-2021 with two internationally well-known LCI databases, the Greenhouse gases, Regulated Emissions, and Energy use in Technologies Mode (GREET) and ecoinvent 3.6. We found fairly high consistency among the databases (see Table S17).”

We added a new table (Table S9 in the SI) to describe all the variables used in the estimation and their data sources. Especially for the data which we do not have the rights to publish, more statistical descriptions have been added in the SI (Figure S5-S9) to help the readers better understand what the data look like. For the data which are publicly available, we listed the specific websites. For the LCI data, we compared it with GREET and Ecoinvent 3.6 and found fairly high consistency among the databases, as shown in Table S17. Moreover, we considered the parameter uncertainty in the sensitivity/uncertainty analysis for the robustness check.

Table S9. Variables and data sources used in the estimation.

Category	Resolution	Variables	Time	Description	Range	Data sources	More details
I	Vehicle level (real-world)	Vehicle sold location	Yearly (2012-2018)	The provinces where vehicles are sold	31 provinces ^a	The data of China's Compulsory Traffic Accident Liability Insurance (CTALI), which is provided by the China Automotive Technology & Research Center (CATARC) ^{20,21}	Figure S5-S6.
		Vehicle production location		The provinces where vehicles are produced			
		Battery production location		The provinces where BEV batteries are produced			
		Vehicle model type		Vehicle model types	227 types for BEVs and 1667 types for ICEVs		
II	Vehicle-model level	Curb weight (kg)	Yearly (2012-2018)	The weight of the vehicle, including a full tank of fuel (or battery) and all standard equipment	670-2685 kg for BEVs	Announcement of Vehicle Manufacturing Enterprises and Vehicle Products ²⁴	Figure S7-S9.
		Battery capacity (kWh)		Li-ion battery capacity	645-3200 kg for ICEVs		
		Battery weight (kg)		Battery pack weight	10.56-78.84 kWh		
		Test-based electricity consumption		Official fuel consumption (electricity or fuel)	118-750 kg		

		(kWh/100km)		per 100km based on			
		Test-based fuel consumption (L/100km)		New European Driving Cycle (NEDC) test condition			4.6-21.8 L/100km
III	Provincial level	Vehicle kilometers traveled (km)	Yearly (2012-2030)	Annual mileage traveled of vehicles	678-15927 km/year (2018)	Data for 2018 are obtained from the National Big Data Alliance of New Energy Vehicles (NDANEV) ²⁵ . Data for other years are projected based on assumptions.	Table S10.
		GHG emission factors of electricity generation		Annual GHG emissions per kWh of electricity generation	38-801 gCO ₂ e/kWh (2018)	Data for 2012-2019 are calculated according to the grid structure data from China Electric Power Yearbook 2021 ²⁶ and GHG emission factors of power generation technology from IPCC Fifth Assessment Report (AR5) Annex III Table A.III.2 ²⁷ . Data for 2020-2028 are projected based on assumptions.	Table S13.
IV	Transport-mode level	Life cycle inventories	Constant	Life cycle inventory at different stages of automobile life cycle assessment	N/A	China Automotive Life Cycle Assessment Model (CALCM) ²⁸	Table S16.

Notes: ^a Data for the Hong Kong Special Administrative Region (SAR), Macao SAR, and Taiwan province are unavailable.

- *Methods: some details regarding the analysis are missing, which make it difficult to understand exactly how the comparison between BEVs and ICEVs was made – this may be clarified by some careful rewording and generally adding more to the existing methods section. More detailed comments are below.*

Thanks for pointing this out. Following the reviewer’s insightful suggestions, we substantially improved the methodology section, especially on the description of how the BEVs and the ICEVs were compared. The revised methodology consists of four parts: 1) life cycle assessment of GHGs emissions, 2) calculation of greenhouse gas break-even time, 3) data sources and assumptions, and 4) sensitivity and uncertainty analysis. Specifically, a sub-section “Matching methods between BEVs and ICEVs” and a diagram (Fig. 4) showing the scheme of the GBET estimation process is added to thoroughly explain how BEVs and ICEVs are compared. As the whole section was rewritten, we omit quoting the whole section here and instead presenting part of them in the responses to the specific comments below.

- *References: the use of references in the introduction, methods and discussion are somewhat superficial and would be strengthened by the use of multiple references for some statements. Some specific suggestions are made below.*

Many thanks for the helpful suggestions. We performed another round of literature review and used more relevant references to strengthen the statements. Below please kindly find some examples.

In Lines 42-45: we added more citations and a new table listing selected studies to support the following statement:

“Although assessments vary across studies due to different system boundaries and underlying assumptions, the overall long-term climate benefits of EVs relative to internal combustion engine vehicles (ICEVs) in the context of electricity generation decarbonization dominate the mainstream view²⁻⁵ (see more literature in Table S1).”

In Lines 275-277: more citations are provided for the promotion policies relevant to electric vehicle deployment. The relevant text now reads:

“China and many other countries have leveraged multiple procurement incentives, such as tax credits, discounts or rebates, exemption of EVs from congestion controls, and separate licence plate quotas for electric vehicles³⁰⁻³³”

In Lines 307-308: more references are listed to support the descriptions of mitigation strategy:

“One of the strategies is to reduce the carbon debt by lightweighting^{20,35-39}, material recycling^{40,41}, battery recycling and reuse⁴²⁻⁴⁴.”

Table S1. Selected studies on climate mitigation benefits of EVs relative to ICEVs.

Study	Region	System boundary	Study object	GHG type	Mitigation effect (EV relative to ICEV)
Zhou et al.,2013 ¹	China	Fuel cycle	BEV	CO ₂	35.57%
			PHEV		17.78%
Wu et al.,2018 ²	China	Vehicle+Fuel cycle	BEV	GHG	5.40%
Qiao et al.,2019 ³	China	Vehicle+Fuel cycle	BEV	GHG	18.00%
Zeng et al., 2021 ⁴	China	Vehicle+Fuel cycle	BEV	GHG	23%
			PHEV		17%
			HEV		29%
Wu et al.,2012 ⁵	China	Fuel cycle	PHEV	CO ₂	50%
			BEV		99%
Moro & Lonza,2018 ⁶	EU	Fuel cycle	BEV	GHG	26-47%
Ellingsen et al.,2016 ⁷	EU	Vehicle+Fuel cycle	BEV	GHG	20-27%
Girardi et al.,2015 ⁸	Italy	Vehicle+Fuel cycle	BEV	GHG	50%
Petrauskienė et al., 2020 ⁹	Lithuania	Vehicle+Fuel cycle	BEV	CO ₂	26-47%
Lajunen et al.,2016 ¹⁰	Finland and California	Fuel cycle	BEV	CO ₂	75%
			BEV		50- 80%
Bauer et al.,2015 ¹¹	Global	Vehicle+Fuel cycle	HEV	GHG	16%
			FCEV		3%

Notes: BEV - battery electric vehicles; PHEV - Plug-in Hybrid Electric vehicle; FCEV - fuel cell electric vehicle; HEV - hybrid electric vehicle; ICEV - internal combustion engine vehicle; GHG - greenhouse gas.

In Lines 104-105: we contextualized our study within broader literature and compared the results in the newly added Table S2.

“ As previous GBET studies prepared for the Chinese context are rare, we compared our estimates with those for other countries¹⁹⁻²³, as Table S2 shows.”

Table S2. Selected studies on the GHG emissions break-even points of BEVs.

Study	Region	Scope	Key assumptions			Life cycle inventory (LCI) database	Break-even points
			Emission intensity of electricity mix	Vehicle lifetime	Annual vehicle mileage traveled		
Ellingsen et al., 2016 ⁷	Europe	Four differently-sized EVs	521 g CO ₂ /kWh	12 years	15 000 km	Ecoinvent ¹²	44 000-70 000 km
Kim et al., 2010 ¹³	USA	Light-weighting vehicles using aluminum versus high-strength steel	GHG emission factors of the average U.S. grid	11-16 years	11 000 km	U.S. life cycle inventory (LCI) database (NREL 2007) ¹⁴	1-11 years (differ by light-weighting options)
Patterson et al., 2012 ¹⁵	Europe	Hybrid vehicles and EVs	UK electricity carbon intensity (594 gCO ₂ e/kWh)	10 years	20000 km	Patterson et al., 2011 ¹⁶	1.6-8.3 years (differ by energy scenarios)
International Council on Clean Transportation (ICCT), 2018 ¹⁷	Europe	Battery electric vehicle	European average grid electricity	150000 km	N/A	N/A	2-3 years (differ by carbon intensity of grid electricity)
Ambrose & Kendall, 2016 ¹⁸	USA	Plug-in electric vehicle	Archsmith et al., 2015	N/A	N/A	REET (Argonne National Laboratory, 2014) ¹⁹	Approximately 400–1150 charging cycles
This study	China	All passenger BEVs produced and sold in 2012-2018	38-801 g CO ₂ e/kWh (differ by regional electricity mix)	N/A	678-15927 km (vary by province)	CALCAT ^{20,21}	From zero to over eleven years with an average of 4.5 years

Notes: N/A indicates that the study didn't report such information.

- Statistical analysis: this is also noted for specific lines in the manuscript below.

Many thanks! Following this comment and those in the specific lines, we added more statistical descriptions and correlation analysis in the SI. Below please find some examples.

Figure S2. Sales volume and average VKT of BEVs by province in 2018. Data for the Hong Kong Special Administrative Region (SAR), Macao SAR, and Taiwan province are unavailable.

Figure S6. The number of BEV models per province relative to the provincial market share of national BEV sales, 2012-2018. The horizontal axis is the number of BEV models sold in Mainland China. The vertical axis is the BEVs market share of each province, calculated by dividing the province sales over the national sales. Data for the Hong Kong Special Administrative Region (SAR), Macao SAR, and Taiwan province are unavailable.

Figure S7. Distribution of the vehicle-model specifications for BEVs produced and sold in China from 2012 to 2018. Panel (a) vehicle weight, (b) battery capacity, (c) battery weight, and (d) test-based electricity consumption.

Figure S8. Distribution of the vehicle-model specifications for ICEVs produced and sold in China from 2012 to 2018. Panel (a) vehicle weight, and (b) test-based fuel consumption.

- *Sensitivity analysis: I find myself wondering if a multivariate regression would be more useful? It seems this would be more robust analysis taking advantage of the large dataset the authors have access to, but I admit such analyses are somewhat outside my experience.*

Thanks for the suggestions! As the reviewer pointed out, multivariate regression can analyze the sensitivity of multiple parameters simultaneously, and our large datasets can support the analysis. However, the multivariate regression analysis (MRA) may have a huge bias with the real situation if there are interactions between parameters. In our estimations, some parameters of vehicle information interact with each other. For example, BEVs with higher-capacity batteries usually have heavier vehicle weights for safety concerns. These

interrelationships among parameters imply that MRA is inappropriate for our case's sensitivity analysis. We thus adopt a one-variable-at-a-time approach (OAT) in the sensitivity analysis and consider the combined effects of the parameter uncertainty using an orthogonal experimental design (OED) method.

Moreover, we expanded the parameter ranges considered in the sensitivity analysis to provide a more robust analysis using the OAT approach and rewrote the relevant section. The revised text now reads, in Lines 525-560:

“Parameter uncertainties. We performed a one-variable-at-a-time perturbation sensitivity analysis of the input parameters (Table S6) that influence the GBET. For each variable, sensitivity coefficients (σ_i) were calculated, indicating the percentage change in GBET when the variable changed by 1% (Eq. 4).

$$\sigma_i = \frac{\frac{GBET_i - GBET'_i}{GBET'_i}}{\frac{Inf_i - Inf'_i}{Inf'_i}}$$

where $GBET'_i$ represents the value in the case of the first (baseline) solution; $GBET_i$ represents the value of GBET under the assumed change of variable i ; Inf'_i denotes the initial value of variable i ; and Inf_i represents the changed variable i . Higher sensitivity coefficients denote higher sensitivity of GBET estimation to the changes of variables. Overall, we include 10 influencing factors in our analysis. Since the LCI data are large in volume, we grouped them into four factors for ease of execution: GHG emission factors of vehicle material production, GHG emission factors of battery material production, Electricity consumption during the vehicle production stage, Electricity consumption during the battery production stage (see more details in Table S6). The rest six factors are obtained directly from the databases we use. Based on the sensitivity analysis results, we identified top six sensitive factors in descending order: curb weight, GHG emission factors of vehicle material production, battery capacity, GHG emission factors of battery material production, annual VKT, and GHG emission factors of power grids (shown in Table S7).

We further took these sensitive factors into consideration in the uncertainty analysis. The most common method of uncertainty analysis is the Monte Carlo simulation. However, it is challenging for the GBET estimation as the distribution curve of multiple input parameters, especially those of the LCI data, are elusive. Here, we conduct uncertainty analysis by combining the range approach with orthogonal experimental design (OED). The range approach tests the effects of sampling the parameters at the extreme of their range of variability on the output uncertainty^{69,70} so as to avoid making judgement about the probability of different occurrences⁷¹. We assumed an uniform coverage of the uncertainty input space, i.e., $\pm 5\%$, for these factors: curb weight, battery capacity, GHG emission factors of vehicle material production, and GHG emission factors of battery material production. For GHG emission intensity of the power grids, we used the GHG emission factors of power generation from the IPCC report⁶⁸ in the basic estimation and the low or high values referenced from studies in the Chinese context as the two extreme ends (Table S13-S15). For the annual VKT, we considered a conservative scenario and a radical

development scenario to reflect the variations (Table S10-S11). The OED is an effective method for arranging and analyzing multi-factor interactions. As an alternative to presenting all combination forms of multiple factors, the OED method efficiently schedules multifactorial experiments with optimal combination levels^{72,73}. For the six sensitive factors, we used an orthogonal table (Figure S3) containing 18 representative scenarios to investigate their combined impacts, following the guidelines of ref. Yang et al.⁷⁴.”

- Since insufficient data is provided for reproducing the study, benchmarking to existing literature becomes even more important. While I understand that the data come from a confidential database and therefore cannot be published, I think this makes it even more important to benchmark the results against literature or existing database values. How, then, do the numbers in this study compare to previous findings (e.g., are your production emissions on par with other literature? Are the breakeven points/payback periods similar)?

We appreciate the reviewer’s thoughtful comment and follow their suggestion to compare our results with previous findings. The comparison consists of two parts.

- **Dataset comparison**

First, regarding the life cycle inventories data, we compared the GHG emission factor data we used (CALCD model) with two other widely-used databases, i.e., the Greenhouse gases, Regulated Emissions, and Energy use in Technologies Mode (GREET) and the ecoinvent database, as Table S17 shows. The GREET is the most representative database of vehicle life cycles in the United States, which simulates energy use and emissions for various fuel combinations, sponsored by the U.S. Department of Energy's (DOE) Office of Energy Efficiency and Renewable Energy[1]. The ecoinvent database is a famous Life Cycle Inventory database that supports many environmental assessments of products and processes worldwide[2]. The comparisons, to some extent, confirm the consistency between our data with others, as the magnitude of the data is the same. The differences in the specific values are reasonable, given that the CALCD collects data from China's automotive companies and is probably closer to local realities than the GREET and the ecoinvent.

- **Break-even points comparison**

Second, the break-even points of BEVs were compared with the findings in other studies, as Table S2 shows.

Table S17. GHG emission factors of materials in CALCD, GREET and Ecoinvent (unit:kgCO₂e/kg).

CALCD		GREET-US (GREET 2020)		Ecoinvent 3.6-China		Ecoinvent 3.6-Global	
Variable	Value	Variable	Value	Variable	Value	Variable	Value
Steel	2.38	Average Steel (recycled steel production: 26.4%)	2.56	N/A	N/A	GLO ^c market for steel, unalloyed	1.82
						GLO market for steel, low-alloyed	1.58
						GLO market for steel, chromium steel 18/8	4.39
Cast iron	1.82	Mix (Final Cast Iron: 85%)	1.01	N/A	N/A	GLO market for cast iron	1.71
Aluminum and aluminum alloy	16.38	Virgin Wrought Aluminum Virgin Cast Aluminum	7.98	N/A	N/A	GLO market for aluminum, cast alloy	5.28
			8.87			GLO market for aluminum, wrought alloy	12.7
Magnesium and magnesium alloys	39.55	Virgin Magnesium	36.49	CN ^b magnesium production, pidgeon process	30.3	GLO market for magnesium-alloy, AZ91	26.9
Copper and copper alloys	4.23	Mix: Copper (chilean copper:15.8%)	2.88	N/A	N/A	GLO market for copper	4.65
Plastics average	4.26	Plastics average	2.47	N/A	N/A	Plastics average	4.05
Lithium nickel cobalt manganate	17.4	NMC ^a (622): Coprecipitation	19.78	N/A	N/A	N/A	N/A

CALCD		GREET-US (GREET 2020)		Ecoinvent 3.6-China		Ecoinvent 3.6-Global	
Variable	Value	Variable	Value	Variable	Value	Variable	Value
Graphite	5.48	Graphite	4.81	CN: anode production, graphite, for lithium-ion battery	3.65	GLO market for anode, graphite, for lithium-ion battery	3.66
Electricity	0.635	Distributed - U.S. Mix	0.45	CN market group for electricity, high voltage	1.01	N/A	N/A
				CN market group for electricity, medium voltage	1.05		
				CN market group for electricity, low voltage	1.06		
Gasoline	0.651	E10: Reformulated Gasoline (E10) Blending and Transportation to Refueling Station	0.56	N/A	N/A	N/A	N/A
Diesel	0.644	Low-Sulfur Diesel from Crude Oil	0.44	N/A	N/A	GLO market group for diesel	0.486

Note: ^aNMC: Lithium Nickel Cobalt Manganese Oxide; ^bCN: China; ^cGLO: Global.

Table S2. Selected studies on the GHG emissions break-even points of BEVs.

Study	Region	Scope	Key assumptions			Life cycle inventory (LCI) database	Break-even points
			Emission intensity of electricity mix	Vehicle lifetime	Annual vehicle mileage traveled		
Ellingsen et al., 2016 ⁷	Europe	Four differently-sized EVs	521 g CO ₂ /kWh	12 years	15 000 km	Ecoinvent ¹²	44 000-70 000 km
Kim et al., 2010 ¹³	USA	Light-weighting vehicles using aluminum versus high-strength steel	GHG emission factors of the average U.S. grid	11-16 years	11 000 km	U.S. life cycle inventory (LCI) database (NREL 2007) ¹⁴	1-11 years (differ by light-weighting options)
Patterson et al., 2012 ¹⁵	Europe	Hybrid vehicles and EVs	UK electricity carbon intensity (594 gCO ₂ e/kWh)	10 years	20000 km	Patterson et al., 2011 ¹⁶	1.6-8.3 years (differ by energy scenarios)
International Council on Clean Transportation (ICCT), 2018 ¹⁷	Europe	Battery electric vehicle	European average grid electricity	150000 km	N/A	N/A	2-3 years (differ by carbon intensity of grid electricity)
Ambrose & Kendall, 2016 ¹⁸	USA	Plug-in electric vehicle	Archsmith et al., 2015	N/A	N/A	REET (Argonne National Laboratory, 2014) ¹⁹	Approximately 400–1150 charging cycles
This study	China	All passenger BEVs produced and sold in 2012-2018	38-801 g CO ₂ e/kWh (differ by regional electricity mix)	N/A	678-15927 km (vary by province)	CALCAT ^{20,21}	From zero to over eleven years with an average of 4.5 years

Notes: N/A indicates that the study didn't report such information.

- Given the authors' argument that that climate mitigation effects of BEVs are delayed, there should be a quantification of the difference between this study and other studies that disregard the temporal emissions distributions. In other words: how "off" are calculations that use a lifetime emissions intensity against the temporal distribution here?

Many thanks for the reviewer's insightful comment. The calculation of GBET provides information on how quickly mitigation benefits appear, which is complementary to the total mitigation effect information from the previous life cycle assessment. Due to the differences in system boundaries and underlying assumptions, it is hard to perform a direct and credible quantification of the difference between this study and previous LCA studies. As an alternative, we have added more qualitative analysis on how considering the delayed climate mitigation effects enhances the understanding of BEV climate benefits. The relevant text in the revised manuscript reads, in Lines 295-303:

"It is worth mentioning that a smaller GBET does not necessarily yield higher emission reduction over the life cycle. For example, BEVs with a greater driving range tend to have a longer lifetime, so they can bring more emission reduction benefits over the full life cycle. However, increased driving range often requires greater battery capacity and weight, which increases GHG emissions from battery production and leads to higher GHG debt. Thus, it will take longer to pay off these GHG debts in the GBET. The GBET is a supplementary indicator to the existing metrics, as it provides information on how quickly the climate benefits are generated, while previous LCA assessment suggests the size of the benefits throughout the vehicle's lifetime. Both the information is important to consider in assessing the climate mitigation effects of vehicle electrification."

Moreover, we provided more policy implications for the real world based on the GBET estimation. The relevant text in the revised manuscript reads, in Lines 270-293:

"First, new GBET-based indicators can be developed to guide BEV deployment. For example, the percentage of BEVs that have reached greenhouse gas break-even time (PER-GBET) is an indicator to supplement the widely used indicator EV penetration rate (EPR). In other words, it is not how many EVs are produced and sold but how many have positive emission reductions that contribute to the climate benefits of the transportation sector. Relevant policies targeting the indicators could be modified accordingly. China and many other countries have leveraged multiple procurement incentives, such as tax credits, discounts or rebates, exemption of EVs from congestion controls, and separate licence plate quotas for electric vehicles³⁰⁻³³, to pursue higher EPR. However, once the sale is complete, these policies are no longer in effect, leaving the true climate effect unmanaged³⁴. A direct solution to this problem is to set follow-up policies for BEV deployment, such as performing stage subsidies (i.e., extending the subsidy timeline from the purchase time point to the time when it achieves its GBET), investing in charging infrastructure, and motivating ICEV replacement, to promote higher PER-GBET."

GBET can also inform technical standards for the life expectancy or battery replacement time of BEVs to ensure net climate benefits. For example, China currently requires a

homogeneous battery warranty period of no less than eight years or 120,000 kilometres in mileage (whichever occurs first)²⁶. However, as our estimations in the previous sections show, not all vehicles can achieve their GBET within the battery warranty period, including some large-size or large-mode vehicles (approximately 8% of the total between 2016 and 2018). This situation calls for a longer required warranty time for heavy transport modes. In fact, GBET provides guidelines for differentiated warranty times and other longevity-relevant standards. This process, on the one hand, facilitates climate benefits by avoiding the early replacement of batteries and, on the other hand, motivates BEV suppliers to improve the climate performance of their products.”

- There seems to be large variation in a lot of the vehicle and fuel chain parameters investigated across the country; I think it would strengthen the results if the authors can perform some simple statistical analyses showing the distribution of parameter values across the data. For example, is the VKT across the country normally distributed between 678 and 15927 km? What about the links between vehicle size and battery capacity, or fuel efficiency (both BEV and ICEV)? Some visualizations or tables with this information will help readers understand the overarching results better, I think. There is some mention of parameter value ranges in the SI, but some additional details and perhaps presenting it in a more accessible manner than a block of text would be helpful.

Many thanks for the helpful comment. We have added statistical analyses showing the distribution of parameter values in the SI. Besides those figures that have been present in previous responses (to the comment on statistical analysis), please find more examples below.

- For the distribution of VKT

Figure S1. Average annual mileage traveled of passenger cars in China 2018.

- The distribution curve of vehicle-model specifications and their correlations

Figure S9. Correlations of vehicle-model specifications between BEVs and ICEVs. Panels on the diagonal line show the statistical distribution of each variable. The lower left shows a bivariate scatter plot with fitted lines and correlation coefficients (***) $p < 0.01$, (**) $p < 0.05$, (*) $p < 0.1$).

While the writing is generally good, the article and particularly the SI would benefit from a spellcheck and readthrough by 3rd party for grammar and English phrasing.

Thanks for pointing this out. We have used professional editing services to proofread the manuscript and made extensive changes to improve the clarity. Due to the length of the changes involved, we did not copy them here. Please see the revised manuscript for all the changes.

Specific comments by line number:

35: "EVs": clarify which vehicle classes (light duty, heavy duty) are included in this number. The authors should also explicitly state in the text which vehicle classes they are including in the study.

Thanks for the question. This number in IEA's report includes both light-duty and heavy-duty electric vehicles. We clarified it in the revised manuscript (Lines 40-42):

"According to the International Energy Agency (IEA), electric vehicles (EVs), including both light-duty and heavy-duty vehicles, enabled a net reduction of 40 million tonnes of carbon dioxide-equivalent (CO_{2e}) on a well-to-wheel basis in 2021¹"

In our study, the CALCD database records all light-duty passenger vehicles produced and sold in China from 2012 to 2018. This study compares passenger vehicles powered by the internal combustion engine and battery while not including plug-in hybrid vehicles, hybrid vehicles, and hydrogen fuel cell vehicles. We clarify the study scope in the revision, and the relevant text now reads, in Lines 83-85:

"The data contains almost all the BEVs (nearly 1.6 million) and 82% of the ICEVs (145.9 million) in the light-duty passenger vehicle category produced and sold in China during 2012-2018."

39: "From the life cycle perspective, the mitigation benefits of EVs relative to internal combustion engine vehicles (ICEVs) are reported in the range from 18% to 43% of GHGs ": There have been many, many LCA studies for this – one study is not sufficient for this statement.

Thanks for the valuable comment.. We added more relevant LCA studies and also a new table in the supporting information (SI) to support the statement. The revised text now reads (Lines 42-45):

"Although assessments vary across studies due to different system boundaries and underlying assumptions, the overall long-term climate benefits of EVs relative to internal combustion engine vehicles (ICEVs) in the context of electricity generation decarbonization dominate the mainstream view²⁻⁵ (see more literature in Table S1)."

Table S1. Selected studies on climate mitigation benefits of EVs relative to ICEVs.

Study	Region	System boundary	Study object	GHG type	Mitigation effect (EV relative to ICEV)
Zhou et al.,2013 ¹	China	Fuel cycle	BEV	CO ₂	35.57%
			PHEV		17.78%
Wu et al.,2018 ²	China	Vehicle+Fuel cycle	BEV	GHG	5.40%
Qiao et al.,2019 ³	China	Vehicle+Fuel cycle	BEV	GHG	18.00%
Zeng et al., 2021 ⁴	China	Vehicle+Fuel cycle	BEV	GHG	23%
			PHEV		17%
			HEV		29%
Wu et al.,2012 ⁵	China	Fuel cycle	PHEV	CO ₂	50%
			BEV		99%
Moro & Lonza,2018 ⁶	EU	Fuel cycle	BEV	GHG	26-47%
Ellingsen et al.,2016 ⁷	EU	Vehicle+Fuel cycle	BEV	GHG	20-27%
Girardi et al.,2015 ⁸	Italy	Vehicle+Fuel cycle	BEV	GHG	50%
Petrauskienė et al., 2020 ⁹	Lithuania	Vehicle+Fuel cycle	BEV	CO ₂	26-47%
Lajunen et al.,2016 ¹⁰	Finland and California	Fuel cycle	BEV	CO ₂	75%
			BEV		50- 80%
Bauer et al.,2015 ¹¹	Global	Vehicle+Fuel cycle	HEV	GHG	16%
			FCEV		3%

60: Please define what is included in the term "new energy vehicles" and perhaps what share of these are BEVs

New energy vehicles refer to all vehicles not using gasoline and diesel engines, which includes Battery Electric Vehicles (BEV), Plug-in Hybrid Vehicle (PHEV), Hybrid Electric Vehicle (HEV), Fuel Cell Electric Vehicle (FCEV), etc. By the end of June 2022, the stock of China's new energy vehicles reached 10.1 million, accounting for 3.23% of the total number of vehicles. Among them, the number of BEVs is 8.104 million, accounting for 80.93% of the total new energy vehicles[3].

Thanks to the reviewer's helpful comment, we added this information to the revised manuscript. The relevant text now reads, in Lines 60-66:

“According to the New Energy Automobile Industry Development Plan (2021-2035)⁶ announced by the Chinese government, the targeted penetration rate of new energy vehicles, including BEVs, hybrid electric vehicles (HEVs), and fuel cell electric vehicles (FCEVs), will reach 20% by 2025. According to the Action Plan for Carbon Dioxide Peaking before 2030¹³, the market share of new energy vehicles will reach approximately 40% by 2030. Achieving these ambitious goals put most of the development pressure on BEV sales, as they account for 80% of new energy vehicles¹⁴.”

76-77: This point can be made clearer by rewording. I understand this as: newly purchased, single-occupancy BEVs replace public transport activities rather than single- or low-occupancy ICEVs, then the carbon intensity of those passenger-kms will increase?

Many thanks for pointing out the obscure point. The reviewer has actually reworded the sentence for us, which is exactly what we meant to express. As we have almost rewrote the manuscript, this point has been moved to the discussions and now reads, in Lines 330-341:

“Although our findings have great implications, we also noticed that there are several limitations. First, we assume that a BEV's yearly effective substitution mileage for ICEVs is the annual average VKT in the province it is sold, without considering the rebound effect or spillover effect⁵¹ of BEV uptake on GHG emissions. This assumption might bias the GBET estimation. In the scenario where the first-time car owner purchased a BEV to replace public transportation service rather than ICEVs, the effective substitution mileage is lower than the BEV's annual VKT and results in an underestimation of the GBET. In the other scenario where a BEV is bought to actually replace the already owned ICEV, due to the limitation on driving range, the user might reduce the car usage compared to owning an ICEV. A positive spillover effect occurs, and the effective substitution mileage is higher than the BEV's annual VKT. In fact, to what extent BEVs effectively substitute ICEVs is complicated, as it is relevant to consumers' behaviours³⁴; this relationship has not been fully discussed and leaves ample opportunities for future research.”

82-86: What is the source of dataset? How are the values collected (government mandate?) I assume that the dataset is not publicly available and/or the authors are unable to provide it (but am happy to be wrong!) – but is there somewhere the authors can reference to such as a website where readers can learn more?

Thanks for the questions. The data are collected from multiple sources, as shown in Table S9. For the data which are confidential, we provided more statistical descriptions in the revised materials to promote a better understanding of the dataset. For the data which are publicly available, we listed the specific websites. The relevant descriptions have been added in the new sub-section “**Data sources and assumptions.**” In brief, the data can be classified into four categories:

- Real-world production and sale data at the vehicle level

The first category is the real-world vehicle-level dataset, which contains the model type, production year, and locations of almost all BEVs (nearly 1.6 million units) and 82% of ICEVs (145.9 million units) produced and sold in China from 2012 to 2018 (more statistical descriptions of the data are provided in Figure S5-S6). The dataset is referenced from China's Compulsory Traffic Accident Liability Insurance (CTALI), which is provided by the China Automotive Technology & Research Center (CATARC)[4, 5]. Since CTALI is compulsory for every vehicle registered in China, the data has wide coverage and high credibility. The vehicle-level data allow us to distinguish the GBET of BEVs across vehicle models, years, and locations.

- Technical specifications on vehicle models

The second category is vehicle model specifications. The CTALI database records 227 types of BEV models and 1667 types of ICEV models from 2012 to 2018. For each vehicle model, more technical specifications, including the model type, curb weight, battery weight, battery capacity, and fuel consumption were collected from the Announcement of Vehicle Manufacturing Enterprises and Vehicle Products[5] which is governed by the Ministry of Industry and Information Technology (MIIT) of China. The fuel consumption for each model type was based on the New European Driving Cycle (NEDC) testing conditions[6]. For easier access to the data service, we referenced the website of Automobile Announcement Inquiry (<http://chinacar.com.cn/search.html>) and the website of China automobile fuel consumption inquiry system (<https://yhgscx.miit.gov.cn/fuel-consumption-web/mainPage>) in the revised manuscript.

- Provincial data of VKT and GHG emission factors of electricity grids

The third category of data is reported at the province level, including annual Vehicle Kilometers of Travel (VKT) for both BEVs and ICEVs and emission intensity of power grids. The VKT data in 2018 are empirical and sourced from the National Big Data Alliance of New Energy Vehicles (NDANEV)[7], which records real-world driving, charging, and fault status of vehicles car by car. However, since we have no access to the vehicle identification information to match the NDANEV database with the CTALI database, we aggregated the data at the provincial level, assuming that the VKT of vehicles within the same provinces is homogeneous. Moreover, the vehicle's VKT data by province change annually, and the data before and after 2018 are projected based on national targets and assumptions.

The emission intensity of electricity generation by province is calculated based on the power generation structure and GHG emission factors across power generation types, assuming that the electricity consumption structure is the same as that of electricity generation. Such an assumption might underestimate the emission intensity because marginal electricity consumption for BEVs usually relies on coal and natural gas power plants whose operation is relatively more stable and has higher GHGs emission intensity than the grid structure. The provincial power generation structure from 2012 to 2019 was acquired from China Electricity Council[8], and those from 2020 to 2028 were forecasting data referenced from Li et al. (2021)[9]. The GHGs emission factors of different power generation technology (i.e., coal, wind, solar, nuclear, etc.) are referenced from the IPCC Fifth Assessment Report (AR5)[10].

And we also consider the factor values reported in other studies in parameter uncertainty analysis.

- Life cycle inventory data

The last category of data is the life cycle inventory (LCI) data from the latest China Automotive Life Cycle Database (CALCD)-2021, developed by the CATARC[5]. These data are homogeneous across provinces. We compare the LCI data from CALCD-2021 with two internationally well-known LCI databases, the Greenhouse gases, Regulated Emissions, and Energy use in Technologies Mode (GREET) and the ecoinvent 3.6. We find fairly high consistency among the databases (see Table S17).

Moreover, as quoted in the previous responses (to the comment on the Description of the dataset), we added a new table (Table S9) in the SI to present more details of the data.

89: The phrase “one by one” is unclear. Is this at the individual vehicle level? Or is this performed by vintage and vehicle size? I have made further comments on this in the methods section.

Thanks for the critical questions. Yes, for BEVs, the estimation of GBET is at the individual vehicle level. As a BEV has multiple choices of ICEVs for comparison benchmark, we pair each BEV with their ICEV counterparts by vintage and vehicle size. The revised manuscript clarifies the matching methods, and a new graph (Fig. 4) is added to visualize the estimation process. The relevant text now reads, in Lines 409-422:

“Matching methods between BEVs and ICEVs. Since the GBET of BEVs is calculated at the vehicle level, we find fuel-powered counterparts for each of the BEVs produced and sold from 2012 to 2018. One BEV can have multiple fuel-powered counterparts because, in the real world, consumers’ choices for fuel-powered substitutes of a certain BEV have many possibilities. Considering that in most cases, the replacements happen in the same vehicle class, we compare each of the BEVs with the ICEVs in the same vintage, transport mode (Car, SUV, and MPV), and size class (A00, A0, A, B, and C) (see more details of the vehicle clarification in Table S5) to generate basic estimations, referring to Fig. 4. As the comparison is “one (BEV) to more (ICEVs),” for systematic comparison, we generate an average representative of the selected ICEVs, whose parameters are the average of the matched counterpart ICEVs. Then, the comparison turns to “one (BEV) to one (representative ICEV). The GBET of BEVs within the same stratum are then averaged to generate an overall estimate. In the uncertainty analysis, we consider more possibilities of the substitutes across different classes and more possibilities of the representative ICEVs (see sensitivity and uncertainty analysis for more details).

Fig. 4. Scheme of the GBET estimation process and matching methods between BEVs and ICEVs

The readers are also referred to the sensitivity and uncertainty section for more possibilities of the BEV versus ICEV pairing. The text reads, in Lines 250-260:

“Moreover, the GBET of BEVs is also highly dependent on the comparison benchmark of ICEVs, which varies over a large range. For the robustness check, we paired BEVs with ICEVs across different size classes and used both the most and least efficient ICEVs as benchmarks to present pessimistic and optimistic GBET estimations. Compared between adjacent size classes, the GBET estimations fluctuate from -74% to 156% (Table S8). Compared with the same size class but using the ICEVs whose GHG emissions are in the lower quartile (i.e., top 25% low-emission ICEVs), the GBET increases by 1.9-6.7 years, with an average increase of 3.9 years. In this case, nearly half of BEVs sold in 2018 cannot repay GHG debt within 11 years. When we change the benchmark to the ICEVs whose GHG emissions are in the higher quartile (i.e., top 25% high-emission ICEVs), the GBET decreases by 1.6-5.1 years, with an average decrease of 2.9 years (Figure S4). In this case,

all BEVs sold in 2018 achieve a GBET within 7 years and 95% of them actually within 3 years.”

103: For the vehicles with GBET= 0. Please explain how this occurs, what types of vehicles this occurs in (what are their characteristics) and how robust these findings are?

GHET = 0 means that the BEVs emit less GHGs emissions than their ICEV counterparts in the production phase. The BEVs whose GBET is zero accounts for 1.7% of the total sample, most of which are A00-class. This is probably explained by their their low GHG emissions in the production phase, as the battery capacity and vehicle weight of A00-class cars are significantly lower than other size classes (Figure S7).

The robustness of this finding is further confirmed in the uncertainty analysis. Despite the perturbation of the parameters, the percentage of vehicles whose GBET is zero varies from 0% to 4.2%, and almost all of them are A00-class cars.

Figure S7. Distribution of the vehicle-model specifications for BEVs produced and sold in China from 2012 to 2018. Panel (a) vehicle weight, (b) battery capacity, (c) battery weight, and (d) test-based electricity consumption.

106: battery lifetime values: are these empirically determined? 3-4 years seems quite short; are there any explanations for these values?

Thanks for the question. The battery lifetime values are cited from a Chinese paper (Li, Z. & Li, Y, 2018. Forecast of the decommissioning capacity of a new energy vehicle battery in China. Resource recycling, 34-36, in Chinese), and according to the authors, they are

determined based on empirical surveys in China. We quote more information from the paper below.

“The first phase of demonstration and promotion of new energy vehicles in China started in 2009. In the first few years, the power batteries that were put into use were short-lived and few in number. They were not very mature in technology, and there were no uniform regulations on quality assurance requirements. The battery life was short, generally only 2-3 years. After 2011, with the rapid development of the new energy vehicle industry, the power battery technology has been developed to a certain extent, and the lifespan has been extended to 3-4 years. In 2014, the Ministry of Finance, the State Administration of Taxation and the Ministry of Industry and Information Technology jointly issued the "Announcement on Exemption from the Purchase Tax of New Energy Vehicles", which set the warranty for new energy vehicle power batteries to be no less than 5 years or 10,000 kilometers (whichever comes first), so the life of the power battery since then can reach about 5 years. In 2015, the Ministry of Finance, the Ministry of Science and Technology, the Ministry of Industry and Information Technology, and the National Development and Reform Commission jointly issued the "Notice on Financial Support Policies for the Promotion and Application of New Energy Vehicles in 2016-2020" on the battery shelf life of new energy passenger vehicle companies. The warranty period is not less than 8 years or 120,000 kilometers (whichever comes first).

--Translated from Li & Li's work”

Table R1.1. The distribution of China passenger BEVs' battery lifespan

Year	Average service life (years)	Distribution (%)						
		P2	P3	P4	P5	P6	P7	P8
2012	3	0.1	0.8	0.1				
2013	3.4		0.6	0.4				
2014	3.8		0.2	0.8				
2015	5			0.3	0.4	0.3		
2016	5.9				0.3	0.5	0.2	
2017	7.1					0.2	0.5	0.3
2018	7.6						0.4	0.6

Notes: The table is translated from Li & Li (2018)'s work. P_i is the percentage of all batteries whose battery service lifetime is i years in that year.

However, as there are few empirical data from other studies to cross-check its reliability, we removed this reference and relevant statements. As an alternative, we referenced the vehicle battery warranty periods required officially in China and compared the GBET estimates in our study with the battery warranties. Since battery manufacturing accounts most for the GHG debt of BEVs and the lifetime of batteries is usually shorter than those of vehicles, identifying BEVs that could achieve GBET in the battery warranties (Table S4) provides a supplementary

indicator for climate benefits assessment from the temporal perspective. The relevant text reads, in Lines 144-148 (results):

“Moreover, approximately one-fifth of the BEVs produced and sold before 2016 failed to pay back the GHG debt within five years, which is the EV battery warranty time required by the Chinese government in 2014²⁵. In 2016, the required battery warranty was extended to eight years²⁶, and 8% of BEVs produced and sold between 2016 and 2018 have failed to achieve the GBET within the battery warranty threshold (see more supplementary results in Table S4).”

Table S4. The percentage of BEVs achieving GBET within the battery warranty period.

Transport mode	Year	2012	2013	2014	2015	2016	2017	2018
	Threshold	Five-year battery warranty				Eight-year battery warranty		
Car	A00	100%	100%	98%	89%	95%	97%	99%
	A0	98%	99%	97%	93%	100%	100%	99%
	A	78%	98%	69%	38%	72%	89%	86%
	B	-	-	0%	0%	70%	61%	66%
SUV	A0	100%	100%	100%	-	99%	94%	96%
	A	-	-	100%	90%	99%	100%	100%
	B	-	-	-	-	100%	100%	100%
	C	-	-	-	-	-	-	100%
MPV	A0	29%	10%	6%	1%	1%	42%	36%
	A	-	-	-	-	-	82%	96%
	B	-	-	-	-	41%	93%	74%

Notes: Cell values in the table denote the percentage of BEVs produced and sold in that year achieving GBET within the battery warranty period, i.e., five years required in 2014²² and eight years required in 2016²³. Since there is no official requirement before 2014, we assume five years (2014 requirement) for 2012 and 2013. A blank cell means there is no corresponding vehicle data.

120-122: Is engine power vs weight a possible explanation for the SUV values? Have you investigated this? Perhaps SUVs generally have more powerful engines, making a lower GBET possible? I wonder if some statistics would help identify potential explanations here.

Thanks for the enlightening comment, which provides great ideas for our analysis. We performed a descriptive statistical analysis of the relevant data and found that the statistics

support the reviewer’s conjecture. In general, heavier vehicle weight is inversely associated with fuel economy, with an elasticity of about 0.75, according to our dataset (Figure S9). More specifically, as Table S3 shows, SUVs are generally heavier than Cars, and they are less fuel-efficient because of more powerful engines. These two factors exert opposite-direction influences on the GBET: heavier weight leads to more GHG emission debt of BEVs in the production phase and thus longer GBET, while powerful engines lead to more GHG emissions of ICEVs and relatively more mitigation benefits of BEVs in the fuel cycle, leading to a decreasing trend in GBET. The effect of these two trends determines the GBET simultaneously.

Figure S9. Correlations of vehicle-model specifications between BEVs and ICEVs. Panels on the diagonal line show the statistical distribution of each variable. The lower left shows a bivariate scatter plot with fitted lines and correlation coefficients (***) $p < 0.01$, (**) $p < 0.05$, (*) $p < 0.1$).

Table S3. Comparison of curb weight and test-based fuel consumption between BEVs and ICEVs.

Vehicle model	Size class	Average curb weight		Average test-based fuel consumption		Average battery capacity
		ICEV (kg)	BEV (kg)	ICEV (L/100km)	BEV (kWh/100km)	BEV (kWh)
Car	A00	933	974	6.0	13	22
	A0	1090	1267	7.2	15	24
	A	1277	1581	7.1	15	40
	B	1537	1937	7.8	16	55
SUV	A0	1287	1512	7.0	15	45
	A	1576	1787	8.5	17	47
	B	1829	1788	9.8	15	52
	C	2132	2460	13.1	21	67
MPV	A0	1307	1918	7.2	18	51
	A	1555	1878	7.8	16	55
	B	1909	2302	9.7	23	55

Inspired by this comment, we rewrote the relevant paragraph as follows (in Lines 150-179)

“The GBET of BEVs also showed significant heterogeneity among various transport modes (Car, SUV, and MPV) and size classes (A00, A0, A, B, and C) (see more details of the vehicle classification in Table S5). The impact of influencing factors is bidirectional as well. On the one hand, heavier transport modes and larger vehicle sizes usually have larger battery capacity and heavier battery weight, resulting in higher GHG emissions in the production phase and, thus, more GHG debt (see Table S3). This trend potentially increased GBET. On the other hand, the fuel-powered counterparts of heavier transport modes and larger vehicle sizes are energy intensive (see Table S3) and emit more GHGs during the fuel cycle, resulting in more notable emission reduction benefits of BEVs relative to ICEVs and faster GHG debt pay-off periods. This trend potentially decreased GBET. Under the combined effect of these two trends, the GBET of BEVs showed an overall increasing trend with larger sizes (A00<A0<A<B) and larger transport modes (Car<SUV<MPV).

More specifically, the effects of transport mode and size class interact. The impact of transport mode varies across size classes. For A0-class vehicles, the GBET increases in the order of Car, SUV, and MPV. This ordering implies that the increasing impacts of more GHG debt caused by the heavier weight of larger transport modes exceed the decreasing impacts caused by improving the debt repayment efficiency during the fuel cycle (the terminology fuel is used conventionally, referring to electricity production, transmission, and use for BEVs). For A-class and B-class vehicles, the GBET of the car is the largest (6.3-7.3 years), the SUV is the smallest (3.1-4.8 years), and the MPV is in the middle (5.8-6.1 years). This indicates that under these two size classes, the positive effect from the

increase in the rate of debt repayment in the fuel cycle of SUVs and MPVs completely offsets the negative effect due to the increase in curb weight. Similarly, the effect of size class on GBET is related to the transport mode. For Cars, GBET shows an increasing trend with the larger size. The GBET prolongation effect caused by the increase in GHG debt with the increase in size class exceeds the reduction effect caused by the increase in fuel-cycle emission reduction. SUVs and MPVs showed the opposite trend: GBET decreased with increasing size class. In this case, the relative advantages of BEVs in fuel-cycle emission reduction brought by the increase in size class are more dominant. Thus, overall, SUVs and MPVs with larger size classes and cars with smaller size classes have shorter GBETs.”

125-126: It would be helpful to qualify this statement some more: do SUVs and MPVs have e.g., larger batteries or more powerful engines as cars in the same size class?

Yes, SUV and MPV have larger batteries and more powerful engines than car in the same size class, as Table S3 shows. As quoted in the response above, we used this quantitative information to explain the GBET heterogeneity across size classes and mode types in the revised manuscript.

Table 1: statistical info, such as standard deviation or confidence intervals would be interesting here (or in the SI)

We thank the reviewer for the helpful suggestion. We reproduced Table 1 to include the standard deviation of the GBET in the brackets and also indicated the confidence levels by asterisks.

Table 1. The descriptive statistics of greenhouse gas break-even time of BEVs in the Chinese market by transport mode and size class

Transport mode	Size class	2012	2013	2014	2015	2016	2017	2018
Car	A00	2.2**	3.1**	2.3***	2.5***	3.5***	3.3***	3.2***
		(0.8)	(1.1)	(1.7)	(2.0)	(2.3)	(1.9)	(1.7)
	A0	4.2**	4.1**	4.2**	4.2***	3.8***	3.3***	3.8**
		(0.2)	(0.2)	(0.3)	(0.5)	(0.5)	(0.6)	(1.3)
	A	5.0*	6.2*	4.9*	7.1**	7.1***	6.4***	6.3***
		(0.5)	(0.3)	(0.8)	(2.6)	(1.3)	(1.6)	(1.6)
	B	-	-	8.1*	9.8**	7.8**	7.5**	7.3**
				(0.6)	(1.2)	(0.6)	(1.7)	(1.7)
SUV	A0	3.7*	3.7*	3.6*	-	5.1**	5.7**	5.2***
		(0.1)	(0.1)	(0.1)		(1.1)	(1.2)	(1.2)
	A	-	-	5.0*	3.2*	5.6**	5.0***	4.3***
				(0.1)	(0.9)	(1.2)	(0.7)	(1.0)
	B	-	-	-	-	3.8*	4.1*	3.1**
						(0.1)	(0.2)	(0.3)
	C	-	-	-	-	-	-	4.8***
								(0.7)
MPV	A0	8.6**	9.4**	10.5**	10.9**	10.9***	7.8***	8.4***
		(3.3)	(2.2)	(1.7)	(0.7)	(0.6)	(3.6)	(3.1)
	A	-	-	-	-		7.2*	6.1**
							(1.1)	(0.6)
	B	-	-	-	-	9.0*	5.2**	5.8*
						(1.4)	(1.6)	(2.3)

Notes: SUV = sports utility vehicle; MPV = multipurpose vehicle. A short string indicates insufficient data for the transport mode in the given year. The values in parentheses represent the standard deviations. Low, medium, and high confidence levels correspond to the sample size <1000, [1000,10000], and >10000 represented, denoted by *, **, and ***. The darker shade of red indicates longer GBETs.

149: How is the CO₂e of the electricity mix calculated? (annual average?) how would charging timing assumptions affect conclusions? Any insight as to when most Chinese BEV owners charge their vehicles?

Thanks for the insightful questions. Yes, regarding the calculation of the GHG emissions of the electricity mix, we use the annual average data. The calculation method, as well as the key assumptions, is restated in the revision. The relevant text reads, in Lines 498-511:

“The emission intensity of electricity generation by province was calculated based on the power generation structure and GHG emission factors across power generation types, assuming that the electricity consumption structure is the same as that of electricity generation. Such an assumption might underestimate the emission intensity because marginal electricity consumption for BEVs usually relies on coal and natural gas power plants whose operations are relatively stable with higher GHG emission intensities than the grid structure. The provincial power generation structures from 2012 to 2019 were acquired from the China Electricity Council⁶⁶, and those from 2020 to 2028 were from forecasting data referenced from Li et al. (2021)⁶⁷. The GHG emission factors of different power generation technologies (i.e., coal, wind, solar, nuclear, etc.) were referenced from the IPCC Fifth Assessment Report (AR5)⁶⁸. In the basic estimations, we used the medium value reported by IPCC AR5; this value falls within the range of most existing research on GHG emissions from power generation technologies in China (see more literature in Table S12). In the uncertainty analysis, we employed the maximum and minimum values from existing research. The results of the emission intensity of the power grid by province are presented in Table S13-S15.”

Inspired by the reviewer's second and third questions, we acknowledged the limitation and discussed the potential impact of using annual average data on GBET estimates, considering the charging time characteristics of Chinese EV owners. According to the Application Analysis of Charging Data for New Energy Vehicles in China[11] published by the State Grid Corporation of China, the peak charging time of public charging stations in China in 2020 is between 17:00-21:00, and the peak charging time of community charging stations is between 17:00-24:00 (Figure R1.1). As the charging hours are during the peak time of residential electricity consumption and less renewable energy is available for power generation at night, the grid emission intensity of the nighttime charging is actually higher than the average values[12]. This phenomenon is more obvious in summer than that in winter. Therefore, using the grid average GHG emission intensity of electricity production of each province would underestimate the GHG emission of BEV charging and cause the GBET estimates to be lower than the actual value. The relevant text reads, in Lines 347-353:

“The use of annual average power grid emission factors without considering the seasonal and daily effects on the electricity mix might underestimate our estimation of GBET. For example, most of the BEVs in China are charged at night⁵⁴, when the grid emission intensity is higher than average since residential electricity demand peaks and less renewable energy is available for power generation at this time⁵⁵. Using the marginal electricity emission factors⁵⁶ allows for more accurate estimations, though doing so is challenging due to the lack of data.”

Figure R1.1. Distribution of charging time of public charging stations and community charging stations in China in 2020. The figure is reproduced using the data from the Application Analysis of Charging Data for New Energy Vehicles in China[11], published by the State Grid Corporation of China.

In addition, due to the high complexity of modern power systems, including numerous interacting generation, conversion, transmission, distribution, and end-use technologies, as well as consumer behavior, which all affect the actual consumption of electricity, the results of the short-term marginal power factor have high uncertainty[12-15]. We consider the effect of emission intensity of electricity production fluctuations on the GBET results in the uncertainty analysis.

155: VKT: this is annual vkm travelled, and not lifetime vkm travelled, correct? Please state explicitly.

Yes, this is the annual VKT. We clarified this in the revised manuscript. Now the sentence reads, in Lines 192-194:

“ The cross-province variances of four factors (Fig.2b), including battery capacity, curb weight, annual vehicle kilometres travelled (VKT), and the local power grid’s GHG emission intensity, might explain the regional heterogeneity of GBET. ”

Figure 3b: Is this the average vehicle weight of both powertrain types? Please clarify.

Thanks for the question. It refers to the vehicle weight of BEVs. We provide this information in the newly added Figure 2, in replacement of the original Figure 3. The revised figure is as follows:

Fig. 2. The GBET of BEVs and influencing factors by province in 2018. **a.** The average GBET of BEVs by province in 2018. Data for the Hong Kong Special Administrative Region (SAR), Macao SAR and Taiwan province are unavailable. **b.** Four influencing factors of GBET by province in 2018, including average curb weight, average battery weight, GHG emission factors of power grids, and average annual Vehicle Kilometres of Travel (VKT) at the provincial level.

204-216: The authors point to the importance of further shortening BEV GBET and provide a sensitivity analysis of certain key parameters on the GBET. However, is a 1% improvement in each of these parameters equally likely or “easy” to achieve? The authors mention in the caption for Figure 6 that GHG emissions intensity of the grid has a larger “improvement space,” but has the lowest effect on GBET per Figure 5. How do these counter-acting patterns balance out? Some comment on this by e.g., drawing from historical/projected trends in these parameters may help the relevance.

Many thanks to the insightful comment! The purpose of sensitivity analysis is to identify the sensitivity of the variables, which is the magnitude of the impact of unit changes on the results. As we adopt the one-variable-at-a-time perturbation approach, we compare the sensitivity of variables on the same magnitude of variable changes, i.e., every 1% change in each parameter.

However, the possibility of a 1% change is different for various variables, and low sensitivity does not refer to lower volatility, as the reviewer pointed out. For example, although the GBET estimates are most sensitive to variations in vehicle weight (with an elasticity of 1.7), this parameter only increases by 5.6% from 2012 to 2018. By contrast, although the sensitivity coefficient of GHG emission intensity of the local power grid to GBET is relatively low (i.e., 0.5), the decreasing rate of GHG emission intensity of the local power grid in each province from 2012 to 2018 is up to 78.1%, and there is more room to improve the grid decarbonization according to China’s plan[16].

As such, the uncertainty analysis of GBET estimates is not only relevant to the variable sensitivity but also relevant to their volatility ranges. According to the historical and projected data, the temporal changes in vehicle weight, battery capacity, and GHG emission intensity of materials are relatively smaller than the variations of GHG emission intensity of power grids and VKT. We thus set larger uncertainty ranges for the latter two variables, as Figure S2 shows. The changing range for GHG emission factors is referenced to the reported values in previous literature (Table S12). And the VKT is referenced from the National Big Data Alliance of New Energy Vehicles (NDANEV)[7] (Table S10-S11).

204-216: Can the authors provide further insight in the text with regards to why the relative ranking of the six factors is what it is? Is it because the GBET “front loads” the production emissions and therefore these hold a heavier weight than with average lifecycle emissions intensities that spread production emissions over the entire lifetime?

We really appreciate the insightful question and the helpful thoughts shared by the reviewer. We agree that the estimation of GBET “front loads” the production emissions is one of the reasonable explanations for the dominant role of vehicle/battery weight in the sensitivity analysis. Another reason is that for GBET estimation, we only need to compare the GHG debt with the fuel-cycle GHG emissions before the break-even point comes (the paid-back GHG). This is different from the previous LCA analysis, which calculates the fuel-cycle GHG emissions in the whole vehicle lifetime. The paid-back GHG is much less than the whole fuel-cycle emissions, which weakens the influencing power of fuel-cycle factors relative to vehicle-cycle ones. We added such explanations in the revised manuscript, specifically in

Lines 238-243:

“We can tell that GBETs are more sensitive to vehicle-cycle factors (the former four factors) than fuel-cycle factors (the latter two). This is different from previous LCA studies, which revealed stronger effects of fuel-cycle factors on life-cycle emissions than vehicle-cycle ones^{5,27-29}. The differences are probably because in the estimation of GBET, only GHG emissions prior to the break-even point are counted. The scale of such emissions is much smaller than the life-cycle emissions counted in the LCA, resulting in a weakened influencing power of fuel-cycle factors.”

207-209: What about vehicle lifetime? While I understand that the total lifetime vkm travelled per vehicle does not affect GBET, it is generally the argument that batteries with larger capacities will have longer lifetimes in terms of vkm travelled, which arguably compensates for their higher production emissions. This could be something worth acknowledging – that while GBET is indeed important, there are other factors usually accounted for in other LCA studies that are not captured here (and vice versa) that are important to consider in assessing the climate mitigation effects of vehicle electrification.

We really appreciate the insights provided by the reviewer. We have added a new paragraph to discuss this point. The relevant text reads, in Lines 295-303:

“It is worth mentioning that a smaller GBET does not necessarily yield higher emission reduction over the life cycle. For example, BEVs with a greater driving range tend to have a longer lifetime, so they can bring more emission reduction benefits over the full life cycle. However, increased driving range often requires greater battery capacity and weight, which increases GHG emissions from battery production and leads to higher GHG debt. Thus, it will take longer to pay off these GHG debts in the GBET. The GBET is a supplementary indicator to the existing metrics, as it provides information on how quickly the climate benefits are generated, while previous LCA assessment suggests the size of the benefits throughout the vehicle’s lifetime. Both the information is important to consider in assessing the climate mitigation effects of vehicle electrification.”

Figure 6: I think rather a sequential rather than a diverging colour scheme would be more appropriate here. This colour scheme also seems counterintuitive, as green generally implies “good” while red implies “bad” and yellow “neutral”.

Figure 6: Furthermore, I think this Figure should be expanded to include all parameter pairings; for example, an improvement in VKT does not preclude and decrease in the power grid intensity.

Very helpful comments! We performed the revision from two aspects following the reviewer’s suggestion. First, we re-performed the uncertainty analysis and considered all parameter pairings. Considering there are six groups of sensitive variables needed to be considered in the uncertainty and the full factorial design requires 486 ($3^5 \times 2^1$) mixes, we use the Orthogonal experimental design (OED) to reduce the number of mixes and select the optimal level combinations. Second, we redrew the figure of the uncertainty analysis results. The

revised text and the associated figure are as follows, also in Lines 543-560:

“We further took these sensitive factors into consideration in the uncertainty analysis. The most common method of uncertainty analysis is the Monte Carlo simulation. However, it is challenging for the GBET estimation as the distribution curve of multiple input parameters, especially those of the LCI data, are elusive. Here, we conduct uncertainty analysis by combining the range approach with orthogonal experimental design (OED). The range approach tests the effects of sampling the parameters at the extreme of their range of variability on the output uncertainty^{69,70} so as to avoid making judgement about the probability of different occurrences⁷¹. We assumed an uniform coverage of the uncertainty input space, i.e., $\pm 5\%$, for these factors: curb weight, battery capacity, GHG emission factors of vehicle material production, and GHG emission factors of battery material production. For GHG emission intensity of the power grids, we used the GHG emission factors of power generation from the IPCC report⁶⁸ in the basic estimation and the low or high values referenced from studies in the Chinese context as the two extreme ends (Table S13-S15). For the annual VKT, we considered a conservative scenario and a radical development scenario to reflect the variations (Table S10-S11). The OED is an effective method for arranging and analyzing multi-factor interactions. As an alternative to presenting all combination forms of multiple factors, the OED method efficiently schedules multifactorial experiments with optimal combination levels^{72,73}. For the six sensitive factors, we used an orthogonal table (Figure S3) containing 18 representative scenarios to investigate their combined impacts, following the guidelines of ref. Yang et al.⁷⁴.”

Figure S3. The GBET uncertainty using the range approach and orthogonal experimental design (OED) method. S1-S18 represent the 18 scenarios under the orthogonal experimental design, and the number of each sector in the chart represents the GBET corresponding to each scenario.

243-252: While older BEV vintages indeed generally did not provide climate mitigation effects, these vehicles are likely retired or approaching retirement (another reason to make these data available if possible!); it seems that most BEVs today are expected to provide mitigation effects over ICEVs. What, then, is the policy implication of this? Perhaps tie this back to the policies mentioned in the Introduction. The push for EVs both in China and globally is seemingly inevitable: these production emissions will happen regardless. How can the findings of this paper be better put to use “in the real world”? How does this way of accounting for emissions affect, e.g., China’s climate goals? The benchmarking activity discussed in other comments will help with this.

Thanks for the questions. We reorganized and rewrote the policy implications in the revision. The relevant text reads, in Lines 264-327:

“The GBET estimation in this study alerts policy-makers that BEVs’ climate benefits are not free but are conditional upon paying back their GHG debt accrued in the vehicle production stage. This circumstance also brings an understanding of the delayed climate benefits of China’s BEVs from an abstract level to a concrete threshold. Such findings have enormous implications for the real world.

First, new GBET-based indicators can be developed to guide BEV deployment. For example, the percentage of BEVs that have reached greenhouse gas break-even time (PER-GBET) is an indicator to supplement the widely used indicator EV penetration rate (EPR). In other words, it is not how many EVs are produced and sold but how many have positive emission reductions that contribute to the climate benefits of the transportation sector. Relevant policies targeting the indicators could be modified accordingly. China and many other countries have leveraged multiple procurement incentives, such as tax credits, discounts or rebates, exemption of EVs from congestion controls, and separate licence plate quotas for electric vehicles³⁰⁻³³, to pursue higher EPR. However, once the sale is complete, these policies are no longer in effect, leaving the true climate effect unmanaged³⁴. A direct solution to this problem is to set follow-up policies for BEV deployment, such as performing stage subsidies (i.e., extending the subsidy timeline from the purchase time point to the time when it achieves its GBET), investing in charging infrastructure, and motivating ICEV replacement, to promote higher PER-GBET.

GBET can also inform technical standards for the life expectancy or battery replacement time of BEVs to ensure net climate benefits. For example, China currently requires a homogeneous battery warranty period of no less than eight years or 120,000 kilometres in mileage (whichever occurs first)²⁶. However, as our estimations in the previous sections show, not all vehicles can achieve their GBET within the battery warranty period, including some large-size or large-mode vehicles (approximately 8% of the total between 2016 and 2018). This situation calls for a longer required warranty time for heavy transport modes. In fact, GBET provides guidelines for differentiated warranty times and other longevity-relevant standards. This process, on the one hand, facilitates climate benefits by avoiding the early replacement of batteries and, on the other hand, motivates BEV suppliers to improve the climate performance of their products.

It is worth mentioning that a smaller GBET does not necessarily yield higher emission reduction over the life cycle. For example, BEVs with a greater driving range tend to have a longer lifetime, so they can bring more emission reduction benefits over the full life cycle. However, increased driving range often requires greater battery capacity and weight, which increases GHG emissions from battery production and leads to higher GHG debt. Thus, it will take longer to pay off these GHG debts in the GBET. The GBET is a supplementary indicator to the existing metrics, as it provides information on how quickly the climate benefits are generated, while previous LCA assessment suggests the size of the benefits throughout the vehicle's lifetime. Both the information is important to consider in assessing the climate mitigation effects of vehicle electrification.

In addition, although trade-offs might exist between life-cycle emissions reductions and faster payback times, there is still some room for synergy. Policy-makers can encourage more explorations in these areas to make BEV emission reductions faster and better. One of the strategies is to reduce the carbon debt by lightweighting^{20,35-39}, material recycling^{40,41}, battery recycling and reuse⁴²⁻⁴⁴. Another method is to accelerate GHG debt repayment by intensifying the usage of existing BEVs via car sharing or prioritizing BEVs for taxis⁴⁵. Intensifying BEV use rather than expanding vehicle ownership would simultaneously shorten GBET, achieve better GHG emission reductions and solve other problems, such as traffic congestion, mineral resources depletion, infrastructure construction pressure and environmental pollution^{46,47}. This strategy is feasible, as essentially what people truly need is high-quality transportation service rather than the vehicle itself⁴⁸. Moreover, aligning BEV production sites with the planned renewable power grid can facilitate faster and higher BEV GHG emissions reductions⁴⁹. Currently, China's BEV and battery productions are mainly distributed in the southeast coastal region and the northeast where grid GHG emission intensity is relatively high (see more details in Figure S5). The geographical spread of battery and car manufacturers in China is determined by historical production advantages, such as the availability of mature production lines. For example, Ampere Technology Co., Limited (CATL), the largest EV battery manufacturing company in China, initially produced phone batteries. Its historical production advantages facilitate agglomeration externalities, technology spillover, and productivity gains, allowing it to quickly shift to EV battery production. As a step forwards to match the low-carbon development of power grids, CATL constructed more factories in southwestern provinces with abundant renewable energy, such as the first-zero-carbon factory built in Yibin, Sichuan province⁵⁰. Incorporating cleaner electricity production into the layout of EV production is favourable for both shortening GBET and reducing life-cycle emissions."

272: Please supplement this with other references, such as:

- Luk, Jason M., et al. "Review of the fuel saving, life cycle GHG emission, and ownership cost impacts of lightweighting vehicles with different powertrains." *Environmental science & technology* 51.15 (2017): 8215-8228.*
- Raugei, Marco, et al. "A coherent life cycle assessment of a range of lightweighting strategies for compact vehicles." *Journal of Cleaner Production* 108 (2015): 1168-1176.*

- Burd, Joshua Thomas Jameson, et al. "Improvements in electric vehicle battery technology influence vehicle lightweighting and material substitution decisions." *Applied Energy* 283 (2021): 116269.

- Das, Sujit, et al. "Vehicle lightweighting energy use impacts in US light-duty vehicle fleet." *Sustainable materials and technologies* 8 (2016): 5-13.

Thanks for the helpful suggestions. We have included the suggested references along with more relevant literature to strengthen the statement. Specifically, in Lines 307-308:

"One of the strategies is to reduce the carbon debt by lightweighting^{20,35-39}, material recycling^{40,41}, battery recycling and reuse⁴²⁻⁴⁴."

292-305: *What about intensifying use, e.g., prioritizing BEVs for car sharing programs or taxis? This would lead to a rapid shifting of stock and take better advantage of (presumably) rapidly decarbonizing electricity mixes in production (which would also shorten the GBET).*

Great point! We added it in the following paragraph (Lines 309-314):

"Another method is to accelerate GHG debt repayment by intensifying the usage of existing BEVs via car sharing or prioritizing BEVs for taxis⁴⁵. Intensifying BEV use rather than expanding vehicle ownership would simultaneously shorten GBET, achieve better GHG emission reductions and solve other problems, such as traffic congestion, mineral resources depletion, infrastructure construction pressure and environmental pollution^{46,47}. This strategy is feasible, as essentially what people truly need is high-quality transportation service rather than the vehicle itself⁴⁸."

308-309 *There are certainly many more studies that have come to the same conclusion! I think this statement warrants multiple references.*

Thanks to the reviewer's suggestion. Although this sentence was removed in the rewriting of policy implications, we have paid attention to adding sufficient literature evidence for other statements in the revisions. Please see more examples in the response to the general comment on References.

312-314 *Please add a sentence qualitatively assessing the grid carbon intensity of the regions where existing production lines are – are they relatively high carbon or low?*

Thanks for the suggestion. We have added the sentence, as well as a supporting figure, in the revised materials, specifically in Lines 316-318:

"Currently, China's BEV and battery production are mainly distributed in the southeast coastal region and the northeast, where grid GHG emission intensity is relatively high (see more details in Figure S5)."

Figure S5. Production volume of BEV batteries and vehicles by province from 2012-2018.
a. Battery production by province. **b.** Vehicle production by province. Data for the Hong Kong Special Administrative Region (SAR), Macao SAR, and Taiwan province are unavailable.

Methods:

The methods section is clear with regards to how the calculations are performed. However, it is difficult for the reader to discern the data sources and assumptions made for many aspects of the work. This obscurity, combined with the lack of benchmarking against other studies as previously mentioned, unfortunately makes the entire study a bit of a ‘black box’.

We really appreciate your valuable comments. The revisions to address this critical comment are threefold.

- First, we rewrote the data sources' descriptions and clarified the estimation's key assumptions, as shown in the newly added sub-section “**Data sources and assumptions.**”
- Second, we provided more statistical descriptions and analysis for the empirical data, as Figure S5-S9 shows in previous responses and in the SI.
- Moreover, we compared the LCI data we used and the break-even point estimates with those from other studies, as Table S17 and Table S2 show in previous responses and in the SI.

The matching of BEV and ICEV models seems to be key to the outcomes of the study. The authors write that they compare “each of the BEVs produced and sold during 2012-2018” to “ICEVs in the same vehicle model and size class.”

- Is the dataset used literally a database of each individual vehicle in the Chinese fleet? Are the comparisons made on an individual car basis, or just by vintage and size class?

Thanks for the questions. As mentioned in previous responses (General comments-Methods), the production and sale data are at the individual vehicle level, referenced from China’s Compulsory Traffic Accident Liability Insurance (CTALI), which is provided by the China Automotive Technology & Research Center (CATARC)[4, 5]. The CTALI database records 227 types of BEV models and 1667 types of ICEV models from 2012 to 2018. For each vehicle model, more technical specifications, including the model type, curb weight, battery weight, battery capacity, and fuel consumption were collected from the Announcement of Vehicle Manufacturing Enterprises and Vehicle Products[5] which is governed by the Ministry of Industry and Information Technology (MIIT) of China. The fuel consumption for each model type was based on the New European Driving Cycle (NEDC) testing conditions[6]. Combining the technical specifications with each vehicle’s production year and sale location, we can identify the regional heterogeneity of GBET for the same vehicle model, using the VKT data and GHG emission factors of power grids that vary across provinces.

Enabled by these data and reasonable assumptions, the assessment for BEVs is on an individual vehicle basis, and each of the BEVs has multiple potential ICEV benchmarks. The detailed pairing methods are provided in lines 409-424:

*“**Matching methods between BEVs and ICEVs.** Since the GBET of BEVs is calculated at the vehicle level, we find fuel-powered counterparts for each of the BEVs produced and sold from 2012 to 2018. One BEV can have multiple fuel-powered counterparts because, in the real world, consumers’ choices for fuel-powered substitutes of a certain BEV have many possibilities. Considering that in most cases, the replacements happen in the same vehicle class, we compare each of the BEVs with the ICEVs in the same vintage, transport mode (Car, SUV, and MPV), and size class (A00, A0, A, B, and C) (see more details of the vehicle clarification in Table S5) to generate basic estimations, referring to Fig. 4. As the comparison is “one (BEV) to more (ICEVs),” for systematic comparison, we generate an average representative of the selected ICEVs, whose parameters are the average of the*

matched counterpart ICEVs. Then, the comparison turns to “one (BEV) to one (representative ICEV). The GBET of BEVs within the same stratum are then averaged to generate an overall estimate. In the uncertainty analysis, we consider more possibilities of the substitutes across different classes and more possibilities of the representative ICEVs (see sensitivity and uncertainty analysis for more details).

Fig. 4. Scheme of the GBET estimation process and matching methods between BEVs and ICEVs

- What properties are used to determine the comparative ICEV (do all BEV models in China have a corresponding ICEV model as well)?

Thanks for the question. For the comparative ICEV, we didn't require it to be the same model with the studied BEV. This is because a BEV buyer might not be the potential buyer for the ICEV in the same model. We pair BEVs and ICEVs by their vintage, transport mode and size class. More specifically in the basic estimation, we compare each BEV with the average GHG emissions of ICEVs in the same vintage, transport mode and size class. In the uncertainty analysis, we changed the comparison benchmarks to ICEVs in adjacent size classes and in

various GHG emission intensities, for the purpose of presenting the impact of varying ICEV benchmarks on GBET estimations.

- Perhaps the authors can make a comment as to how good these matches of BEVs and ICEVs are and how these assumptions affect results. Using a parameter such as curb weight without any adjustment would be biased as BEVs would likely weigh more than an ICEV with a similar physical footprint.

Thanks for the insightful recommendation. In the revision, we added the uncertainty analysis regarding the matching methods. More pairing possibilities across size classes or among various fuel efficient vehicles were considered, in addition to the benchmark of using the average of the same size class in the basic estimation. The relevant text reads, in Lines 562-573 (methods):

“Different comparison methods between BEVs and ICEVs. The GBET of BEVs is also highly dependent on the comparison benchmark of ICEVs. In the basic estimations, we used the average level of ICEVs in the same vehicle classification (i.e., production year, transport mode, and size class) as the benchmark for each BEV. Considering the possibility that BEV buyers might not be potential buyers for an ICEV in the same size class, we compared each BEV with ICEVs in adjacent size classes (see Table S8). Moreover, to present the impact of varying ICEV benchmarks on GBET estimations, we not only used the average level of ICEVs as references but also considered the pessimistic and optimistic situations by comparing the BEVs to the most and least efficient ICEVs in the uncertainty analysis. More specifically, if the studied BEV is an A0-class SUV, we used the average emission level and the top and bottom 25% emissions level of fuel-powered A0-class SUVs as benchmarks in the comparison (see Figure S4). Compiling these scenarios facilitates a more comprehensive understanding of the GBET estimations.”

And in Lines 254-261 (results):

“Compared between adjacent size classes, the GBET estimations fluctuate from -74% to 156% (Table S8). Compared with the same size class but using the ICEVs whose GHG emissions are in the lower quartile (i.e., top 25% low-emission ICEVs), the GBET increases by 1.9-6.7 years, with an average increase of 3.9 years. In this case, nearly half of BEVs sold in 2018 cannot repay GHG debt within 11 years. When we change the benchmark to the ICEVs whose GHG emissions are in the higher quartile (i.e., top 25% high-emission ICEVs), the GBET decreases by 1.6-5.1 years, with an average decrease of 2.9 years (Figure S4). In this case, all BEVs sold in 2018 achieve a GBET within 7 years and 95% of them actually within 3 years.”

Table S8. The average GBET of BEVs in China by transport mode and size class, compared with ICEVs in various size classes.

Transport mode	Size class	ICEV's size class	2012	2013	2014	2015	2016	2017	2018
Car	A00	-	-	-	-	-	-	-	-
		A00	2.2	3.1	2.3	2.5	3.5	3.3	3.2
		A0	1.2	1.9	1.3	1.5	2.3	2	1.8
	A0	A00	5.6	5.6	5.9	5.9	5.3	5.2	5.3
		A0	4.2	4.1	4.2	4.2	3.8	3.3	3.8
		A	3	2.9	2.9	2.9	2.4	1.9	2.5
	A	A0	6.5	6.2	6.6	8.2	8.7	8	7.6
		A	5.0	4.7	4.9	7.1	7.1	6.4	6.3
		B	3.1	2.8	2.7	4.4	4.4	3.8	3.3
	B	A	-	-	10.9	11	10.8	9	9.6
		B	-	-	8.1	9.8	7.8	7.5	7.3
		-	-	-	-	-	-	-	-
SUV	A0	A00	-	-	-	-	-	-	-
		A0	3.7	3.7	3.6	-	5.1	5.7	5.2
		A	1.2	1.7	1.7	-	3.2	3.7	3.2
	A	A0	-	-	5.0	8.2	8.2	7.5	6.4
		A	-	-	2.9	3.2	5.6	5.0	4.3
		B	-	-	1.4	1.6	3.5	3.2	2.7
	B	A	-	-	-	-	5.6	6.2	4.7
		B	-	-	-	-	3.8	4.1	3.1
		C	-	-	-	-	1.7	1.9	1.3
	C	B	-	-	-	-	-	-	8
C		-	-	-	-	-	-	4.8	
-		-	-	-	-	-	-	-	
MPV	A0	A00	-	-	-	-	-	-	-
		A0	8.6	9.4	10.5	10.9	10.9	7.8	8.4
		A	8.2	8.8	9.4	9.7	9.6	2	4
	A	A0	-	-	-	-	-	8.8	7.7
		A	-	-	-	-	-	7.2	6.1
		B	-	-	-	-	-	4.3	3.5
	B	A	-	-	-	-	11	7.6	8.6
		B	-	-	-	-	9.0	5.2	5.8
C		-	-	-	-	-	-	-	

Notes: SUV = sports utility vehicle; MPV = multi-purpose vehicle. Short string - indicate insufficient data for the transport mode in that year. The confidence level corresponds with the sample size. Sample size <1000, [1000,10000], >10000 corresponds to low confidence, medium confidence, and high confidence, represented by *, **, and ***, respectively.

Figure S4. GBET of BEVs compared with various emission-level ICEVs. Panel (b) compares all of the BEV samples with the average-emission ICEVs in the same vehicle transport mode and size class car by car. Panel (a) and (c) compare all of the BEV samples with the low-emission (bottom 25%) ICEVs and high-emission (top 25%) ICEVs in the same transport mode and size class, respectively. The bars represent the share of BEVs whose GBET is in a certain range (distinguished by color).

Do the BEVs and ICEVs that are compared have different VKT (the description in lines 365-366 is ambiguous as to this).

Thanks for the question. Although the annual VKT of BEVs and ICEVs are different in the real world, we compare them based on the same annual mileages travelled, which are assumed to be equal to BEVs' VKT. The reason of such processing is as follows: the estimation of GBET focus on the GHG debt in the production phase and the payback effect in the use phase, the latter of which relies on the effective substitution mileage of BEVs for ICEVs. In other words, how much GHG debt is paid off in a given year depends on how many mileages travelled of ICEVs are replaced by BEVs. Since the real values of effective substitution mileage are elusive as they are highly relevant to consumer behaviors, we made an assumption that the effective substitution mileage yearly is equal to the annual VKT of the studied BEV. For example, if a BEV is rarely used with an annual VKT of 1,000 km, its effective replacing mileage for ICEVs is assumed as 1,000 km yearly. The mitigation effect brought by the substitution is the difference in GHG emissions between BEVs and ICEVs when both of them drive the same 1,000 km. As such, the annual mileages travelled used in the GBET estimation are the same for both of the powertrain types. We have clarified this point and admit its limitations in the revised manuscript, as presented in Lines 330-341:

“Although our findings have great implications, we also noticed that there are several limitations. First, we assume that a BEV’s yearly effective substitution mileage for ICEVs is the annual average VKT in the province it is sold, without considering the rebound effect or spillover effect⁵¹ of BEV uptake on GHG emissions. This assumption might bias the GBET estimation. In the scenario where the first-time car owner purchased a BEV to replace public transportation service rather than ICEVs, the effective substitution mileage is lower than the BEV’s annual VKT and results in an underestimation of the GBET. In the other scenario where a BEV is bought to actually replace the already owned ICEV, due to the limitation on driving range, the user might reduce the car usage compared to owning an ICEV. A positive spillover effect occurs, and the effective substitution mileage is higher than the BEV’s annual VKT. In fact, to what extent BEVs effectively substitute ICEVs is complicated, as it is relevant to consumers’ behaviours³⁴; this relationship has not been fully discussed and leaves ample opportunities for future research.”

In general, please be very clear and explicit as to how this is done (the term “one by one” is also somewhat open to interpretation).

Thanks for the comment. We have rewritten the whole method section. In particular, we clarify that by “one by one,” we mean the GBET of each BEV is assessed by comparing it with its fuel-powered counterparts at the vehicle level. The relevant text reads, in Lines 409-422:

*“**Matching methods between BEVs and ICEVs.** Since the GBET of BEVs is calculated at the vehicle level, we find fuel-powered counterparts for each of the BEVs produced and sold from 2012 to 2018. One BEV can have multiple fuel-powered counterparts because, in the real world, consumers’ choices for fuel-powered substitutes of a certain BEV have many*

possibilities. Considering that in most cases, the replacements happen in the same vehicle class, we compare each of the BEVs with the ICEVs in the same vintage, transport mode (Car, SUV, and MPV), and size class (A00, A0, A, B, and C) (see more details of the vehicle clarification in Table S5) to generate basic estimations, referring to Fig. 4. As the comparison is “one (BEV) to more (ICEVs),” for systematic comparison, we generate an average representative of the selected ICEVs, whose parameters are the average of the matched counterpart ICEVs. Then, the comparison turns to “one (BEV) to one (representative ICEV). The GBET of BEVs within the same stratum are then averaged to generate an overall estimate. In the uncertainty analysis, we consider more possibilities of the substitutes across different classes and more possibilities of the representative ICEVs (see sensitivity and uncertainty analysis for more details).”

Given the regionalized grid GHG emission factors presented in table S5: are vehicle production emissions also regionalized? If so, please describe where is this information from (or what assumptions are made). On a similar note, please also document the geographical assumptions for the LCA factors presented in Table S3 in the methods or the SI.

Yes, vehicle production emission is regionalized. According to the data of China’s Compulsory Traffic Accident Liability Insurance, which is provided by the China Automotive Technology & Research Center, we acquire production and sale information at the vehicle level, including battery production location, vehicle production location, vehicle sale location, and specific vehicle model of each BEV and each ICEV. The information allows us to link each vehicle with the GHG emission intensity of the power grid in its production location. However, for other LCI data, such as the GHG emission factors of materials used in the production, we didn’t distinguish them by region and made a homogeneous assumption in the estimation. We clarified this in the Methods. The relevant text reads, in Lines 498-518:

“The emission intensity of electricity generation by province was calculated based on the power generation structure and GHG emission factors across power generation types, assuming that the electricity consumption structure is the same as that of electricity generation. Such an assumption might underestimate the emission intensity because marginal electricity consumption for BEVs usually relies on coal and natural gas power plants whose operations are relatively stable with higher GHG emission intensities than the grid structure. The provincial power generation structures from 2012 to 2019 were acquired from the China Electricity Council⁶⁶, and those from 2020 to 2028 were from forecasting data referenced from Li et al. (2021)⁶⁷. The GHG emission factors of different power generation technologies (i.e., coal, wind, solar, nuclear, etc.) were referenced from the IPCC Fifth Assessment Report (AR5)⁶⁸. In the basic estimations, we used the medium value reported by IPCC AR5; this value falls within the range of most existing research on GHG emissions from power generation technologies in China (see more literature in Table S12). In the uncertainty analysis, we employed the maximum and minimum values from existing research. The results of the emission intensity of the power grid by province are presented in Table S13-S15.

The last category of data is the life cycle inventory (LCI) data from the latest China

Automotive Life Cycle Database (CALCD)-2021 (see Table S16-S17), developed by the CATARC⁶². These data are homogeneous across provinces. We compared the LCI data from CALCD-2021 with two internationally well-known LCI databases, the Greenhouse gases, Regulated Emissions, and Energy use in Technologies Mode (GREET) and ecoinvent 3.6. We found fairly high consistency among the databases (see Table S17).”

What assumptions/data are used for the use phase energy intensity (fuel intensity)? Are these values reported (i.e., empirical), or standard driving cycles, or something else? Are they specific to each vehicle/model? Can these values be published?

Thanks for the question. We use the official fuel efficiency data based on New European Driving Cycle (NEDC) test condition as the use-phase energy intensity. These data are specific to each model type and obtained from the Ministry of Industry and Information Technology (MIIT) of China[6]. The values are published on the following website:
<https://yhgsx.miit.gov.cn/fuel-consumption-web/mainPage>.

403: Please add citations for this statement.

Many thanks. Although the statement is removed, we provided more citations to support our work wherever possible in the revised manuscript.

404-405: given the range of VKT given in these lines, it is difficult to conclude that the annual mileage in China is, in fact, much less than 15 000 km (for example, if Tibet were an outlier). Can the authors provide more information regarding the VKT dataset?

Yes, we can. We added more descriptions as well as the statistical analysis of the VKT dataset. Relevant text can be found in Lines 474-485:

“The third category of data was reported at the province level, including annual Vehicle Kilometres of Travel (VKT) for both BEVs and ICEVs and emission intensity of power grids. The VKT data for 2018 were extracted from the National Big Data Alliance of New Energy Vehicles (NDANEV)⁶⁵, which records the real-world driving, charging, and fault status of vehicles car by car. According to the requirements of national standard GB/T 32960, the data are uploaded to the platform every 30 seconds when the vehicle is driving, and the fault state is uploaded every second. Between 2018 and July 17, 2022, the NDANEV accessed 9.27 million new energy vehicles with a total VKT of 295.5 billion kilometres. Although the data are real-world and car-by-car, car-level VKT data were not used in the GBET analysis since we had no access to the vehicle identification information to match the NDANEV database with the CTALI database. Thus, we aggregated the data at the provincial level, assuming that the VKT of vehicles within the same provinces is homogeneous. More statistical information on the real-world VKT data is provided in the SI (see Figure S1-S2).”

More statistical analysis can be found in the SI:

Figure S1. Average annual mileage distribution of passenger cars in China in 2018.

Figure S2. Sales volume and average VKT of BEVs by province in 2018. Data for the Hong Kong Special Administrative Region (SAR), Macao SAR, and Taiwan province are unavailable.

427: is there a reference (website or otherwise) for this data source? Is trade between provinces and neighbouring countries included and if so, how is this considered in the calculations?

Yes, we have added the reference as follows:

National Bureau of Statistics of China. China Electric Power Yearbook, <<https://www.yearbookchina.com/navibooklist-n3021112403-1.html>> (2021).

The electricity transmission between provinces and neighboring countries was not considered in our estimations. We acknowledge this limitation in the revised manuscript. To compensate for this limitation, we consider the impact of possible changes in the GHG emission intensity of electricity on the GBET calculation in the sensitivity and uncertainty analysis.

432-439: Can the authors comment as to how applicable the global median electricity generation factors are to China? Are there any China-specific factors that can be used instead or as part of a sensitivity analysis?

Great comment! Inspired by the reviewer’s question, we performed another round of literature review and collected more China-specific factors as follows (Table S11). It is found that the global median electricity generation factors are within the wide range of China-specific factors collected from other studies.

Table S12. GHG emission factors of selected power generation technologies in China from existing studies (unit: gCO₂eq/kWh).

Power generation technology	Coal-fired power	Hydro power	Wind power	Nuclear power	Photovoltaic power
Ding et al. (2017) ²⁹	1045	15.5	8.42	6.31	50.2
Wang et al. (2019) ³⁰	-	3.84	28.3	12.4	-
Li et al. (2019) ³¹	744.5	-	31.32	-	-
Gao et al. (2019) ³²	-	-	51.57	-	13.5
Wang et al. (2021) ³³	660-1050	12.4	3-41	3-35	13-190
IPCC (this study) ²⁷	820	24	11	12	48

For the robustness check, we took the variances of these values into consideration in uncertainty analysis. Specifically for the GHG emission factors of power generation, we adopt low, high, and median values, as shown in the SI Table S13-S15.

432-439: Can the authors comment as to the effect of seasonal and diurnal effects on the electricity mix?

It is our pleasure to add such information to clarify the question. In our estimation, we assume the GHG emission intensity of electricity use by BEVs is the same as the annual average intensity of power generation. However, the assumption would underestimate the GHG

emission of BEV charging and cause the GBET estimates to be lower than the actual value. According to the Application Analysis of Charging Data for New Energy Vehicles in China[11] published by the State Grid Corporation of China, the charging time of public charging stations in China in 2020 peaked during 17:00-21:00, and the charging time of community charging stations peaked in 17:00-24:00 (Figure R1.1). As the peak charging hours are also the peak time of residential electricity consumption and less renewable energy is available for power generation in the night, the grid emission intensity of the nighttime charging is actually higher than the average values[12]. This phenomenon is stronger in summer than that in winter. The relevant text in the revised manuscript reads, in Lines 348-354:

“The use of annual average power grid emission factors without considering the seasonal and daily effects on the electricity mix might underestimate our estimation of GBET. For example, most of the BEVs in China are charged at night⁵⁴, when the grid emission intensity is higher than average since residential electricity demand peaks and less renewable energy is available for power generation at this time⁵⁵. Using the marginal electricity emission factors⁵⁶ allows for more accurate estimations, though doing so is challenging due to the lack of data.”

Figure R1.1. Distribution of charging time of public charging stations and community charging stations in China in 2020. The figure is reproduced using the data from the Application Analysis of Charging Data for New Energy Vehicles in China[11], published by the State Grid Corporation of China

Supplementary Information:

Figure S2 warrant a full page, I think. It may also be worth considering a different way of presenting the information as in its current state, it is somewhat difficult to interpret. Changing the caption title and titling the colour legend might help – do the colours represent the change in GBET (as implied by the caption), or the absolute GBET? For propriety, it would be interesting to see the figures for the corner comparisons as well.

Thanks to the reviewer's comments, we have replaced the obscure figure with the following one.

Figure S4. GBET of BEVs compared with various emission-level ICEVs. Panel (b) compares all of the BEV samples with the average-emission ICEVs in the same vehicle transport mode and size class car by car. Panel (a) and (c) compare all of the BEV samples with the low-emission (bottom 25%) ICEVs and high-emission (top 25%) ICEVs in the same transport mode and size class, respectively. The bars represent the share of BEVs whose GBET is in a certain range (distinguished by color).

Can the authors comment on the current state of where vehicles are produced (i.e., in the main analysis)? How was it determined where regions get their BEVs from in Figure S3? The additional analysis in the SI regarding the effect of regional seems valuable, but is rather confusing. What is meant by “vehicle percentage”? This is very unclear from the text, caption and the corresponding results – it seems that each column should sum to 100%, but Guizhou, for example, seems to have 100% imports from Beijing, but also 50% imports (yellow) from most of the other regions.

Yes, we added more explanation on how the current geographical spread of battery and vehicle production is formed. Specifically, in Lines 318-323:

“The geographical spread of battery and car manufacturers in China is determined by historical production advantages, such as the availability of mature production lines. For example, Amperex Technology Co., Limited (CATL), the largest EV battery manufacturing company in China, initially produced phone batteries. Its historical production advantages facilitate agglomeration externalities, technology spillover, and productivity gains, allowing it to quickly shift to EV battery production. ”

We drew two new figures to explicitly present where the regions get their batteries and vehicles. With regards to how it was determined where regions get their BEVs from, we found very limited supporting materials at the macroeconomic scale since the purchase decisions are made by corporate, which are affected by the commercial influencing factors such as partnership, price advantages, and technology preference.

Figure R1.2. The trade volume of BEV batteries among provinces from 2012-2018.

Figure R1.3. The trade volume of BEV vehicles among provinces from 2012-2018.

This additional analysis also warrants some further discussion, particularly in where the data come from and how the results are produced: were the authors able to substitute the energy use in vehicle and battery production with the specific energy mix from each province, or is the lifecycle inventory database regionalized, and this is a representation of that? Perhaps the authors can also highlight the provinces with the largest production volumes.

Thanks for the helpful suggestions. As mentioned in previous responses, we added a new section, “Data and key assumptions,” and clarified that the GHG emissions associated with energy use in the vehicle are rationalized, and the LCI data are not. We also added a new figure (Figure S3) to highlight the provinces with the largest production volumes.

Figure S5. Production volume of BEV batteries and vehicles by province from 2012-2018. **a.** Battery production by province. **b.** Vehicle production by province. Data for the Hong Kong Special Administrative Region (SAR), Macao SAR, and Taiwan province are unavailable.

Minor comments:

*81 recommendation: use alternative term for "one-by-one" (is this literally car by car?)
Car by car*

Yes, it is literally car-by-car. We have replaced "one-by-one" with “car-by-car” or “at the vehicle level” in the revised manuscript for clarification.

137 "hatchures" : -

It has been corrected. Many thanks!

170 Fix: kwh  kWh

Thank you for correcting the spelling for us.

172 Change ~ to -

Thank you for your reminder, we have replaced the "~" here with "-" in the manuscript.

174-176 Please expand on what "other factors" are/could be

We rewrote the explanations for the regional heterogeneity of GBET, and the sentence has been removed from the revised manuscript.

Fig 3a use of the terms "top" and "bottom" regions is confusing

Thanks for pointing it out. By “top” and “bottom,” we mean the provinces whose GBET is the longest and shortest, respectively. We noticed the expression is obscure. The previous Figure 3 has been removed, and its key information is combined in the new Figure 2.

189: 38 kg CO₂e/kWh should be g CO₂e/kWh. This is also a problem in Table S5 – please check the manuscript and SI thoroughly for other occurrences.

Thanks for the observation and we have thoroughly checked all the materials and corrected the typos.

Figure 4: kg CO₂e: are these values for the entire vehicle lifetime? Per year? Please specify. What is the assumed lifetime?

Thanks for the question. We intended to mean the annual GHG emissions of BEVs associated with their effective replacing mileages in a specific year. However, as the manuscript has been almost rewritten, Figure 4 has been removed and clarified in other parts in the revised version.

204: “GHG emission factors” please specify – are these production emission factors?

Yes, they are production emission factors. We modified the sentence as follows, also in Lines 235-237:

“The top six most sensitive factors, in descending order, are curb weight, GHG emission factors of vehicle material production, battery capacity, GHG emission factors of battery material production, annual VKT, and GHG emission factors of power grids (shown in Table S7).”

As for the GHG emission factors of vehicle component production and battery material production, they are grouped variables calculated by multiple GHG emission factors of materials and the component/material weight, whose unit is kgCO_{2e} per kg of vehicle weight/battery weight. The purpose of grouping is to simplify the presentation of sensitivity analysis results. We clarified this in Lines 534-537, as well as in Table S6:

“Since the LCI data are large in volume, we grouped them into four factors for ease of execution: GHG emission factors of vehicle material production, GHG emission factors of battery material production, Electricity consumption during the vehicle production stage, Electricity consumption during the battery production stage (see more details in Table S6).”

Table S6. Variables considered in the sensitivity analysis.

Grouped variable	Life cycle inventories included	More details
Curb weight	N/A	Refer to Figure S7
Battery capacity		
Battery weight		
Test-based fuel consumption		Refer to Table S13-S15.
GHG emission factors of the power grid		
Annual vehicle kilometers traveled (VKT)		
GHG emission factors of vehicle material production	Vehicle Components (%)	Refer to Table S16.
	Tire Components (%)	
	Lead-acid Battery Materials Components (%)	
	Fluids Components(%)	
	Material GHG emission factor (kgCO _{2e} /kg)	Refer to Table S17.
GHG emission factors of battery material production	Li-ion Battery Materials Components(%)	Refer to Table S16.
	Material GHG emission factor (kgCO _{2e} /kg)	Refer to Table S17.
Electricity consumption during the vehicle production stage	Electricity (kWh per vehicle)	297-345
	Curb weight (kg)	Refer to Figure S7
Electricity consumption during the battery production stage	Electricity (kWh/kWh)	5.8
	Battery capacity (kWh)	Refer to Figure S7

222: I would argue that the VKT are part of the vehicle cycle, and not the fuel cycle.

Thanks for the comment. The VKT in our manuscript refers to the mileage traveled annually rather than the lifetime VKT. The VKT is probably different due to varying charging infrastructure availability and different consumer behaviors of the same vehicle models that are driven in various provinces or in different years. Moreover, the VKT is one of the key parameters for estimating fuel-cycle GHG emissions (for BEVs, there is no fuel in the use phase, but the terminology is used conventionally). Therefore, we grouped the VKT as a fuel-cycle parameter and clarified this in the revised manuscript.

297: “charging infrastructure availability”, not “charging infrastructure’s availability”

Thanks for the correction and we also took this opportunity to proofread the manuscript with professional English editing services.

408: NDANEV is not defined anywhere – what does it stand for and is there a reference?

NDANEV stands for the National Big Data Alliance of New Energy Vehicles. It is the database where we acquire the VKT data. The full name of the abbreviations was explained, and a reference was added. Specifically in Lines 475-479:

“The VKT data for 2018 were extracted from the National Big Data Alliance of New Energy Vehicles (NDANEV)⁶⁵, which records the real-world driving, charging, and fault status of vehicles car by car. According to the requirements of national standard GB/T 32960, the data are uploaded to the platform every 30 seconds when the vehicle is driving, and the fault state is uploaded every second.”

437: sentence fragment

Thank you and this sentence has been removed from the revised manuscript.

There does not seem to be a Figure 1.

Thanks for the neat observation. We have renumbered the figures in the manuscript and double-checked the format before resubmission.

The monochromatic colourmaps in Figures 2, 3 and 4 can be challenging to distinguish – perhaps consider at least a two colour theme or using hatching patterns.

Thanks for the suggestion. The information present in previous Figures 2, 3, and 4 have been reproduced and presented in a new Figure 2 in the revised manuscript. The new figure uses a two-color theme, as Figure 2a shows.

Fig. 2. The GBET of BEVs and influencing factors by province in 2018. **a.** The average GBET of BEVs by province in 2018. Data for the Hong Kong Special Administrative Region (SAR), Macao SAR and Taiwan province are unavailable. **b.** Four influencing factors of GBET by province in 2018, including average curb weight, average battery weight, GHG emission factors of power grids, and average annual Vehicle Kilometres of Travel (VKT) at the provincial level.

Figure 6: I would argue that the VKT is a parameter describing the vehicle cycle (i.e., is a property of the vehicle, not the fuel chain)

Thanks for the comment. The VKT in our manuscript refers to the mileage traveled annually rather than the lifetime VKT. For the same vehicle models that are driven in various provinces or different years, the VKT is probably different due to varying charging infrastructure availability and consumer behaviors. Moreover, the VKT is one of the key parameters for estimating fuel-cycle GHG emissions (for BEVs, there is no fuel in the use phase, but the terminology is used conventionally). Therefore, we grouped the VKT as a fuel-cycle parameter and clarified this in the revised manuscript.

Do you have any data regarding the distribution of vehicle lifetimes in China, especially split by powertrain type? It would be interesting to see how many BEVs exit the fleet before their GBET if this data is available.

It is really an interesting point! However, we have little access to empirical data of vehicle lifetimes in China. We will consider it as future work once the data is available.

As an alternative, we use battery warranties as an assessment threshold and investigate how many BEVs achieve GBET within the battery warranty. This assessment provides multiple implications for climate policy decisions and longevity-relevant standard design. The relevant discussion reads, in Lines 144-148:

“Moreover, approximately one-fifth of the BEVs produced and sold before 2016 failed to pay back the GHG debt within five years, which is the EV battery warranty time required by the Chinese government in 2014²⁵. In 2016, the required battery warranty was extended to eight years²⁶, and 8% of BEVs produced and sold between 2016 and 2018 have failed to achieve the GBET within the battery warranty threshold (see more supplementary results in Table S4).”

In Lines 284-293 (policy implications):

“GBET can also inform technical standards for the life expectancy or battery replacement time of BEVs to ensure net climate benefits. For example, China currently requires a homogeneous battery warranty period of no less than eight years or 120,000 kilometres in mileage (whichever occurs first)²⁶. However, as our estimations in the previous sections show, not all vehicles can achieve their GBET within the battery warranty period, including some large-size or large-mode vehicles (approximately 8% of the total between 2016 and 2018). This situation calls for a longer required warranty time for heavy transport modes. In fact, GBET provides guidelines for differentiated warranty times and other longevity-relevant standards. This process, on the one hand, facilitates climate benefits by avoiding the early replacement of batteries and, on the other hand, motivates BEV suppliers to improve the climate performance of their products.”

We would like to reiterate our gratitude to the reviewers who contributed excellent comments to our paper. Thanks to your thoughtful suggestions, the manuscript has been greatly improved. Below please kindly find the references mentioned in the responses.

References

1. laboratory, A.n. *The Greenhouse gases, Regulated Emissions, and Energy use in Technologies Model*. 2022; Available from: <https://greet.es.anl.gov/>.
2. ecoinvent. *ecoinvent Database*. Available from: <https://ecoinvent.org/the-ecoinvent-database/>.
3. *The Ministry of Public Security released national motor vehicle and driver data for the first half of 2022*. Road traffic management, 2022 (07): 5. (in Chinese).
4. (EACA) Energy-saving and Green-development Assessment Center for Automobile Industrial. *China Automobile Low Carbon Action Plan (CALCP) Research Report 2021*. 2021; Available from: <http://www.auto-eaca.com/a/chengguofabunarong/ziliaoxiazai/zhongguoqichedit/2021/1206/416.html>.
5. *Auto Data Center of CATARC (only in Chinese now)*. 2022; Available from: <http://www.catarc.info/>.
6. MIIT (Ministry of Industry and Information Technology). *China automobile fuel consumption inquiry system*. 2022; Available from: <https://yhgscx.miit.gov.cn/fuel-consumption-web/mainPage>.
7. *National Monitoring and Management Center for New Energy Vehicles*. 2022; Available from: <https://www.evsmc.cn/>.
8. National Bureau of Statistics of China. *China Electric Power Yearbook*. 2021; Available from: <https://www.yearbookchina.com/navibooklist-n3021112403-1.html>.
9. Li, H.R., et al., *Catchment-level water stress risk of coal power transition in China under 2 degrees C/1.5 degrees C targets*. Applied Energy, 2021. **294**.
10. Schlömer S., T.B., L. Fulton, E. Hertwich, A. McKinnon, D. Perczyk, J. Roy, R. Schaeffer, R. Sims, P. Smith, and R. Wiser,, 2014: *Annex III: Technology-specific cost and performance parameters*. In: *Climate Change 2014: Mitigation of Climate Change. Contribution of Working Group III to the Fifth Assessment Report of the Intergovernmental Panel on Climate Change [Edenhofer, O., R. Pichs-Madruga, Y. Sokona, E. Farahani, S. Kadner, K. Seyboth, A. Adler, I. Baum, S. Brunner, P. Eickemeier, B. Kriemann, J. Savolainen, S. Schlömer, C. von Stechow, T. Zwickel and J.C. Minx (eds.)]*. Cambridge University Press, Cambridge, United Kingdom and New York, NY, USA.
11. *Application Analysis of Charging Data for New Energy Vehicles in China*. State Grid Electric Vehicle Service Co., Ltd., 2021 (in Chinese).
12. Arvesen, A., et al., *Emissions of electric vehicle charging in future scenarios: The effects of time of charging*. Journal of Industrial Ecology, 2021. **25**(5): p. 1250-1263.
13. McCollum, D.L., et al., *Improving the behavioral realism of global integrated assessment models: An application to consumers' vehicle choices*. Transportation Research Part D-Transport and Environment, 2017. **55**: p. 322-342.
14. Arvesen, A., et al., *Life cycle assessment of transport of electricity via different voltage levels: A case study for Nord-Trøndelag county in Norway*. Applied Energy, 2015. **157**: p. 144-151.
15. Amjad, M., et al., *A review of EVs charging: From the perspective of energy optimization, optimization approaches, and charging techniques*. Transportation Research Part D-Transport and Environment, 2018. **62**: p. 386-417.
16. Commission, N.D.a.R. *Clean Energy Consumption Action Plan (2018-2020)*. 2018; Available from: https://www.ndrc.gov.cn/xxgk/zcfb/ghxwj/201812/t20181204_960958.html?code=&state=123.

Reviewer #2 (Remarks to the Author):

1. *“The White House stated that half of all new cars sold in the United States in 2030 should be EVs”*

The White House announcement is just a target, not a mandatory one.

Thanks for pointing out the inaccurate expressions. We have revised the sentence, and now it reads, in Lines 48-50 (the line numbers in this response letter correspond to those in the **clean version** of manuscript):

“On August 5, 2021, the White House announced a target of 50% electric for all new vehicles sold in 2030⁸.”

2. *“ From 2012 to 2013, only 15% of the BEVs could pay back the carbon debt within the battery lifetime (the real lifetime of China’s BEV battery in this period is approximately 3-4 years). In 2014-2015 and 2016-2018, this number increased to 80% and 92%, respectively, as the batteries’ real lifetime extended to around five years and ...”*

I see only one reference for the battery lifetime. I am curious whether they are any other references that show similar results. The number, 3-4 years seem low. Also, what is considered the end of a lifetime? Is the battery capacity below a certain threshold?

Thanks for the valuable questions. The number is cited from a Chinese publication, and according to the authors, it is determined based on empirical surveys in China (more details of the data provided by the study are shown in the table below). According to this study, the first phase of the demonstration and promotion of new energy vehicles in China started in 2009. In the first few years, the power batteries in use were short-lived (generally 2-3 year). They were not very mature in technology, and there were no uniform regulations on quality assurance requirements. After 2011, with the rapid development of the new energy vehicle industry, power battery technology has been developed to a certain extent, and the lifespan has been extended to 3-4 years. In 2014, the Ministry of Finance, the State Administration of Taxation, and the Ministry of Industry and Information Technology jointly issued the "Announcement on Exemption from the Purchase Tax of New Energy Vehicles"[1], which set the warranty for new energy vehicle power batteries to be no less than five years or 10,000 kilometers (whichever comes first). In 2015, the Ministry of Finance, the Ministry of Science and Technology, the Ministry of Industry and Information Technology, and the National Development and Reform Commission jointly issued the "Notice on Financial Support Policies for the Promotion and Application of New Energy Vehicles in 2016-2020"[2] on the battery shelf life of new energy passenger vehicle companies. The warranty period was set to no less than eight years or 120,000 kilometers (whichever comes first).

Table R1.1. The distribution of the passenger BEVs' battery lifespan in China

Year	Average service life (years)	Distribution (%)						
		P2	P3	P4	P5	P6	P7	P8
2012	3	0.1	0.8	0.1				
2013	3.4		0.6	0.4				
2014	3.8		0.2	0.8				
2015	5			0.3	0.4	0.3		
2016	5.9				0.3	0.5	0.2	
2017	7.1					0.2	0.5	0.3
2018	7.6						0.4	0.6

Notes: The table is a translation of Li & Li (2018)'s work. P_i is the percentage of all batteries whose battery service lifetime is i years in each given year.

However, as there are few empirical data from other studies to cross-check its reliability, we removed this reference and relevant statements. As an alternative, we referenced the official vehicle battery warranty periods in China and compared the GBET estimates in our study with the battery warranties. Since battery manufacturing accounts for most of the GHG debt of BEVs and the lifespan of batteries is usually shorter than the vehicle itself, identifying BEVs that could achieve GBET in the battery warranties (Table S4) provides a supplementary indicator for climate benefits assessment from the temporal perspective. The relevant text reads, in Lines 144-148 :

“Moreover, approximately one-fifth of the BEVs produced and sold before 2016 failed to pay back the GHG debt within five years, which is the EV battery warranty time required by the Chinese government in 2014²⁵. In 2016, the required battery warranty was extended to eight years²⁶, and 8% of BEVs produced and sold between 2016 and 2018 have failed to achieve the GBET within the battery warranty threshold (see more supplementary results in Table S4).”

And in Lines 284-293 (policy implications):

“GBET can also inform technical standards for the life expectancy or battery replacement time of BEVs to ensure net climate benefits. For example, China currently requires a homogeneous battery warranty period of no less than eight years or 120,000 kilometres in mileage (whichever occurs first)²⁶. However, as our estimations in the previous sections show, not all vehicles can achieve their GBET within the battery warranty period, including some large-size or large-mode vehicles (approximately 8% of the total between 2016 and 2018). This situation calls for a longer required warranty time for heavy transport modes. In fact, GBET provides guidelines for differentiated warranty times and other longevity-relevant standards. This process, on the one hand, facilitates climate benefits by avoiding the early replacement of batteries and, on the other hand, motivates BEV suppliers to improve the climate performance of their products.”

Table S4. The percentage of BEVs achieving GBET within the battery warranty period.

Transport mode	Year	2012	2013	2014	2015	2016	2017	2018
	Threshold	Five-year battery warranty				Eight-year battery warranty		
Car	A00	100%	100%	98%	89%	95%	97%	99%
	A0	98%	99%	97%	93%	100%	100%	99%
	A	78%	98%	69%	38%	72%	89%	86%
	B	-	-	0%	0%	70%	61%	66%
SUV	A0	100%	100%	100%	-	99%	94%	96%
	A	-	-	100%	90%	99%	100%	100%
	B	-	-	-	-	100%	100%	100%
	C	-	-	-	-	-	-	100%
MPV	A0	29%	10%	6%	1%	1%	42%	36%
	A	-	-	-	-	-	82%	96%
	B	-	-	-	-	41%	93%	74%

Notes: Cell values in the table denote the percentage of BEVs produced and sold in that year achieving GBET within the battery warranty period, i.e., five years required in 2014²² and eight years required in 2016²³. Since there is no official requirement before 2014, we assume five years (2014 requirement) for 2012 and 2013. A blank cell means there is no corresponding vehicle data.

3. Besides the contributing factors that the author indicated in the paper, the fuel economy of the comparable ICEV is also very important factor that changes the results.

Have you also considered a sensitivity study of comparing with a different size of ICEV? The EV buyers are probably not potential buyers for an average ICEV in the same size class. As the authors also indicated that the GBET of BEVs is highly dependent on the reference of ICEVs for comparison.

Thanks for the insightful comments. As the reviewer pointed out, the fuel economy of the comparable ICEV is indeed a contributing factor to GBET, and the BEV buyers do not always pick ICEVs in the same size class as BEVs. In order to present the impacts of ICEV reference selection on GBET estimates, we performed the comparisons between BEVs and ICEVs across size classes; we also used the most and least efficient ICEVs as benchmarks to present pessimistic and optimistic estimations. The relevant method description are provided in Lines 521-573:

*“We considered two sources of uncertainties that might modulate the estimations of GBET. One is the parameter uncertainties in LCA analysis, and **the other is the comparison methods between BEVs and ICEVs.**”*

Parameter uncertainties. ... (the quotation for this part is omit here)

Different comparison methods between BEVs and ICEVs. The GBET of BEVs is also highly dependent on the comparison benchmark of ICEVs. In the basic estimations, we used the average level of ICEVs in the same vehicle classification (i.e., production year, transport mode, and size class) as the benchmark for each BEV. Considering the possibility that BEV buyers might not be potential buyers for an ICEV in the same size class, we compared each BEV with ICEVs in adjacent size classes (see Table S8). Moreover, to present the impact of varying ICEV benchmarks on GBET estimations, we not only used the average level of ICEVs as references but also considered the pessimistic and optimistic situations by comparing the BEVs to the most and least efficient ICEVs in the uncertainty analysis. More specifically, if the studied BEV is an A0-class SUV, we used the average emission level and the top and bottom 25% emissions level of fuel-powered A0-class SUVs as benchmarks in the comparison (see Figure S4). Compiling these scenarios facilitates a more comprehensive understanding of the GBET estimations.”

And the results are in Lines 253-260:

“Compared between adjacent size classes, the GBET estimations fluctuate from -74% to 156% (Table S8). Compared with the same size class but using the ICEVs whose GHG emissions are in the lower quartile (i.e., top 25% low-emission ICEVs), the GBET increases by 1.9-6.7 years, with an average increase of 3.9 years. In this case, nearly half of BEVs sold in 2018 cannot repay GHG debt within 11 years. When we change the benchmark to the ICEVs whose GHG emissions are in the higher quartile (i.e., top 25% high-emission ICEVs), the GBET decreases by 1.6-5.1 years, with an average decrease of 2.9 years (Figure S4). In this case, all BEVs sold in 2018 achieve a GBET within 7 years and 95% of them actually within 3 years.”

Table S8. The average GBET of BEVs in China by transport mode and size class, compared with ICEVs in various size classes.

Transport mode	Size class	ICEV's size class	2012	2013	2014	2015	2016	2017	2018
Car	A00	-	-	-	-	-	-	-	-
		A00	2.2	3.1	2.3	2.5	3.5	3.3	3.2
		A0	1.2	1.9	1.3	1.5	2.3	2	1.8
	A0	A00	5.6	5.6	5.9	5.9	5.3	5.2	5.3
		A0	4.2	4.1	4.2	4.2	3.8	3.3	3.8
		A	3	2.9	2.9	2.9	2.4	1.9	2.5
	A	A0	6.5	6.2	6.6	8.2	8.7	8	7.6
		A	5.0	4.7	4.9	7.1	7.1	6.4	6.3
		B	3.1	2.8	2.7	4.4	4.4	3.8	3.3
	B	A	-	-	10.9	11	10.8	9	9.6
		B	-	-	8.1	9.8	7.8	7.5	7.3
		-	-	-	-	-	-	-	-
SUV	A0	A00	-	-	-	-	-	-	-
		A0	3.7	3.7	3.6	-	5.1	5.7	5.2
		A	1.2	1.7	1.7	-	3.2	3.7	3.2
	A	A0	-	-	5.0	8.2	8.2	7.5	6.4
		A	-	-	2.9	3.2	5.6	5.0	4.3
		B	-	-	1.4	1.6	3.5	3.2	2.7
	B	A	-	-	-	-	5.6	6.2	4.7
		B	-	-	-	-	3.8	4.1	3.1
		C	-	-	-	-	1.7	1.9	1.3
	C	B	-	-	-	-	-	-	8
C		-	-	-	-	-	-	4.8	
-		-	-	-	-	-	-	-	
MPV	A0	A00	-	-	-	-	-	-	-
		A0	8.6	9.4	10.5	10.9	10.9	7.8	8.4
		A	8.2	8.8	9.4	9.7	9.6	2	4
	A	A0	-	-	-	-	-	8.8	7.7
		A	-	-	-	-	-	7.2	6.1
		B	-	-	-	-	-	4.3	3.5
	B	A	-	-	-	-	11	7.6	8.6
		B	-	-	-	-	9.0	5.2	5.8
C		-	-	-	-	-	-	-	

Notes: SUV = sports utility vehicle; MPV = multi-purpose vehicle. Short string - indicate insufficient data for the transport mode in that year. The confidence level corresponds with the sample size. Sample size <1000, [1000,10000], >10000 corresponds to low confidence, medium confidence, and high confidence, represented by *, **, and ***, respectively.

Figure S4. GBET of BEVs compared with various emission-level ICEVs. Panel (b) compares all of the BEV samples with the average-emission ICEVs in the same vehicle transport mode and size class car by car. Panel (a) and (c) compare all of the BEV samples with the low-emission (bottom 25%) ICEVs and high-emission (top 25%) ICEVs in the same transport mode and size class, respectively. The bars represent the share of BEVs whose GBET is in a certain range (distinguished by color).

4. I see “GHG emission factors” and “GHG emissions intensity” are used interchangeably in the paper. Are they the same or different?

Thanks for the question. Grid emissions intensity is measured by GHG emission factors, which represent the amount of GHGs produced per unit of material/energy. In some cases the related terms emission factor and emission intensity are used interchangeably. During the revision, we paid attention to the phase use throughout the manuscript to enhance the overall flow and clarity.

5. Figure 4, please add the right unit, kg CO₂e per kwh. Also, Why do you use different scales of the unit in the figure and bar chart (kg vs g)?

Many thanks for the kind question. The information provided in the original Figure 2-4 has been combined and replaced by a new figure (Figure 2 in the revised manuscript). The

complement and consistency of the unit and legend in the new figure have been checked before resubmission.

Fig. 2. The GBET of BEVs and influencing factors by province in 2018. a. The average GBET of BEVs by province in 2018. Data for the Hong Kong Special Administrative Region (SAR), Macao SAR and Taiwan province are unavailable. **b.** Four influencing factors of GBET by province in 2018, including average curb weight, average battery weight, GHG emission factors of power grids, and average annual Vehicle Kilometres of Travel (VKT) at the provincial level.

6. Page 3: “...early stage of development19...” please delete “19”

Thanks! We have carefully proofread all the materials in the revision.

7. Table 1: what are the numbers in the parenthesis? I assume they are the number of samples. Better to state it clearly in the table title.

Yes, they are. To better present the information, we use the number of asterisks to represent the sample size level and put standard deviations in the parenthesis. The revised table reads, in Lines 182-187:

Table 1. The descriptive statistics of greenhouse gas break-even time of China’s battery electric vehicles by transport mode and size class

Transport mode	Size class	2012	2013	2014	2015	2016	2017	2018
Car	A00	2.2**	3.1**	2.3***	2.5***	3.5***	3.3***	3.2***
		(0.8)	(1.1)	(1.7)	(2.0)	(2.3)	(1.9)	(1.7)
	A0	4.2**	4.1**	4.2**	4.2***	3.8***	3.3***	3.8**
		(0.2)	(0.2)	(0.3)	(0.5)	(0.5)	(0.6)	(1.3)
	A	5.0*	6.2*	4.9*	7.1**	7.1***	6.4***	6.3***
		(0.5)	(0.3)	(0.8)	(2.6)	(1.3)	(1.6)	(1.6)
	B	-	-	8.1*	9.8**	7.8**	7.5**	7.3**
				(0.6)	(1.2)	(0.6)	(1.7)	(1.7)
SUV	A0	3.7*	3.7*	3.6*	-	5.1**	5.7**	5.2***
		(0.1)	(0.1)	(0.1)		(1.1)	(1.2)	(1.2)
	A	-	-	5.0*	3.2*	5.6**	5.0***	4.3***
				(0.1)	(0.9)	(1.2)	(0.7)	(1.0)
	B	-	-	-	-	3.8*	4.1*	3.1**
						(0.1)	(0.2)	(0.3)
	C	-	-	-	-	-	-	4.8***
								(0.7)
MPV	A0	8.6**	9.4**	10.5**	10.9**	10.9***	7.8***	8.4***
		(3.3)	(2.2)	(1.7)	(0.7)	(0.6)	(3.6)	(3.1)
	A	-	-	-	-		7.2*	6.1**
							(1.1)	(0.6)
	B	-	-	-	-	9.0*	5.2**	5.8*
						(1.4)	(1.6)	(2.3)

Notes: SUV = sports utility vehicle; MPV = multipurpose vehicle. A short string indicates insufficient data for the transport mode in the given year. The values in parentheses represent the standard deviations. Low, medium, and high confidence levels correspond to the sample size <1000, [1000,10000], and >10000 represented, denoted by *, **, and ***. The darker shade of red indicates longer GBETs.

8. *Figure 2: Add analysis year to the figure title. “...in 2018”*

Thanks, the analysis year has been added. Now the figure title reads, in Lines 223: “**Fig.2 The GBET of BEVs and influencing factors by province in 2018.**”

9. *“that more than 80% of BEVs produced and sold since 2014 can bring positive climate benefits within their battery lifetime.”*

This seems an important conclusion. I don’t see any figure or table indicating the results. Please advise.

Also, was a different battery lifetime considered for different vehicle model years? 3-4 years for MY2012-2013 vs. 8 years for MY2014 and beyond? Please be clear.

Thanks for pointing this out. We realize it could be an important conclusion. Accordingly, we revised from two aspects.

First, to make the findings more robust, we use the battery warranty period rather than the battery lifetime as the threshold in describing the findings. The change is mainly because there are few published empirical data for vehicle battery lifetime in China. To the best of our knowledge, there is a Chinese study, i.e., Li & Li (2018), providing such data but the reliability of the data is obscure as we cannot find other empirical data to do the cross-check. By contrast, the battery warranty period data is traceable as they are released in the official documents. Battery warranty periods are relevant to battery lifespan[3], although the latter, in most cases, are longer than the former (current forecasts is 10 to 20 years before they need to be replaced[4, 5]). Using the warranty period as the threshold, we describe our findings as follows, in Lines 144-148:

“Moreover, approximately one-fifth of the BEVs produced and sold before 2016 failed to pay back the GHG debt within five years, which is the EV battery warranty time required by the Chinese government in 2014²⁵. In 2016, the required battery warranty was extended to eight years²⁶, and 8% of BEVs produced and sold between 2016 and 2018 have failed to achieve the GBET within the battery warranty threshold (see more supplementary results in Table S4).”

Table S4. The percentage of BEVs achieving GBET within the battery warranty period.

Transport mode	Year	2012	2013	2014	2015	2016	2017	2018
	Threshold	Five-year battery warranty				Eight-year battery warranty		
Car	A00	100%	100%	98%	89%	95%	97%	99%
	A0	98%	99%	97%	93%	100%	100%	99%
	A	78%	98%	69%	38%	72%	89%	86%
	B	-	-	0%	0%	70%	61%	66%
SUV	A0	100%	100%	100%	-	99%	94%	96%
	A	-	-	100%	90%	99%	100%	100%
	B	-	-	-	-	100%	100%	100%
	C	-	-	-	-	-	-	100%
MPV	A0	29%	10%	6%	1%	1%	42%	36%
	A	-	-	-	-	-	82%	96%
	B	-	-	-	-	41%	93%	74%

Notes: Cell values in the table denote the percentage of BEVs produced and sold in that year achieving GBET within the battery warranty period, i.e., five years required in 2014²² and eight years required in 2016²³. Since there is no official requirement before 2014, we assume five years (2014 requirement) for 2012 and 2013. A blank cell means there is no corresponding vehicle data.

Second, we added more policy implications relevant to these findings. The revised text reads, in Lines 270-293:

“First, new GBET-based indicators can be developed to guide BEV deployment. For example, the percentage of BEVs that have reached greenhouse gas break-even time (PER-GBET) is an indicator to supplement the widely used indicator EV penetration rate (EPR). In other words, it is not how many EVs are produced and sold but how many have positive emission reductions that contribute to the climate benefits of the transportation sector. Relevant policies targeting the indicators could be modified accordingly. China and many other countries have leveraged multiple procurement incentives, such as tax credits, discounts or rebates, exemption of EVs from congestion controls, and separate licence plate quotas for electric vehicles³⁰⁻³³, to pursue higher EPR. However, once the sale is complete, these policies are no longer in effect, leaving the true climate effect unmanaged³⁴. A direct solution to this problem is to set follow-up policies for BEV deployment, such as performing stage subsidies (i.e., extending the subsidy timeline from the purchase time point to the time when it achieves its GBET), investing in charging infrastructure, and motivating ICEV replacement, to promote higher PER-GBET.

GBET can also inform technical standards for the life expectancy or battery replacement time of BEVs to ensure net climate benefits. For example, China currently requires a homogeneous battery warranty period of no less than eight years or 120,000 kilometres in mileage (whichever occurs first)²⁶. However, as our estimations in the previous sections show, not all vehicles can achieve their GBET within the battery warranty period, including some large-size or large-mode vehicles (approximately 8% of the total between 2016 and 2018). This situation calls for a longer required warranty time for heavy transport modes. In fact, GBET provides guidelines for differentiated warranty times and other longevity-relevant standards. This process, on the one hand, facilitates climate benefits by avoiding the early replacement of batteries and, on the other hand, motivates BEV suppliers to improve the climate performance of their products.”

*10. For “... as lightweight materials substitution may increase the emissions during BEV’s life cycle..” Please also consider the following reference.
Impacts of Vehicle Weight Reduction via Material Substitution on Life-Cycle Greenhouse Gas Emissions, <https://pubs.acs.org/doi/full/10.1021/acs.est.5b03192>*

Thanks for the suggestion. We have added the reference, as well as other relevant ones, in the revised manuscript. Please kindly see Lines 308 in the clean version of manuscript.

11. The language seems off in some places. Please do another editor review.

Thanks! We have proofread the manuscript with professional English editing services. Phrases and sentences in many places were revised or rewritten to improve the overall flow and clarity.

Again, we appreciate the reviewer’s invaluable contributions to our paper, which help improve the manuscript’s clarity and enrich the discussion and implications of the results significantly. References presented in the responses are listed below.

References

1. State Taxation Bureau of the People's Republic of China(in Chinese). *Announcement on Exemption from New Energy Vehicle Purchase Tax (in Chinese)*. 2014; Available from: <http://www.chinatax.gov.cn/n810341/n810755/c1150779/content.html>.
2. General Office of the State Council. *The Financial Support for the Promotion and Application of New Energy Vehicles During 2016-2020 (in Chinese)*. 2015; Available from: http://www.gov.cn/xinwen/2015-04/29/content_2855040.htm.
3. Sulzer, V., et al., *The challenge and opportunity of battery lifetime prediction from field data*. Joule, 2021. 5(8): p. 1934-1955.
4. eDF. *All about electric car batteries*. Available from: <https://www.edfenergy.com/electric-cars/batteries>.
5. J.D.POWER. *How Long Do Electric Car Batteries Last?* 2022; Available from: <https://www.jdpower.com/cars/shopping-guides/how-long-do-electric-car-batteries-last>.

Reviewer #3 (Remarks to the Author):

First of all, I would like to thank the Editorial Board for considering me as a reviewer for the manuscript. The manuscript is well written and provides a valuable discussion on battery electric vehicle (BEV) deployment. I believe it provides interesting insights for BEV deployment. However, I believe some aspects are preventing its publication, at its current state, in Nature Communications.

We really appreciate the reviewer's for favorable comments on the writing and the research significance of our study. Following the reviewer's constructive comments and suggestions, we have made significant changes to improve the manuscript. Point-to-point responses (in black) to each comment (*in brown italics*) are provided below. The line numbers in this response letter correspond to those in the clean version of manuscript.

First, the methodology lacks robustness in its definition and assumptions.

Thanks for the critical comment. We rewrote the methodology section to clarify the definitions and assumptions and improve the sensitivity/uncertainty analysis for the robustness check. As the whole section was rewritten, we omit the quote here but present part of them in the responses to specific comments below.

In addition, several aspects identified by the authors have already been identified in previous studies, e.g., the effective substitution mileage. Therefore, I believe the manuscript does not present innovative aspects in its assessment compared to existing studies. This does not mean the article is irrelevant, but it creates the need to contextualize the relevance of the results compared to other available studies. In light of this, further discussion is necessary to incorporate the implication of the findings in comparison to current results in the scientific literature.

Many thanks for the thoughtful comment and suggestions. As the reviewer mentioned, the concept of GBET is not new. There are several previous studies which have calculated the break-even points of BEVs with regard to the climate effect, such as ref. 19-23 cited in the revised manuscript. By contextualizing our study within the wider literature, we clarify that the novelty of the paper lies in threefold:

- **The large-scale data facilitating a comprehensive understanding.** Unlike previous studies which estimated the break-even points of specific vehicle models, the dataset we used consists of almost all of the BEVs (nearly 1.6 million) and over 80% of the ICEVs (145.9 million) produced and sold in China from 2012 to 2018. The large scale allows us to simultaneously investigate the climate performance from the full picture aspect (i.e., national perspective and regional heterogeneity) and down to the details (model perspective).
- **A timely reminder for paying attention to the temporal effect.** The full picture of GBET in the world's largest EV market (i.e., the Chinese market) would be a timely

reminder for policymakers to pay more attention to the temporal climate effect. Although the existence of “GHG debt” is not a secret, its role is overshadowed by the halo of emission reduction benefits in the use phase. Bringing the understanding of the delayed climate benefits of China's BEVs from an abstract level to concrete thresholds is the key to breaking the halo effect.

- **New implications for BEV deployment policies and longevity-relevant standard design.** Assessing the climate benefits from the temporal perspective provides new implications for BEV deployment policies and longevity-relevant standard design. GBET-based indicators, such as the share of BEVs that have achieved the GBET, could be a vital supplement for existing indicators of EV penetration rate. They provide additional dimensions for policymakers to consider, especially in how to promote effective substitution of BEVs for ICEVs, rather than the single-minded pursuit of penetration rate.

We added these clarifications in the main text. Moreover, thanks to the reviewer’s suggestion of comparing results across studies, we added a table (Table S2) in the SI to present our results and the relevant ones. We quote the table on the next page. Also, more discussions on the cross-study comparisons are provided in the main text (Lines 105-113):

“From the comparisons we can tell, our GBET estimates for BEVs in China are in general longer than the estimates for that in Europe²¹, which is about 2-3 years. There are two reasons for the differences. First, there are higher GHG emission factors for power grids in China than in Europe, given the dominant role of coal-fired power generation in China. Higher GHG emission factors weaken the mitigation effect of BEVs in the use phase and result in higher GBET. Second, the annual VKT of most BEVs in the Chinese market (as Figure S1-S2 shows) is lower than 15000 km, which has been widely assumed in previous studies. Lower annual VKT implies shorter effective substitution mileage for ICEVs and results in higher GBET.”

Table S2. Selected studies on the GHG emissions break-even points of BEVs.

Study	Region	Scope	Key assumptions			Life cycle inventory (LCI) database	Break-even points
			Emission intensity of electricity mix	Vehicle lifetime	Annual vehicle mileage traveled		
Ellingsen et al., 2016 ⁷	Europe	Four differently-sized EVs	521 g CO ₂ /kWh	12 years	15 000 km	Ecoinvent ¹²	44 000-70 000 km
Kim et al., 2010 ¹³	USA	Light-weighting vehicles using aluminum versus high-strength steel	GHG emission factors of the average U.S. grid	11-16 years	11 000 km	U.S. life cycle inventory (LCI) database (NREL 2007) ¹⁴	1-11 years (differ by light-weighting options)
Patterson et al., 2012 ¹⁵	Europe	Hybrid vehicles and EVs	UK electricity carbon intensity (594 gCO ₂ e/kWh)	10 years	20000 km	Patterson et al., 2011 ¹⁶	1.6-8.3 years (differ by energy scenarios)
International Council on Clean Transportation (ICCT), 2018 ¹⁷	Europe	Battery electric vehicle	European average grid electricity	150000 km	N/A	N/A	2-3 years (differ by carbon intensity of grid electricity)
Ambrose & Kendall, 2016 ¹⁸	USA	Plug-in electric vehicle	Archsmith et al., 2015	N/A	N/A	REET (Argonne National Laboratory, 2014) ¹⁹	Approximately 400–1150 charging cycles
This study	China	All passenger BEVs produced and sold in 2012-2018	38-801 g CO ₂ e/kWh (differ by regional electricity mix)	N/A	678-15927 km (vary by province)	CALCAT ^{20,21}	From zero to over eleven years with an average of 4.5 years

Notes: N/A indicates that the study didn't report such information.

I strongly encourage the authors to consider these changes and resubmit the manuscript. Further comments are presented below:

Thanks. These comments are important and help us improve the manuscript significantly. We have addressed them wherever possible in the revisions. Please find the point-to-point responses below.

- Line 18 (abstract): The so-called consensus still does not exist for the entire life cycle of BEV, just to their use phase. If a consensus existed, research on the emissions and impacts of BEV or even the trade-off of using biofuels during the transition to an electrified transportation system would not exist today. Therefore, this sentence is not accurate. However, the same does not apply to the sentence on line 64, as it provides a more accurate definition.

Thanks for this valuable suggestion. We agree with the reviewer and revised the sentence in the Abstract accordingly as (in Lines 21-24):

“While BEVs generate more climate benefits in the use phase than internal combustion engine vehicles (ICEVs), it is commonly ignored that their mitigation effect is usually delayed. BEV production is actually more carbon intensive than ICEVs, leaving a GHG debt to be paid back in the use phase.”

Similarly, the sentence in original Line 64 (now in Line 68-69) was also revised and now reads:

“Although the climate mitigation benefits of BEVs relative to ICEVs are favourable^{15,16}, a fact often ignored is that the benefits do not come for ‘free’.”

- Check the spacing in Table 1 caption.

Many thanks. The extra spaces in Table 1 have been removed. We also took this opportunity to check the manuscript format elsewhere.

- Line 184: From this line, the authors use the terminology 'fuel cycle' to describe the electricity use in BEV. This is not accurate. I understood that the authors are referring to energy to power BEVs, not fuel-related emissions. Please clarify.

Following this helpful suggestion, we added a note where the terminology ‘fuel cycle’ for BEVs first appears. The relevant text now reads, in Lines 165-167:

“This ordering implies that the increasing impacts of more GHG debt caused by the heavier weight of larger transport modes exceed the decreasing impacts caused by improving the debt repayment efficiency during the fuel cycle (the terminology fuel is used conventionally, referring to electricity production, transmission, and use for BEVs).”

Moreover, we further clarify the terminology in the Methods when the system boundary is defined. Specifically in Lines 393-401:

“The fuel cycle refers to “Well to Wheels (WTW)”, including the production of fuel (Well to Pump/WTP) and the use of energy (Pump to Wheels/PTW). For ICEVs, WTP includes crude oil extraction, refining, and processing; PTW refers to fuel combustion. For BEVs, the fuel terminology is used in a conventional sense, referring to electricity production, transmission, and use. GHG emissions of BEVs in WTP occur with electricity production and transmission, while GHG emissions of BEVs in PTW are zero, as there is no GHG emission during the use phase of electricity.”

- Line 230: Due to the missing units on the axis, it is difficult to understand what the authors want to show with the figure. Is it %?

Many thanks. We intended to present the sensitivity coefficients of GBET influencing factors, which measures the percentage change in the GBET for a 1% change in its influencing factor. In the revision, the figure has been replaced by Table S7 in the SI. We clarify the units of the values in the table.

Table S7. Sensitivity coefficients of the influencing factors.

Influencing factors	Life-cycle stage	Change in influencing factors	GBET change rate	Sensitive coefficient
Curb weight (kg)	Vehicle cycle	5.0%	8.5%	1.7
Vehicle Material GHG Emission Factor (kgCO ₂ e/kg)	Vehicle cycle	5.0%	6.0%	1.2
Battery capacity (kWh)	Vehicle cycle	5.0%	5.5%	1.1
Battery Material GHG Emission Factor (kgCO ₂ e/kg)	Vehicle cycle	5.0%	4.7%	0.9
Vehicle kilometers traveled (km/year)	Fuel cycle	5.0%	4.6%	0.9
GHG emission intensity of power grid (kgCO ₂ e/kWh)	Fuel cycle and Vehicle cycle	5.0%	2.4%	0.5
Battery weight (kg)	Vehicle cycle	5.0%	1.8%	0.4
Test-based fuel consumption (kWh/100km)	Fuel cycle	5.0%	1.6%	0.3
Vehicle Production_Electricity (kWh/kg)	Vehicle cycle	5.0%	0.6%	0.1
Battery Production__Electricity (kWh/kWh)	Vehicle cycle	5.0%	0.6%	0.1

Note: The sensitivity analysis is performed using the one-variable-at-a-time perturbation approach. The sensitivity coefficient refers to the change in GBET for every 1% change in the influencing factor.

- Line 235: check Figure 6 caption.

Thanks, as we have re-performed the sensitivity and uncertainty analysis, the Figure has been re-drew, and the caption is revised as well. The revised figure is provided below.

Figure S3. The GBET uncertainty using the range approach and orthogonal experimental design (OED) method. S1-S18 represent the 18 scenarios under the orthogonal experimental design, and the number of each sector in the chart represents the GBET corresponding to each scenario.

- The sentence in lines 255 to 257 is not valid. Several authors have been assessing other pathways to reduce the climate change impacts of BEV, based on its weight, including the weight of the batteries, among several other strategies. Therefore, this sentence should be revised.

We appreciate the reviewer pointing out this. We have deleted the sentence and restated the pathways to reduce the climate change impacts in Lines 305-327:

“In addition, although trade-offs might exist between life-cycle emissions reductions and faster payback times, there is still some room for synergy. Policy-makers can encourage more explorations in these areas to make BEV emission reductions faster and better. One of the strategies is to reduce the carbon debt by lightweighting^{20,35-39}, material recycling^{40,41}, battery recycling and reuse⁴²⁻⁴⁴. Another method is to accelerate GHG debt repayment by intensifying the usage of existing BEVs via car sharing or prioritizing BEVs for taxis⁴⁵. Intensifying BEV use rather than expanding vehicle ownership would simultaneously shorten GBET, achieve better GHG emission reductions and solve other problems, such as traffic congestion, mineral resources depletion, infrastructure construction pressure and environmental pollution^{46,47}. This strategy is feasible, as essentially what people truly need is high-quality transportation service rather than the vehicle itself⁴⁸. Moreover, aligning BEV production sites with the planned renewable power grid can facilitate faster and higher BEV GHG emissions reductions⁴⁹. Currently, China’s BEV and battery productions are mainly distributed in the southeast coastal region and the northeast where grid GHG emission intensity is relatively high (see more details in Figure S5). The geographical spread of battery and car manufacturers in China is determined by historical production advantages, such as the availability of mature production lines. For example, Amperex Technology Co., Limited (CATL), the largest EV battery manufacturing company in China, initially produced phone batteries. Its historical production advantages facilitate agglomeration externalities, technology spillover, and productivity gains, allowing it to quickly shift to EV battery production. As a step forwards to match the low-carbon development of power grids, CATL constructed more factories in southwestern provinces with abundant renewable energy, such as the first-zero-carbon factory built in Yibin, Sichuan province⁵⁰. Incorporating cleaner electricity production into the layout of EV production is favourable for both shortening GBET and reducing life-cycle emissions.”

- Line 281 to 284: The authors need to clarify these sentences as it is confusing. I believe they mean that BEVs are exempt from the lottery system. Is that correct? A deeper discussion on the role of policy in adopting the BEV should be included. See, e.g., Zhuge et al., 2020.

Exactly, many thanks. We rephrased the words as “*separate license plate quotas for electric vehicles*” in the revised manuscript. Following the reviewer’s insightful suggestion, we rewrote the discussions on the policy implications of GBET. The relevant text now reads, in Lines 270-282:

“First, new GBET-based indicators can be developed to guide BEV deployment. For example, the percentage of BEVs that have reached greenhouse gas break-even time

(PER-GBET) is an indicator to supplement the widely used indicator EV penetration rate (EPR). In other words, it is not how many EVs are produced and sold but how many have positive emission reductions that contribute to the climate benefits of the transportation sector. Relevant policies targeting the indicators could be modified accordingly. China and many other countries have leveraged multiple procurement incentives, such as tax credits, discounts or rebates, exemption of EVs from congestion controls, and separate licence plate quotas for electric vehicles³⁰⁻³³, to pursue higher EPR. However, once the sale is complete, these policies are no longer in effect, leaving the true climate effect unmanaged³⁴. A direct solution to this problem is to set follow-up policies for BEV deployment, such as performing stage subsidies (i.e., extending the subsidy timeline from the purchase time point to the time when it achieves its GBET), investing in charging infrastructure, and motivating ICEV replacement, to promote higher PER-GBET.”

- Lines 324-325: Although the authors acknowledged that battery recycling was not included, a better justification for its exclusion should be presented. To date, several articles provide secondary data to provide this assessment. The effects of battery recycling should be better discussed. Besides, the discussion of other strategies for the battery's end of life should be included, e.g., its use in a secondary application. See, e.g., Koroma et al. 2022.

Thanks for the suggestions. We excluded the effect of battery recycling and secondary application from our estimations for the following reasons: As the estimation of GBET focuses on the GHG debt in the production phase and the paid-back GHG before the break-even point comes, the battery's end-of-life strategies is actually beyond the scope of estimation. If considered, they would affect the production phase by reducing the raw material use. However, as the main task of this manuscript is to estimate the GBET of BEVs in the Chinese market from 2012 to 2018, when the electric cars are just starting to become popular in China, few recycling and secondary used batteries were used in the battery production (Table R3 showing the timeline of policy releases related to electric vehicle battery recycling). As such, for the period we estimated, excluding the end-of-life phase has very limited impact on the results.

Table R3. Policies related to electric vehicle battery recycling.

Document	Issuing Agency	Time	Reference URL
Interim Measures for the Management of New Energy Vehicle Power Battery Recycling (in Chinese)	Ministry Information Technology People’s Republic China; Ministry of Science and Technology of the People's Republic of China et al.	Feb 2018	http://www.gov.cn/xinwen/2018-02/26/content_5268875.htm
Interim regulations on traceability management of new energy vehicle power battery recycling (in Chinese)	Ministry Information Technology People’s Republic China	July 2018	http://www.huaibin.gov.cn/wzd/webinfo/2018/08/1534706988493194.htm
Notice on the new energy vehicle power battery recycling pilot work (in Chinese)	Ministry Information Technology People’s Republic China; Ministry of Science and Technology of the People's Republic of China et al.	July 2018	http://www.gov.cn/zhengce/zhengceku/2018-12/31/content_5440228.htm
People's Republic of China Solid Waste Pollution Prevention and Control Law (in Chinese)	National People's Congress Standing Committee	April 2020	http://www.npc.gov.cn/npc/c30834/202004/b6cf2e57b63b47818a275a37819c6b02.shtml

Despite the exclusion, we addressed the importance of battery end-of-life strategies on shortening the GBET in the future, by as Lines 307-308 show:

“One of the strategies is to reduce the carbon debt by lightweighting^{20,35-39}, material recycling^{40,41}, battery recycling and reuse⁴²⁻⁴⁴.”

- The sentence on lines 346-348 is inferential and does not add new information to the manuscript.

Many thanks. We have deleted the sentence and improved the ending paragraph in lines 363-372:

“Despite the limitations, our study expanded the understanding of BEVs’ climate benefit delays from a conceptual level to a concrete threshold measure using Chinese data. The scale of the data allows us to simultaneously investigate from a full picture perspective (i.e., national perspective and regional heterogeneity) down to a detailed one (vehicle model perspective). This study is a timely reminder for policy-makers to pay more attention to the temporal distribution of climate effects and provide guidelines for BEV deployment policies and longevity standard design. GBET-based indicators, such as the share of BEVs that have achieved the GBET, could be a vital supplemental factor for existing indicators of EV penetration rate. They provide additional dimensions for policy-makers to consider,

especially in promoting the effective substitution of BEVs for ICEVs, rather than just speeding up the EV deployment race.”

- Section "Methods", line 358. The authors do not describe the functional unit effectively used to provide the assessment.

Thanks for the critical comment. We rewrote the Methods section. The GBET estimations are based on the life cycle assessment (LCA) of vehicle GHG emissions and cross-vehicle comparisons between BEVs and their fuel-powered counterparts year by year. We begin this section by establishing the LCA setup. Then, we present how BEVs are paired with ICEV benchmarks for GBET estimation, as well as the data sources and key assumptions. Sensitivity and uncertainty analysis is finally performed to demonstrate how the results change with parameter assumptions and various paired ICEV benchmarks. We quote the titles of the sub-sections below:

Life cycle assessment of GHG emissions

Calculation of greenhouse gas break-even time

- Matching methods between BEVs and ICEVs.

- GBET estimations generated by comparing the matched BEVs and ICEVs.

Data sources and assumptions

Sensitivity and uncertainty analysis

- Parameter uncertainties.

- Different comparison methods between BEVs and ICEVs.

- Lines 380-381: The manuscript describes multiple system boundaries. However, it does not disclose the one adopted in the manuscript. When analyzing Figure S1, the system boundaries are disclosed, but they refer to fuel production and usage. Did the authors mean electricity production? The definition of the outputs also does not reflect the reality as other emissions occur during the manufacturing and use of BEV.

Thanks for pointing this out. We re-described the system boundary of LCA and redrew the figure as follows (in Lines 385-403):

“Life cycle assessment of GHG emissions

The GHG emissions of BEVs and ICEVs are estimated using the China Automotive Life Cycle Assessment Model (CALCM). The model is the compilation and evaluation of a vehicel system’s inputs, outputs, and potential environmental impacts over its life cycle⁶⁰. Here, we followed the instructions of national standards GB/T24040-2008, GB/T 24044-2008, and international standard ISO 14067-2018 to perform the assessment. For both BEVs and ICEVs, the life cycle system boundary evaluated in this study includes the vehicle cycle and fuel cycle of passenger vehicles. The vehicle cycle starts with raw material acquisition, then moves to material processing and manufacturing, complete vehicle production, and maintenance (tire, lead battery, and fluid replacement). The fuel cycle refers to “Well to Wheels (WTW)”, including the production of fuel (Well to Pump/WTP) and the use of energy (Pump to Wheels/PTW). For ICEVs, WTP includes

crude oil extraction, refining, and processing; PTW refers to fuel combustion. For BEVs, the fuel terminology is used in a conventional sense, referring to electricity production, transmission, and use. GHG emissions of BEVs in WTP occur with electricity production and transmission, while GHG emissions of BEVs in PTW are zero, as there is no GHG emission during the use phase of electricity. The transportation of materials, the manufacturing of equipment and infrastructure, and the production and treatment of manufacturing wastes are excluded (Fig. 3).”

Fig. 3. System boundary of life cycle assessment for both BEVs and ICEVs in this study

- The use of acronyms should be revised, for example, in lines 368 and 369.

Thanks for the comments. We have consistently use “LCA” to denote life cycle assessment.

- Lines 393-400: As the authors can not disclose the used primary data, more information is necessary to understand the relevance and coverage of the source what the database. This is a major shortcoming in the assessment of data quality. If possible, an assessment of its robustness or a comparison with other available data sources should be conducted to demonstrate its relevance.

Thanks to the reviewer’s helpful comment, we revised from three aspects. First, we rewrote the data source descriptions and clarified their coverage and our assumptions. The relevant text now reads, in Lines 450-518:

“Data sources and assumptions

The data used in the estimations of GBET can be classified into four categories according to their resolution levels (see Table S9). The first category is the real-world vehicle-level dataset, which contains the model type, year, and location of production and sales of almost all BEVs (nearly 1.6 million units) and 82% of ICEVs (145.9 million units) in China from 2012 to 2018 (see Figure S5-S6). The dataset was referenced from China’s Compulsory Traffic Accident Liability Insurance (CTALI), which is provided by the China Automotive Technology & Research Center (CATARC)^{61,62}. Since CTALI is compulsory for every vehicle registered in China, the data have wide coverage and high credibility. The vehicle-level data allow us to distinguish the GBET of BEVs across vehicle models, years, and locations.

The second category is vehicle model information. The CTALI database recorded 227 types of BEV models and 1667 types of ICEV models from 2012 to 2018. For each vehicle model, more technical specifications, including the model type, curb weight, battery weight, battery capacity, and fuel consumption were collected from the Announcement of Vehicle Manufacturing Enterprises and Vehicle Products⁶³ which is governed by the Ministry of Industry and Information Technology (MIIT) of China. The fuel consumption for each model type was based on the New European Driving Cycle (NEDC) testing conditions⁶⁴. More statistical descriptions of the data are provided in Figure S7-S9. These technical details were used in the LCA analysis, enabling the estimation of the GHG emissions at the vehicle model level. Moreover, by combining this information with each vehicle's production year and sale location, we can identify the regional heterogeneity of GBET for the same vehicle model using the VKT data and GHG emission factors of power grids that vary across provinces.

The third category of data was reported at the province level, including annual Vehicle Kilometres of Travel (VKT) for both BEVs and ICEVs and emission intensity of power grids. The VKT data for 2018 were extracted from the National Big Data Alliance of New Energy Vehicles (NDANEV)⁶⁵, which records the real-world driving, charging, and fault status of vehicles car by car. According to the requirements of national standard GB/T 32960, the data are uploaded to the platform every 30 seconds when the vehicle is driving, and the fault state is uploaded every second. Between 2018 and July 17, 2022, the NDANEV accessed 9.27 million new energy vehicles with a total VKT of 295.5 billion kilometres. Although the data are real-world and car-by-car, car-level VKT data were not used in the GBET analysis since we had no access to the vehicle identification information to match the NDANEV database with the CTALI database. Thus, we aggregated the data at the provincial level, assuming that the VKT of vehicles within the same provinces is homogeneous. More statistical information on the real-world VKT data is provided in the SI (see Figure SI-S2).

In the estimation of GBET, the vehicle's VKT data by province change annually. Based upon the real-world data from NDANEV in 2018, we projected the VKT before and after 2018 using two methods: one holds conservative attitudes towards the VKT increase, while the other is more radical. Under the conservative estimation, the targeted VKT in 2030 by province contains five levels, i.e., 18,000 km, 15,000 km, 13,000 km, 12,000 km, and 8,000 km, to reflect the regional heterogeneity in new energy vehicle development speed. The VKT in each province before and after 2018 was then projected linearly, assuming that the VKT increases at distinct speeds across provinces (see Table S10). The radical estimation set a more ambitious VKT goal for 2030, reflecting a possible scenario where passenger BEVs and the associated charging infrastructure in China develop at a dramatic speed (see Table S11).

The emission intensity of electricity generation by province was calculated based on the power generation structure and GHG emission factors across power generation types,

assuming that the electricity consumption structure is the same as that of electricity generation. Such an assumption might underestimate the emission intensity because marginal electricity consumption for BEVs usually relies on coal and natural gas power plants whose operations are relatively stable with higher GHG emission intensities than the grid structure. The provincial power generation structures from 2012 to 2019 were acquired from the China Electricity Council⁶⁶, and those from 2020 to 2028 were from forecasting data referenced from Li et al. (2021)⁶⁷. The GHG emission factors of different power generation technologies (i.e., coal, wind, solar, nuclear, etc.) were referenced from the IPCC Fifth Assessment Report (AR5)⁶⁸. In the basic estimations, we used the medium value reported by IPCC AR5; this value falls within the range of most existing research on GHG emissions from power generation technologies in China (see more literature in Table S12). In the uncertainty analysis, we employed the maximum and minimum values from existing research. The results of the emission intensity of the power grid by province are presented in Table S13-S15.

The last category of data is the life cycle inventory (LCI) data from the latest China Automotive Life Cycle Database (CALCD)-2021 (see Table S16-S17), developed by the CATARC⁶². These data are homogeneous across provinces. We compared the LCI data from CALCD-2021 with two internationally well-known LCI databases, the Greenhouse gases, Regulated Emissions, and Energy use in Technologies Mode (GREET) and ecoinvent 3.6. We found fairly high consistency among the databases (see Table S17).”

Table S9. Variables and data sources used in the estimation.

Category	Resolution	Variables	Time	Description	Range	Data sources	More details
I	Vehicle level (real-world)	Vehicle sold location	Yearly (2012-2018)	The provinces where vehicles are sold	31 provinces ^a	The data of China's Compulsory Traffic Accident Liability Insurance (CTALI), which is provided by the China Automotive Technology & Research Center (CATARC) ^{20,21}	Figure S5-S6.
		Vehicle production location		The provinces where vehicles are produced			
		Battery production location		The provinces where BEV batteries are produced			
		Vehicle model type		Vehicle model types	227 types for BEVs and 1667 types for ICEVs		
II	Vehicle-model level	Curb weight (kg)	Yearly (2012-2018)	The weight of the vehicle, including a full tank of fuel (or battery) and all standard equipment	670-2685 kg for BEVs	Announcement of Vehicle Manufacturing Enterprises and Vehicle Products ²⁴	Figure S7-S9.
		Battery capacity (kWh)		Li-ion battery capacity	645-3200 kg for ICEVs		
		Battery weight (kg)		Battery pack weight	10.56-78.84 kWh		
		Test-based electricity consumption		Official fuel consumption (electricity or fuel)	118-750 kg		

		(kWh/100km)		per 100km based on			
		Test-based fuel consumption (L/100km)		New European Driving Cycle (NEDC) test condition			
III	Provincial level	Vehicle kilometers traveled (km)	Yearly (2012-2030)	Annual mileage traveled of vehicles	678-15927 km/year (2018)	Data for 2018 are obtained from the National Big Data Alliance of New Energy Vehicles (NDANEV) ²⁵ . Data for other years are projected based on assumptions.	Table S10.
		GHG emission factors of electricity generation		Annual GHG emissions per kWh of electricity generation	38-801 gCO ₂ e/kWh (2018)		
IV	Transport-mode level	Life cycle inventories	Constant	Life cycle inventory at different stages of automobile life cycle assessment	N/A	China Automotive Life Cycle Assessment Model (CALCM) ²⁸	Table S16.

Notes: ^a Data for the Hong Kong Special Administrative Region (SAR), Macao SAR, and Taiwan province are unavailable.

Second, for the real-world vehicle data, we provided more statistical descriptions in the SI, to help the readers better understand how the data look like. Below please find some examples.

Figure S2. Sales volume and average VKT of BEVs by province in 2018. Data for the Hong Kong Special Administrative Region (SAR), Macao SAR, and Taiwan province are unavailable.

Figure S6. The number of BEV models per province relative to the provincial market share of national BEV sales, 2012-2018. The horizontal axis is the number of BEV models sold in Mainland China. The vertical axis is the BEVs market share of each province, calculated by dividing the province sales over the national sales. Data for the Hong Kong Special Administrative Region (SAR), Macao SAR, and Taiwan province are unavailable.

Figure S7. Distribution of the vehicle-model specifications for BEVs produced and sold in China from 2012 to 2018. Panel (a) vehicle weight, (b) battery capacity, (c) battery weight, and (d) test-based electricity consumption.

Figure S8. Distribution of the vehicle-model specifications for ICEVs produced and sold in China from 2012 to 2018. Panel (a) vehicle weight, and (b) test-based fuel consumption.

Figure S9. Correlations of vehicle-model specifications between BEVs and ICEVs. Panels on the diagonal line show the statistical distribution of each variable. The lower left shows a bivariate scatter plot with fitted lines and correlation coefficients (***) $p < 0.01$, (**) $p < 0.05$, (*) $p < 0.1$.

Moreover, for the life cycle inventory data, we compared the GHG emission factor data we used (CALCD model) with two other widely-used databases, i.e., the Greenhouse gases, Regulated Emissions, and Energy use in Technologies Mode (GREET) and the ecoinvent database, as Table S17 shows. The GREET is the most representative database of vehicle life cycles in the United States, which simulates energy use and emissions for various fuel combinations, sponsored by the U.S. Department of Energy's (DOE) Office of Energy Efficiency and Renewable Energy[1]. The ecoinvent database is a famous Life Cycle Inventory database that supports many environmental assessments of products and processes worldwide[2]. We find fairly high consistency among the databases (see Table S17).

Table S17. GHG emission factors of materials in CALCD, GREET and Ecoinvent (unit:kgCO₂e/kg).

CALCD		GREET-US (GREET 2020)		Ecoinvent 3.6-China		Ecoinvent 3.6-Global	
Variable	Value	Variable	Value	Variable	Value	Variable	Value
Steel	2.38	Average Steel (recycled steel production: 26.4%)	2.56	N/A	N/A	GLO ^c market for steel, unalloyed	1.82
						GLO market for steel, low-alloyed	1.58
						GLO market for steel, chromium steel 18/8	4.39
Cast iron	1.82	Mix (Final Cast Iron: 85%)	1.01	N/A	N/A	GLO market for cast iron	1.71
Aluminum and aluminum alloy	16.38	Virgin Wrought Aluminum Virgin Cast Aluminum	7.98	N/A	N/A	GLO market for aluminum, cast alloy	5.28
			8.87			GLO market for aluminum, wrought alloy	12.7
Magnesium and magnesium alloys	39.55	Virgin Magnesium	36.49	CN ^b magnesium production, pidgeon process	30.3	GLO market for magnesium-alloy, AZ91	26.9
Copper and copper alloys	4.23	Mix: Copper (chilean copper:15.8%)	2.88	N/A	N/A	GLO market for copper	4.65
Plastics average	4.26	Plastics average	2.47	N/A	N/A	Plastics average	4.05
Lithium nickel cobalt manganate	17.4	NMC ^a (622): Coprecipitation	19.78	N/A	N/A	N/A	N/A

CALCD		GREET-US (GREET 2020)		Ecoinvent 3.6-China		Ecoinvent 3.6-Global	
Variable	Value	Variable	Value	Variable	Value	Variable	Value
Graphite	5.48	Graphite	4.81	CN: anode production, graphite, for lithium-ion battery	3.65	GLO market for anode, graphite, for lithium-ion battery	3.66
Electricity	0.635	Distributed - U.S. Mix	0.45	CN market group for electricity, high voltage	1.01	N/A	N/A
				CN market group for electricity, medium voltage	1.05		
				CN market group for electricity, low voltage	1.06		
Gasoline	0.651	E10: Reformulated Gasoline (E10) Blending and Transportation to Refueling Station	0.56	N/A	N/A	N/A	N/A
Diesel	0.644	Low-Sulfur Diesel from Crude Oil	0.44	N/A	N/A	GLO market group for diesel	0.486

Note: ^aNMC: Lithium Nickel Cobalt Manganese Oxide; ^bCN: China; ^cGLO: Global.

- Lines 408-409: Revise the sentence.

Thanks, we have rewritten the whole Methods section. The relevant text now reads (Lines 475-477):

“The VKT data for 2018 were extracted from the National Big Data Alliance of New Energy Vehicles (NDANEV)⁶⁵, which records the real-world driving, charging, and fault status of vehicles car by car.”

- Section 441: Are the influencing factors described in the section beginning in line 203? If yes, please specify. The sensitivity analysis, which is a perturbation analysis (even though the method is not identified by the authors and is referred to as an elasticity index), does not present the reasoning for assessing the selected parameters. Table S2 is missing the units of the current and new parameters.

Thanks for the comment, we clarified the approach of the sensitivity analysis used in our study. The relevant text reads, in Lines 525-526:

“We performed a one-variable-at-a-time perturbation sensitivity analysis of the input parameters (Table S6) that influence the GBET.”

Moreover, we expanded the perturbation sensitivity analysis to all possible influencing factors in the revision. However, the number of these factors is over a hundred, and most of them have subtle influencing effects on the results. Thus we group them into ten groups (see more details of the parameters included in each of the groups in Table S6). The sensitivity analysis results for the ten groups of parameters are present in the table below and in SI Table S7 (unit has been added).

Table S6. Variables considered in the sensitivity analysis.

Grouped variable	Life cycle inventories included	More details
Curb weight	N/A	Refer to Figure S7
Battery capacity		
Battery weight		
Test-based fuel consumption		Refer to Table S13-S15.
GHG emission factors of the power grid		
Annual vehicle kilometers traveled (VKT)	Refer to Table S10-S11.	
GHG emission factors of vehicle material production	Vehicle Components (%)	Refer to Table S16.
	Tire Components (%)	
	Lead-acid Battery Materials Components (%)	
	Fluids Components(%)	
	Material GHG emission factor (kgCO ₂ e/kg)	Refer to Table S17.
GHG emission factors of battery material production	Li-ion Battery Materials Components(%)	Refer to Table S16.
	Material GHG emission factor (kgCO ₂ e/kg)	Refer to Table S17.
Electricity consumption during the vehicle production stage	Electricity (kWh per vehicle)	297-345
	Curb weight (kg)	Refer to Figure S7
Electricity consumption during the battery production stage	Electricity (kWh/kWh)	5.8
	Battery capacity (kWh)	Refer to Figure S7

Table S7. Sensitivity coefficients of the influencing factors.

Influencing factors	Life-cycle stage	Change in influencing factors	GBET change rate	Sensitive coefficient
Curb weight (kg)	Vehicle cycle	5.0%	8.5%	1.7
Vehicle Material GHG Emission Factor (kgCO ₂ e/kg)	Vehicle cycle	5.0%	6.0%	1.2
Battery capacity (kWh)	Vehicle cycle	5.0%	5.5%	1.1
Battery Material GHG Emission Factor (kgCO ₂ e/kg)	Vehicle cycle	5.0%	4.7%	0.9
Vehicle kilometers traveled (km/year)	Fuel cycle	5.0%	4.6%	0.9
GHG emission intensity of power grid (kgCO ₂ e/kWh)	Fuel cycle and Vehicle cycle	5.0%	2.4%	0.5
Battery weight (kg)	Vehicle cycle	5.0%	1.8%	0.4
Test-based fuel consumption (kWh/100km)	Fuel cycle	5.0%	1.6%	0.3
Vehicle Production_Electricity (kWh/kg)	Vehicle cycle	5.0%	0.6%	0.1
Battery Production__Electricity (kWh/kWh)	Vehicle cycle	5.0%	0.6%	0.1

Note: The sensitivity analysis is performed using the one-variable-at-a-time perturbation approach. The sensitivity coefficient refers to the change in GBET for every 1% change in the influencing factor.

Additional comments:

- Please revise the use of acronyms in the Supplementary Material (SM). This is an independent document and must present be elaborated accordingly.

Thanks for the suggestion, we have added explanations for the abbreviations "GBET", "BEV", "ICEV", etc., in the Supplementary Material.

- I identified some typos in the SM, such as "emisons".

Thanks for pointing this out. We have corrected the typos and asked professional English language editing services to proofread the manuscript.

- *What is the legend of Figure S3 implying? Vehicle imports percentage?*

Yes, it is the vehicle imports percentage. However, given the major revision, we have removed the original Figure S3 and presented similar information in a newly added Figure.

Figure S5. Production volume of BEV batteries and vehicles by province from 2012-2018. a. Battery production by province. b. Vehicle production by province. Data for the Hong Kong Special Administrative Region (SAR), Macao SAR, and Taiwan province are unavailable.

- *The additional results are never mentioned in the main manuscript. Reference to its content should be provided.*

Many thanks. We have double-checked to make sure that the reference to all the supplementary materials has been provided in the main text.

- *It is unclear if the authors accessed the end-of-life of the vehicles. Please clarify this aspect.*

Thanks for the comment. We excluded the effect of battery recycling and secondary application from our estimations for the following reasons: As the estimation of GBET focuses on the GHG debt in the production phase and the paid-back GHG before the break-even point comes, the battery's end-of-life strategies is actually beyond the scope of estimation. If considered, they would affect the production phase by reducing the raw material use. However, as the main task of this manuscript is to estimate the GBET of BEVs in the Chinese market from 2012 to 2018, when the electric cars are just starting to become popular in China, few recycling and secondary used batteries were used in the battery production (Table R3 showing the timeline of policy releases related to electric vehicle battery recycling). As such, for the period we estimated, excluding the end-of-life phase has very limited impact on the results.

Table R3. Policies related to electric vehicle battery recycling.

Document	Issuing Agency	Time	Reference URL
Interim Measures for the Management of New Energy Vehicle Power Battery Recycling (in Chinese)	Ministry Information Technology People's Republic China; Ministry of Science and Technology of the People's Republic of China et al.	Feb 2018	http://www.gov.cn/xinwen/2018-02/26/content_5268875.htm
Interim regulations on traceability management of new energy vehicle power battery recycling (in Chinese)	Ministry Information Technology People's Republic China	July 2018	http://www.huaibin.gov.cn/wzd/webinfo/2018/08/1534706988493194.htm
Notice on the new energy vehicle power battery recycling pilot work (in Chinese)	Ministry Information Technology People's Republic China; Ministry of Science and Technology of the People's Republic of China et al.	July 2018	http://www.gov.cn/zhengce/zhengceku/2018-12/31/content_5440228.htm
People's Republic of China Solid Waste Pollution Prevention and Control Law (in Chinese)	National People's Congress Standing Committee	April 2020	http://www.npc.gov.cn/npc/c30834/202004/b6cf2e57b63b47818a275a37819c6b02.shtml

- *At any moment, the manuscript discusses other barriers to BEV deployment. Even though climate change is a great driver of the decisions, other impact categories, such as mineral*

resources depletion, and the use of critical raw materials, are highly relevant and might delay their production and availability in the market.

Many thanks! We added this point from a different perspective, as shown in Lines 309-314:

“Another method is to accelerate GHG debt repayment by intensifying the usage of existing BEVs via car sharing or prioritizing BEVs for taxis⁴⁵. Intensifying BEV use rather than expanding vehicle ownership would simultaneously shorten GBET, achieve better GHG emission reductions and solve other problems, such as traffic congestion, mineral resources depletion, infrastructure construction pressure and environmental pollution^{46,47}. This strategy is feasible, as essentially what people truly need is high-quality transportation service rather than the vehicle itself⁴⁸.”

We appreciate the reviewer's substantial efforts and contributions to our manuscript. Below please find the references mentioned in the responses.

References

1. laboratory, A.n. *The Greenhouse gases, Regulated Emissions, and Energy use in Technologies Model*. 2022; Available from: <https://greet.es.anl.gov/>.
2. ecoinvent. *ecoinvent Database*. Available from: <https://ecoinvent.org/the-ecoinvent-database/>.

REVIEWERS' COMMENTS

Reviewer #2 (Remarks to the Author):

The authors have answered my questions. Also, I see the authors have made significant effort to provide additional information to address other reviewers' comments.

Reviewer #3 (Remarks to the Author):

Dear Editors and authors,

I believe the authors made a great effort to review the manuscript, and its quality has improved significantly. As a result, I believe the manuscript offers a high-quality contribution to the deployment of electric vehicles.

The methodology is now clear, and the assumptions are reproducible. The sensitivity analysis also improved.

My only comments at this point are:

- I recommend a revision in the reference list. In the current version, some references are reported incorrectly (e.g., references 12, 14, 21, among others, in the Supplementary Material).
- Some references are not reported at all, e.g., the LCI databases mentioned in lines 516-519 of the main manuscript.
- The references to standards are not done.
- The functional unit for the Life Cycle Assessment should be explicitly defined in the manuscript.

Reviewer #2

- The authors have answered my questions. Also, I see the authors have made significant effort to provide additional information to address other reviewers' comments.

Many thanks to the reviewer for appreciating our substantial efforts in the revision. We are grateful that the manuscript is much improved with the great help from the editors and the reviewers.

Reviewer #3

Dear Editors and authors,

I believe the authors made a great effort to review the manuscript, and its quality has improved significantly. As a result, I believe the manuscript offers a high-quality contribution to the deployment of electric vehicles.

The methodology is now clear, and the assumptions are reproducible. The sensitivity analysis also improved.

We really appreciate the reviewer's positive feedback on the quality and contribution of this work. Please find our detailed response to the latest comments below. The accompanying manuscript has been amended accordingly (changes highlighted in yellow).

My only comments at this point are:

- I recommend a revision in the reference list. In the current version, some references are reported incorrectly (e.g., references 12, 14, 21, among others, in the Supplementary Material).

Many thanks for the helpful comments. We have carefully checked and corrected the reference list, including references 12, 14 & 21.

12. China Association of Automobile Manufacturers. Industry Operation | Economic operation of the automotive industry in 2021 <http://www.caam.org.cn/chn/4/cate_154/con_5235337.html> (2022).

14. The data of national motor vehicles and drivers for the first half of 2022 released by the Ministry of Public Security (in Chinese). Road traffic management, 5 (2022)

21. The International Council on Clean Transportation. Effects of battery manufacturing on electric vehicle life-cycle greenhouse gas emissions.

<<https://theicct.org/publication/effects-of-battery-manufacturing-on-electric-vehicle-life-cycle-greenhouse-gas-emissions/>> (2018).

- Some references are not reported at all, e.g., the LCI databases mentioned in lines 516-519 of the main manuscript.

The missing references have been added as well. For example, references 74 & 75 were added for the LCI database.

74. Wernet, G. et al. The ecoinvent database version 3 (part I): overview and methodology. *Int. J. Life Cycle Assess.* 21, 1218–1230 (2016).

75. Argonne National Laboratory. The Greenhouse Gases, Regulated Emissions, and Energy Use in Transportation (GREET®) Model - GREET 2, Version 2020. <https://greet.es.anl.gov/> (2020).

- The references to standards are not done.

Thanks for the helpful observations. References 61-64 have been added for the standards.

61. ISO 14040:2006(en) Environmental management-Life cycle assessment-Principles and framework. <<https://www.iso.org/obp/ui/#iso:std:iso:14040:ed-2:v1:en>> (2006).

62. GB/T 24044-2008 Environmental management - Life cycle assessment - Requirements and guidelines. <<https://openstd.samr.gov.cn/bzgk/gb/newGbInfo?hcno=329770D2F0539B875B094A56C308EC4E>> (2008).

63. GB/T 24040-2008 Environmental management - Life cycle assessment - Principles and frameworks. <<https://openstd.samr.gov.cn/bzgk/gb/newGbInfo?hcno=49702C7C8B0EB91E7E62D63CB55292A4>> (2008).

64. ISO 14067:2018 Greenhouse gases-Carbon footprint of products-Requirements and guidelines for quantification. <<https://www.iso.org/standard/71206.html>> (2018).

- The functional unit for the Life Cycle Assessment should be explicitly defined in the manuscript.

Many thanks for the reminder. We added a sentence to define the functional unit for the Life Cycle Assessment. Specifically, in Lines 357-358:

“The functional unit of this LCA is 1 km travelled by a passenger vehicle in China during 11 years.”